# ALETHEIA: A MULTI-FREQUENCY EDDY CURRENT PULSED THERMOGRAPHY DATASET FOR NEURAL OPERATOR LEARNING IN NONDESTRUCTIVE TESTING

## ABSTRACT

Learning neural solvers for spatiotemporal partial differential equations (PDEs) under real-world constraints remains a key challenge in scientific machine learning, especially for inverse tasks with sparse and noisy boundary observations. We present the **Aletheia** dataset, the first 3D benchmark for learning data-driven solvers in the context of **nondestructive testing (NDT)**. The dataset simulates eddy-current-induced heating in conductive solids and models the resulting transient heat propagation governed by the heat equation. Aletheia contains over 4,700 high-resolution samples across 10 excitation frequencies (1-100 kHz), each providing volumetric heat source and temperature fields over time. It supports both forward prediction of temperature evolution and inverse reconstruction of internal heat sources or defects from surface infrared measurements. Real infrared thermography data from cracked rail specimens are included for calibration and generalization studies. We define three canonical tasks on both regular and irregular grids and benchmark them using various neural operators. Aletheia establishes a unified platform for evaluating neural PDE solvers under realistic NDT conditions, enabling progress in reliable, data-driven inverse modeling.

## 1 INTRODUCTION

Neural operator methods, such as the Fourier Neural Operator (FNO) (Li et al., 2021, 2023b; Tran et al., 2023; Xiao et al., 2024) and Transformer-based solvers (Li et al., 2023a; Wu et al., 2024; Lee & Oh, 2024), have emerged as a transformative approach for learning solution operators of partial differential equations (PDEs) directly from data. Unlike traditional methods, these architectures bypass mesh-dependent discretizations, enabling robust generalization across parameterized PDE families. However, they are typically evaluated on academic datasets (e.g., Darcy flow, Navier–Stokes) with fully observed fields and simplified geometries, which fail to capture the complexities of real-world inverse problems. In applications like nondestructive testing (NDT) (Gupta et al., 2022; Xiong et al., 2023; Yuan et al., 2021; Gong et al., 2022; Tuschl et al., 2021), challenges such as sparse or noisy boundary observations, unknown source terms, and heterogeneous domains demand more robust benchmarks (Molinaro et al., 2023; Azizzadenesheli et al., 2024).

In NDT, reconstructing hidden defects (Lin et al., 2023; Zhao et al., 2022; Tao et al., 2022; Wu et al., 2021) from surface temperature measurements, as in inverse heat conduction problems (Silva et al., 2023), is inherently ill-posed: distinct subsurface defects or excitation conditions can produce nearly identical surface temperature patterns (Woodbury et al., 2023), as illustrated in Figure 1. To address this, we employ multi-frequency pulsed induction heating, where different excitation frequencies probe the material at varying depths—lower frequencies penetrate deeper to capture internal defect responses, while higher frequencies reveal surface-level thermal behavior (Liang et al., 2024). As shown in Figure 1, while some frequencies (e.g., 25 kHz) may yield similar surface temperatures for different defects, others (e.g., 9 kHz) reveal distinct patterns. This frequency-dependent response diversity breaks single-frequency ambiguity, enhancing defect discriminability.

Existing PDE-learning benchmarks lack realistic thermal-boundary coupling and 3D scenarios tailored to heat-source inversion or volumetric temperature prediction in NDT. To bridge this gap, we introduce the **Aletheia** dataset (Figure 2), a comprehensive 3D benchmark that integrates high-fidelity

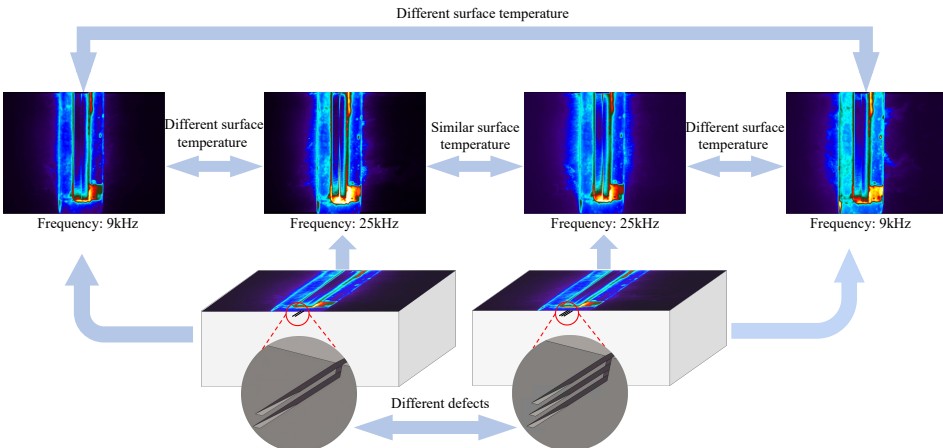

Figure 1: **Multi-frequency stimulation in the Aletheia dataset.** Different defects may produce similar surface temperatures under the same frequency (25 kHz), but show clear differences at another frequency (9 kHz). Using multiple frequencies helps distinguish hidden defects by capturing varied thermal responses at different depths.

simulations with real infrared measurements from rail specimens with internal fatigue cracks. The dataset includes over 4,700 defect cases across 10 excitation frequencies (1–100 kHz), providing volumetric heat-source maps (eddy-current-induced Joule heating) and time-resolved temperature fields on both regular and irregular grids. Calibrated thermography captures transient surface temperature sequences, while multi-frequency conditions supply depth-sensitive signals to mitigate the ill-posedness of surface-only observations.

Using Aletheia, we address three key tasks in eddy current thermography and PDE benchmarking: (1) **forward thermal prediction** of full 3D temperature evolution from known sources; (2) **inverse source reconstruction** of latent heat distributions or defect geometries from sparse surface data; and (3) **out-of-distribution (OOD) generalization** to unseen frequencies, defect shapes, and material variants. Overall, our contributions are summarized as follows:

- We present the first publicly available simulation dataset **Aletheia** in the context of electromagnetic-thermal coupling, enabling the datatization of eddy current thermography.

- Aletheia provides a multi-frequency dimension: data covering a range of excitation frequencies from low to high such as 1—100 kHz, capturing the effect of frequency on the depth and effectiveness of the heat.

- Aletheia contains three-dimensional, temporally-evolving data, such as the evolution of the entire temperature field after pulse heating, not just steady-state or two-dimensional observations

- Aletheia combines high-fidelity simulations and experimental measurements. Simulation data provide comprehensive information on field distributions and "true value" defects, while experimental data introduce real noise and variability and verify the reliability of the simulation.

- Built around the real engineering application of rail crack detection, Aletheia covers a wide range of defect types and sizes with direct engineering relevance.

## 2 BACKGROUND

Neural operators have revolutionized data-driven PDE modeling by learning mapping between function spaces (Lu et al., 2021; Li et al., 2021, 2020), but existing benchmarks remain narrowly focused on fully observed, forward-only problems and lack realistic inverse or measurement-sparse scenarios. Eddy-current thermography(ECT) offers a rich real-world setting in NDT, yet no public

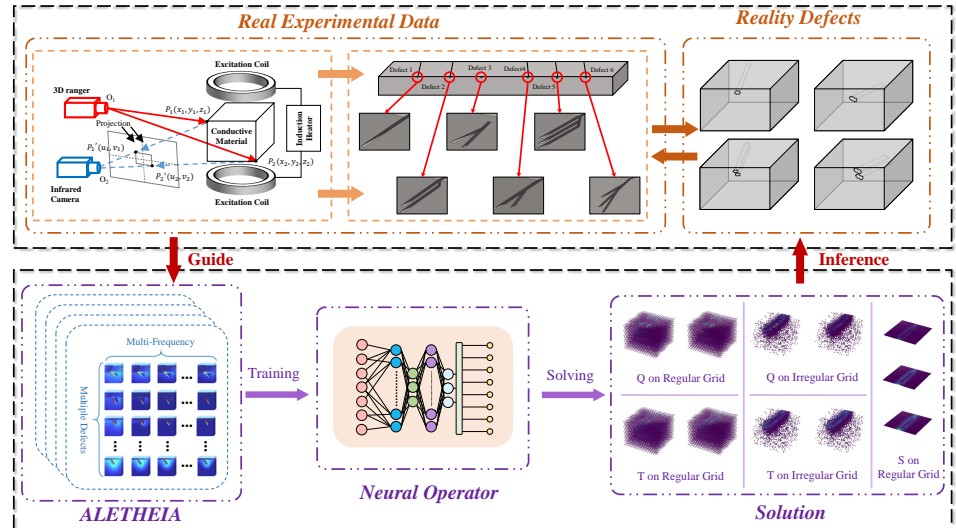

Figure 2: **Overview of the Aletheia dataset.** *Top*: real infrared thermography sequences from rail specimens with fatigue cracks. *Bottom*: synthetic data pipeline generating multi-frequency (1—100 kHz) volumetric heat sources and corresponding transient temperature fields for 4,700+ defect cases, on both regular and irregular grids. This combined sim-to-real benchmark supports forward prediction, inverse reconstruction, and cross-frequency evaluation in NDT.

3D dataset exists for learning heat-field inversion under varying excitation. We therefore position our dataset at the intersection of these gaps, enabling the study of both forward prediction and inverse source reconstruction for transient heat conduction from surface measurements.

## 2.1 CHALLENGES AND APPLICATIONS OF NEURAL PDE SOLVERS

Neural operator methods (e.g., DeepONet (Lu et al., 2021), FNO (Li et al., 2021)) learn PDE solution operators directly from data, achieving mesh- and resolution-invariance and outperforming classical surrogates on benchmarks like Darcy flow and NavierStokes (Takamoto et al., 2022). Despite their promise, these models struggle to capture high-frequency components due to spectral truncation in Fourier layers, exhibit poor extrapolation to OOD parameters, and lack robustness under noisy or incomplete boundary observations common in inverse problems. Moreover, most evaluations assume full-field availability, whereas many applications demand reconstructing latent sources or fields from sparse measurements.

## 2.2 THERMAL HOLOGRAPHY FOR NDT

ECT (Gao et al., 2024; Zou et al., 2022; Zu et al., 2023) combines electromagnetic induction heating with infrared imaging to detect subsurface defects by capturing surface temperature anomalies. A common implementation, pulsed ECT (ECPT) (Zhang et al., 2024; Chen et al., 2021), uses short bursts of alternating current to induce volumetric Joule heating in conductive materials, enabling high-sensitivity, non-contact inspection of internal structures (Yin et al., 2021; Ma et al., 2024; Zhang et al., 2021). This process gives rise to the concept of thermal holography (Utadiya et al., 2023), where internal heat sources perturbed by defects such as cracks are inferred from transient surface temperature fields. Mathematically, this is governed by the heat equation with a volumetric heat source:

$$\frac{\partial u}{\partial t} = \alpha \nabla^2 u + q, \tag{1}$$

where $u(\mathbf{x}, t)$ is the temperature field at position $\mathbf{x} = (x, y, z)$ and time $t$, $\alpha$ is the thermal diffusivity, and $q(\mathbf{x}, t)$ represents the internal heat sources induced by eddy currents. The inverse problem in thermal holography seeks to reconstruct the spatial distribution of the heat source $q(x, y, z)$ from sparse and noisy surface temperature measurements $u(x, y, z_s, t)$, where $z_s$ denotes the surface of the domain $\partial \Omega$, and the observation set is defined over coordinates $(x, y, z_s, t)$. In many practical

settings, the primary goal is to reconstruct the volumetric distribution of these internal heat (Hong et al., 2023; Wang et al., 2021) sources rather than directly visualize the defects themselves. This inverse heat conduction task is inherently ill-posed and only sparse, noisy boundary data are typically available. Despite its critical role in nondestructive testing across domains such as rail, aerospace (Gebrehiwet et al., 2023; Gholizadeh & Gholizadeh, 2022; Jacob & Raddatz, 2022), and pipeline inspection (Cheng et al., 2021; Wang et al., 2024), there exists no standardized benchmark dataset for learning data-driven solvers that can tackle this class of inverse thermal problems under realistic conditions.

## 2.3 LIMITATION OF EXISTING PDE BENCHMARKS

As summarized in Table 1, most existing PDE benchmark datasets exhibit limited diversity in terms of task settings and data characteristics. Specifically, widely used datasets such as *Darcy Flow*, *NavierStokes*, *Burgers' Equation*, and *Shallow Water* primarily focus on forward and inverse problems in regular 2D geometries with full observations and lack support for more complex learning scenarios. Among commonly used benchmarks (Takamoto et al., 2022; Herde et al., 2024; Dulny et al., 2023) we surveyed none of these datasets simultaneously offer partial observability and multi-task evaluation capabilities. Although the *FWI-F/L/FL* (Zhu et al., 2023) suite introduces inverse modeling and partial observations, it remains confined to 2D domains with relatively simple geometries and lacks support for multitask learning. Similarly, *BubbleML* (Hassan et al., 2023) is an excellent multiphase, multiphysics dataset, yet it still lacks testing capabilities for inverse problems and OOD problems. Furthermore, mainstream and widely used benchmarks predominantly focus on low spatial dimensions(1D or 2D), which falls short of the complexity found in real-world scientific and engineering problems that often involve high-dimensional, irregular domains with heterogeneous observability and spatiotemporal dynamics.

Table 1: Comparison of PDE benchmark datasets. Each checkmark ($\checkmark$) indicates the presence of a specific feature in the dataset. *Spatial Dim.* denotes the predominant dimensionality used in common benchmarks, not a limitation of the PDE itself.

| Benchmark Dataset | Spatial Dim. | Inverse | Partial Obs. | Irregular Geo. | Multi-task | OOD |
|---|---|---|---|---|---|---|
| Advection | 1D | $\checkmark$ | $\times$ | $\times$ | $\times$ | $\checkmark$ |
| Darcy Flow | 2D | $\checkmark$ | $\times$ | $\times$ | $\times$ | $\checkmark$ |
| NavierStokes | 1D/2D/3D | $\checkmark$ | $\times$ | $\times$ | $\times$ | $\checkmark$ |
| Burgers' Equation | 1D/2D | $\checkmark$ | $\times$ | $\times$ | $\times$ | $\checkmark$ |
| Airfoil Flow | 2D | $\checkmark$ | $\times$ | $\checkmark$ | $\times$ | $\times$ |
| Diffusion Reaction | 1D/2D | $\checkmark$ | $\times$ | $\times$ | $\times$ | $\checkmark$ |
| Shallow Water | 2D | $\times$ | $\times$ | $\times$ | $\times$ | $\times$ |
| Plasticity / Elasticity | 2D/3D | $\times$ | $\times$ | $\checkmark$ | $\times$ | $\times$ |
| BubbleML | 2D/3D | $\times$ | $\times$ | $\checkmark$ | $\checkmark$ | $\times$ |
| FWI-F / L / FL | 2D | $\checkmark$ | $\checkmark$ | $\times$ | $\times$ | $\checkmark$ |
| **Aletheia (Ours)** | 3D | $\checkmark$ | $\checkmark$ | $\checkmark$ | $\checkmark$ | $\checkmark$ |

These limitations hinder comprehensive evaluation of model generalization, robustness, and versatility. In contrast, our proposed dataset, **Aletheia**, is designed to fill this gap by incorporating inverse tasks, partial and sparse observations, irregular 3D geometries with temporal dynamics, multi-task learning, and OOD generalization, providing a more realistic and challenging benchmark for PDE learning.

## 3 DATASET CONSTRUCTION AND DETAILS

### 3.1 OVERVIEW OF THE DATASET

Aletheia is derived from two complementary sources. The first source consists of high-fidelity multi-physics simulations, which generate synthetic data by numerically solving a fully coupled electric–magnetic–thermal transient process. The second source comprises experimental measurements, where actual data are collected via frequency-swept pulsed eddy current thermography experiments performed on steel rail samples containing artificial defects. Due to the scarcity and complexity of real defect specimens, experimental data primarily serve to calibrate and validate the simulation

model. Conversely, simulated data are extensively utilized for model training as they offer complete three-dimensional temperature field evolutions, internal heat source distributions, and precise defect geometries—details which real ECPT experiments cannot provide, being limited to surface temperature time series measurements. Recognizing that the real experimental environment closely mimics authentic engineering inspection scenarios, we meticulously calibrated our simulation models. Specifically, we fine-tuned material properties and excitation parameters so that simulated surface temperature curves align closely with experimental measurements, achieving an accuracy within ±1°C. This rigorous calibration ensures both the reliability and realistic nature of simulated data.

## 3.2 SIMULATION DATA ACQUISITION

We used COMSOL Multiphysics[1] to develop a three-dimensional finite element simulation pipeline modeling the coupled process of electromagnetic induction heating and thermal diffusion. For each simulated sample, we first constructed a 3D rail model containing a defect and set material properties such as electrical conductivity $\sigma$, thermal conductivity $k$, and relative permeability $\mu_r$. We then specified the defect geometry and input parameters, including the excitation frequency $f$ and coil current. After the model was built, an adaptive mesh was generated with finer elements in the defect region due to the defect's small size to ensure computational accuracy. The simulation employed a 600 ms pulsed heating process. For more detailed simulation parameter settings, please refer to Appendix A. When an alternating current pulse at the specified frequency was applied to an excitation coil fixed just above the rail, it induced eddy currents in the metal specimens according to Faraday's law. Meanwhile, according to Ampere's Law, the magnetic field generated by induced eddy currents in turn affects the original field. The Maxwell equations can describe this electromagnetic process:

$$\nabla \cdot \mathbf{D} = \rho, \quad \nabla \cdot \mathbf{B} = 0, \quad \nabla \times \mathbf{E} = -\frac{\partial \mathbf{B}}{\partial t}, \quad \nabla \times \mathbf{H} = \mathbf{J} + \frac{\partial \mathbf{D}}{\partial t}. \tag{2}$$

where $\mathbf{D}$ denotes the electric displacement vector, $\rho$ is the free charge density; $\mathbf{B} = \mu\mathbf{H}$ represents the magnetic induction intensity; $\mathbf{H}$ is the magnetic field intensity; $\mathbf{E}$ is the electric field intensity; $\mathbf{J} = \sigma\mathbf{E}$ indicates the eddy current density. Eddy current inside the conductor generated Joule heat due to resistance, acting as an internal heat source: $q(x, y, z) = \frac{|\mathbf{J}|^2}{\sigma}$. After converting this electromagnetic energy into thermal energy, it diffused through thermal conduction in the material.

The depth of eddy current penetration, known as the skin depth, is frequency-dependent and given by: $\delta = \sqrt{\frac{1}{\pi f \mu \sigma}}$, where $\mu = \mu_r \mu_0$ is the magnetic permeability ($\mu_0$ is the permeability of free space). Higher frequencies reduce $\delta$, concentrating eddy currents and heat near the surface, while lower frequencies allow deeper penetration, enabling differentiation of subsurface defect responses. To overcome the inherent ill-posedness of mapping surface temperature back to defect characteristics, we employ multiple excitation frequencies in the pulsed heating process—while some frequencies may yield indistinguishable temperature profiles for certain defect types, others reveal distinct thermal distributions, enabling us to accurately distinguish different types of defects. The electromagnetic and thermal phenomena within our simulations were fully coupled, mutually influencing one another throughout the transient process. Upon completion of each simulation, we systematically recorded the temporal evolution of the temperature field $u(x, y, z, t)$ across the specimen's surface and internal volume, as well as the spatial distribution of the internal heat sources $q(x, y, z)$ generated by induced eddy currents.

## 3.3 EXPERIMENTAL DATA ACQUISITION

We additionally performed pulsed eddy current thermography experiments to acquire authentic measurement data. The experimental apparatus comprised a custom-built XZ-series DSP-controlled inverter power supply paired with a specially designed excitation coil featuring a central slot, as depicted in Figure 3. During each experimental run, a sinusoidal pulse current with a duration of 600 ms was applied at selected frequencies within the range of 1—100 kHz. The excitation coil was maintained at a constant lift-off distance of 5 mm above the steel rail sample. A high-resolution FLIR SC6550A infrared thermal imager was employed to capture the evolution of surface temperatures, providing imagery at a spatial resolution of 640×480 pixels, a frame rate of 50 frames per second, and

---

[1]https://www.comsol.com

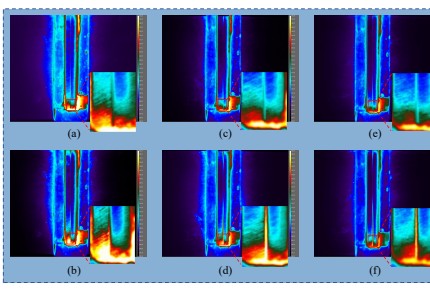

(a) Experimental setup for pulse eddy current thermal imaging.

(b) Thermal imaging results for different defect types and excitation frequencies.

Figure 3: Overview of the pulse eddy current thermal imaging experimental platform and selected thermal imaging results.

an intensity depth of 14 bits. This allowed detailed recording of the temporal temperature distribution throughout both the heating and cooling phases. The collected experimental data served to introduce realistic ambient noise into our dataset and was used to rigorously validate the accuracy and reliability of our simulation model.

## 3.4 DATASET STRUCTURE

Leveraging our calibrated simulation framework, we systematically generated a comprehensive set of defect samples using automated scripts and parallelized computations. After extensive simulations and rigorous calibration processes, the final dataset comprises a total of 4,782 unique defect instances, categorized into 2,407 open-crack and 2,375 closed-crack samples. Detailed statistics for each defect type are summarized in Table 2. Each defect instance includes simulation data captured across ten discrete excitation frequencies—specifically, 1 kHz, 4 kHz, 9 kHz, 16 kHz, 25 kHz, 36 kHz, 49 kHz, 64 kHz, 81 kHz, and 100 kHz—to reflect varied depth sensitivities and thermal responses. For versatility across different modeling approaches, the thermal and temperature fields within each defect instance were sampled using two distinct strategies: regular grid sampling and irregular point sampling. Regular grid sampling aligns directly with the structured grid employed by experimental infrared imagery, providing consistent surface temperature time-series data for model input. Conversely, the irregular sampling strategy mimics the actual positioning of defects relative to the infrared camera used in experimental setups. These irregularly sampled points follow a multivariate Gaussian distribution concentrated around the central surface line of the specimen, accurately reflecting defect locations encountered in practical inspections. Points near defect regions are sampled densely to ensure higher resolution and precise reconstruction of defect contours, whereas areas less relevant to defect detection are sampled more sparsely, thus reducing computational overhead. This carefully designed sampling method optimally balances computational efficiency with reconstruction accuracy.

Table 2: Number of different types of defects. There are a total of 6 types of defects, please refer to Appendix C for more detailed information on each type of defect.

| Single layer | Type I double-layer | Type II double-layer | Type III double-layer | Type I multi-layer | Type II multi-layer | Total |
|---|---|---|---|---|---|---|
| 88 | 1050 | 69 | 2285 | 90 | 1200 | 4782 |

## 4 BENCHMARK

To systematically assess the modeling capabilities of neural operators in 3D spatiotemporal heat transfer problems, we construct a comprehensive benchmark suite based on the Aletheia dataset. This benchmark spans six task settings, encompassing both *Same-Frequency* and *Out-of-Distribution* scenarios. Each task is evaluated on two types of spatial grids: regular and irregular to rigorously

test the generalization and robustness of neural operator models across diverse spatial discretizations. We benchmark several representative neural operator architectures, including FNO and Transolver, under consistent training and evaluation protocols. This unified framework offers a strong baseline for future research and provides critical insight into the strengths and limitations of current neural PDE solvers in realistic NDT contexts.

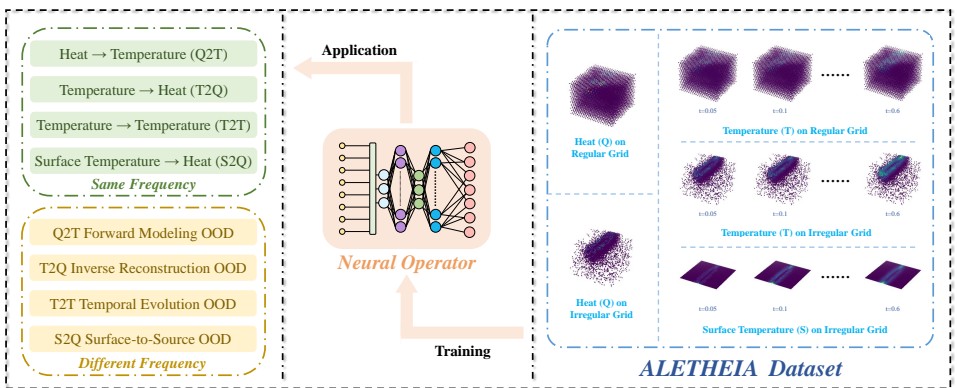

Figure 4: Our benchmark covers tasks such as heat source to temperature, temperature reconstruction, surface temperature reconstruction, and out-of-distribution detection at different frequencies.

## 4.1 TASK SETUP

We organize the tasks into two categories according to the alignment between training and testing distributions:

**1. Full-frequency tasks (in-distribution / closed-set)**   These tasks evaluate the accuracy of the model when the training and testing data are sampled from the same excitation frequencies and boundary conditions:

- **Forward Modeling (Q2T):** Learn the mapping $q(\mathbf{x}, t) \mapsto u(\mathbf{x}, t)$, i.e., predict the transient temperature field given the internal heat source.
- **Inverse Source Reconstruction (T2Q):** Learn the inverse mapping $u(\mathbf{x}, t) \mapsto q(\mathbf{x}, t)$, recovering the latent volumetric heat source from the transient temperature field $u(x, y, z, t)$.
- **Temporal Evolution Prediction (T2T):** Learn $\{u(\mathbf{x}, t_i)\}_{i=0}^{10} \mapsto \{u(\mathbf{x}, t_j)\}_{j=11}^{12}$, forecasting future temperature fields based on historical states.
- **Surface-to-Source Reconstruction (S2Q):** Learn $s(\mathbf{x}_\partial, t) \mapsto q(\mathbf{x}, t)$, Reconstructing the implicit volumetric heat source from the top surface temperature trace $s(x, y, z, t)$, when the values of $z$ are 0.

**2. Out-of-distribution (OOD) generalization tasks**   These tasks introduce controlled distributional shifts to test generalization under unseen physical conditions. In ALETHEIA we instantiate three physically motivated OOD regimes, each applied to the forward, inverse, temporal, and surface-to-source tasks:

- **Frequency-band OOD (Low/Mid/High):** The 10 excitation frequencies are partitioned into low, mid, and high bands (e.g., $1/4$ kHz, $25/36$ kHz, $81/100$ kHz). For each band, it trains on the remaining frequencies and tests only on the held-out band, probing the ability of models to extrapolate across qualitatively different penetration depths.
- **Single-frequency OOD (SFO):** For a given defect and grid type, the model is trained on a *single* excitation frequency and evaluated on the remaining frequencies. This setting isolates how much multi-frequency supervision is needed for stable inverse reconstruction, and quantifies the degradation when moving from multi-frequency to single-frequency training.

- **Geometric OOD (fsplit):** The model is trained on samples from five defect types and tested on the held-out sixth type (e.g., Type II multi-layer defect). This evaluates extrapolation to unseen defect geometries while keeping the excitation protocol fixed.

To further enhance diversity and realism, each task is evaluated on both **regular** and **irregular** spatial grids. This enables us to systematically investigate the robustness of neural operators under varying spatial discretization schemes, mimicking real-world sensing constraints.

## 4.2 EXPERIMENTAL SETUP

We benchmark all models on both regular and irregular 3D geometries using simulations from all defect types. Each sample was downsampled to 8000 points to achieve 2D-3D mapping alignment between surface temperatures and internal heat sources. Each experiment involved 600 samples. Prior to training, all input data were normalized using global statistical measures.

To handle the dataset, we implement a VTU parser that supports streaming and normalization, and define one in-distribution and three OOD partitioning schemes. In the normal setting, all 10 excitation frequencies are randomly shuffled at the sample level. For frequency-band OOD, we hold out one band (Low: 1/4 kHz, Mid: 25/36 kHz, High: 81/100 kHz) for testing and train on the remaining frequencies. For single-frequency OOD (SFO), models are trained on a single frequency and evaluated on the other frequencies data. For geometric OOD (fsplit), models are trained on several kind of defect types and tested on the other unseen type. These splits assess generalization to unseen excitation conditions and defect geometries beyond random train–test shuffling.

We compare a range of baseline and state-of-the-art neural operator models. These include Fourier-based models such as FNO (Li et al., 2021), FFNO (Tran et al., 2023), FCNO (Li et al., 2024), and GeoFNO (Li et al., 2023b), as well as attention-based models like LNO (Wang & Wang, 2024) and Transolver (Wu et al., 2024). Additionally, we include DeepONet and a simple MLP as baseline models to provide a fair and comprehensive comparison.

All models were trained under the same configurations, please refer to Appendix F for more details.

## 4.3 EXPERIMENTAL RESULTS

We report representative operator-learning performance under both in-distribution and OOD settings using diverse metrics (e.g., MSE, RMSE, SSIM; see Appendix E for details). Figure 5 reports a subset of S2Q results, while Table 3 provides a concise overview of T2Q and S2Q performance across normal and OOD regimes on both irregular and regular grids. More detailed quantitative results and visualizations can be found in Appendix J.

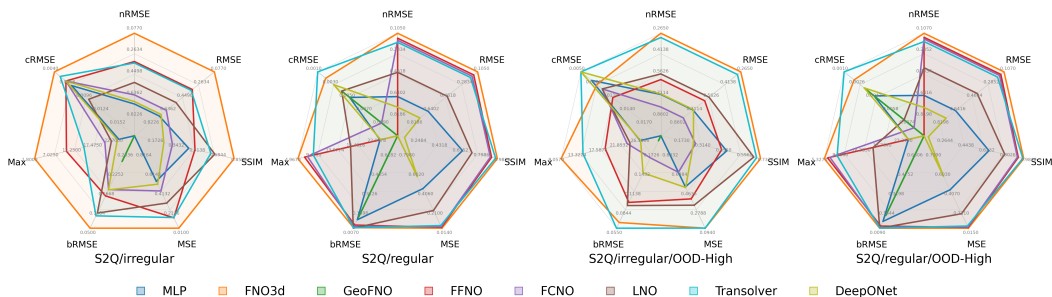

Figure 5: Comparison of the performance of eight models in S2Q task based on seven metrics, with different grids and OOD-High type. The results were derived from Type II multi-layer subset.

Across the four T2Q configurations, FNO provides consistently strong recovery of volumetric sources from full-field temperature data on both irregular and regular meshes. Its global Fourier layers smooth frequency-structured thermodynamics and support stable extrapolation when excitation frequencies shift, so its MSE typically remains among the lowest under Normal, OOD-High, and SFO settings. Although it is not always the top performer, especially in the hardest geometric OOD regime, FNO generally offers a tight upper envelope on core error metrics, indicating that spectral operators capture

Table 3: Concise summary of T2Q and S2Q results under normal training and three OOD regimes: OOD-High (high-frequency hold-out), SFO (9 kHz single-frequency training), and fsplit (geometric OOD, testing on unseen Type II multi), evaluated on irregular and regular grids. Except for fsplit, all results are obtained through Type II multi-layer subset training.

| Model | Irregular Data | | | | | | | | Regular Data | | | | | | | |
|---|---|---|---|---|---|---|---|---|---|---|---|---|---|---|---|---|
| | Normal | | OOD-High | | SFO (9kHz) | | fsplit | | Normal | | OOD-High | | SFO (9kHz) | | fsplit | |
| | T2Q | S2Q | T2Q | S2Q | T2Q | S2Q | T2Q | S2Q | T2Q | S2Q | T2Q | S2Q | T2Q | S2Q | T2Q | S2Q |
| MLP | 0.387 | 0.720 | 0.385 | 0.709 | 0.395 | 0.692 | 0.434 | 0.775 | 0.096 | 0.656 | 0.096 | 0.647 | 0.137 | 0.694 | 0.130 | 0.682 |
| FNO | 0.190 | 0.077 | 0.214 | 0.265 | 0.235 | 0.472 | 0.622 | 0.894 | 0.046 | 0.105 | 0.047 | 0.107 | 0.166 | 0.416 | 0.272 | 0.385 |
| GeoFNO | 0.418 | 1.009 | 0.412 | 1.009 | 0.860 | 1.100 | 0.455 | 1.006 | 0.112 | 0.997 | 0.127 | 0.998 | 1.628 | 2.363 | 0.130 | 1.001 |
| FFNO | 0.287 | 0.334 | 0.302 | 0.601 | 0.314 | 0.752 | 0.557 | 1.006 | 0.052 | 0.148 | 0.052 | 0.142 | 0.125 | 0.440 | 0.252 | 0.396 |
| FCNO | 0.314 | 0.637 | 0.316 | 0.800 | 0.343 | 0.917 | 0.511 | 1.004 | 0.056 | 0.166 | 0.055 | 0.157 | 0.150 | 0.397 | 0.255 | 0.400 |
| LNO | 0.457 | 0.524 | 0.456 | 0.560 | 0.599 | 0.599 | 0.568 | 0.603 | 0.444 | 0.445 | 0.413 | 0.407 | 0.683 | 0.601 | 0.482 | 0.537 |
| Transolver | 0.131 | 0.347 | 0.137 | 0.296 | 0.314 | 0.454 | 0.307 | 0.442 | 0.047 | 0.184 | 0.045 | 0.179 | 0.238 | 0.475 | 0.104 | 0.400 |
| DeepONet | 0.339 | 0.698 | 0.338 | 0.705 | 0.354 | 0.716 | 0.412 | 0.772 | 0.084 | 0.748 | 0.084 | 0.752 | 0.111 | 0.726 | 0.126 | 0.746 |

large-scale thermal conduction with limited aliasing while preserving global accuracy more effectively than most alternatives. To simulate real-world conditions, we designed a noise injection experiment, as detailed in Appendix I.

Transolver demonstrates superior performance in S2Q, where only the surface temperature is accessible and internal states remain unobservable. Across regular and irregular layouts it remains competitive, and its advantage becomes most pronounced in the geometric OOD regime, which is closest to real deployment where crack geometries differ from those seen during training: in this setting Transolver attains the lowest S2Q errors among all baselines on both grids, while several alternatives degrade substantially. Its attention-based, data-dependent weights generate adaptive receptive fields that fuse long-range cross-surface coupling effects and emphasize critical boundary features, effectively suppressing pseudo-correlations induced by sparse or heterogeneous sampling. Further analysis of attention can be found in Appendix H.

It should be noted that, compared to T2Q with full-field observations, S2Q only accesses top surface temperatures and cannot directly observe the interior, substantially reducing identifiability and exacerbating ill posedness. Empirically, almost all models degrade in S2Q, with the drop most pronounced in irregular meshes and under OOD excitations. It proves that limited observations and sparse/heterogeneous sampling further weaken the identifiability of the inverse problem and promote error accumulation across space and frequency. This mirrors the core challenge of NDT: inferring internal heat sources/defects from boundary-only measurements is inherently information-limited.

## 5 CONCLUSION

We have developed and released the first large-scale, multi-frequency coupled 3D Pulsed Eddy Current Thermography benchmark dataset, aimed at advancing the application of machine learning methods in internal crack detection for NDT. This dataset is derived from high fidelity electromagnetic thermal multiphysics simulation data and calibrated with real infrared thermal imaging experimental data, covering various defect types and frequency response scenarios. It supports a range of forward and inverse modeling tasks. Benchmark results show that neural operator models can effectively learn the dynamic process of heat diffusion and reconstruct the primary internal heat source distribution from surface temperature sequences. However, challenges remain in accurately reconstructing fine-grained three-dimensional morphologies of complex defects.

This dataset is the first to encompass the electromagnetic-thermal coupling response of rail materials under different excitation frequencies, providing a standardized testing platform for research into neural network surrogate models, operator learning methods, and defect inversion. Looking ahead, we plan to extend this framework to additional materials, such as composites and pipeline structures, as well as to a broader range of NDT scenarios, further promoting the widespread application of data-driven modeling in industrial inspections. We hope that the Aletheia benchmark will serve as a foundational resource for multi-physics modeling and model generalization research.

ETHICS STATEMENT

This work adheres to the ICLR Code of Ethics. Our study concerns methodological advances for learning neural operators on thermo-electromagnetic simulations and lab measurements of inanimate rail specimens. It involves no human subjects, personally identifiable information, or sensitive user data. Real measurements are acquired from controlled ECPT experiments on metal rails; no living beings are involved, and the procedures present no biological, environmental, or privacy risks. The proposed benchmark aims to improve the reliability of inverse modeling for nondestructive testing. While the dataset and models may inform industrial inspection workflows, they do not directly enable harmful applications. Any future deployment in safety-critical domains must consider regulatory, ethical, and societal constraints beyond the scope of this work (e.g., responsible use, failure modes under distribution shift). We report all methods and results transparently and disclose no conflicts of interest or external sponsorship. All experiments were designed and conducted in accordance with standards of research integrity.

REPRODUCIBILITY STATEMENT

We provide the information necessary to reproduce our results. Dataset construction details (simulation pipeline, experimental setup, sampling strategies, frequencies, and annotations) are described in Section 3 and Appendices A to C. Task definitions, data splits, data distribution types, and evaluation metrics are specified in Section 4 and Appendix E. For each model, we list architectures, hyperparameters, training schedules, and preprocessing/normalization in Section 4.2, with additional tables in Appendix F. We report the exact point counts and time steps, and use standardized evaluation scripts. The dataset and anonymized code will be made publicly available together with scripts to reproduce all tables and figures.

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

## A    SIMULATION EXPERIMENT DETAILS

In COMSOL Multiphysics, the simulation domain consisted of a steel rail geometry with dimensions set to $100 \times 70 \times 45$ which is embedded artificial defects of configurable size, orientation, and depth, surrounded by air domain; The rail material used in the physical experiments was U71Mn steel, and the material parameters used in the simulation were modified from U71Mn properties based on calibration against experimental surface temperature measurements. The constant material properties for steel, air, and excitation coil were assigned as listed in Table 4. All external surfaces of the rail were thermally insulated except for the top surface, which had a convective cooling boundary condition to ambient air, with a heat transfer coefficient set to 5 W/(m$^2$·K); the excitation coil was driven via an applied current boundary condition to simulate inductive heating. The initial temperature of the entire model was uniformly set to 20 °C. A free tetrahedral mesh was employed, featuring fine elements around defect regions and within the electromagnetic skin depth, while coarser elements were used elsewhere; an adaptive refinement scheme further adjusted the mesh based on local conductivity gradients. The simulation was run using a time-dependent solver with a backward differentiation formula (BDF) scheme, covering a time span from 0 to 0.6 s with a fixed time step of 0.05 s.

Table 4: Material property parameters. In the ECPT experiment, due to the extremely short heating process and relatively small temperature rise changes, the impact on the various properties of the material can be ignored. Therefore, the material's property parameters are set to a fixed constant. $\sim$ represents the physical quantity of the material that is not involved in actual calculations.

| Material | Conductivity (S/m) | Relative permeability | Relative dielectric constant | Thermal conductivity (W/m·K) | Density (kg/m$^3$) | Constant pressure heat capacity (J/kg·K) |
|---|---|---|---|---|---|---|
| Steel | $1.3 \times 10^7$ | 200 | 2 | 48 | 8000 | 450 |
| Air | 0 | 1 | 1 | 0.0257 | 1.205 | 1005 |
| Copper | $5.998 \times 10^7$ | 1 | 1 | $\sim$ | $\sim$ | $\sim$ |

## B    DATA GENERATION AND DETAILS

The diversity of defect samples in our dataset is achieved through systematic variations in defect orientation, depth, and length. For single-layer defects, the angle between the defect extension direction and the upper surface differs from sample to sample. In double-layer and multi-layer defects, not only do the individual cracks vary in their inclination to the surface, but the angles between multiple cracks also change. These controlled variations result in a wide range of geometric configurations. Detailed statistical parameters of defect angles are presented in Table 5.

Table 5: Angle parameters for different types of defects. Among them, angle 1 represents the angle formed between the defect direction and the upper surface, while angles 2 and 3 represent the angles between multiple cracks. All units are based on angle system.

| Defect type | Range of different angles | | |
|---|---|---|---|
| | angle I | angle II | angle III |
| Single layer defect | 3~90 | - | - |
| Type I double-layer defect | 24.47~75.47 | 5.72~81.27 | - |
| Type II double-layer defect | 3~71 | - | - |
| Type III double-layer defect | 10.06~80.06 | 0.83~24.03 | - |
| Type I multi-layer defect | 0~89 | - | - |
| Type II multi-layer defect | 30~90 | 12.09~31.1 | 12.12~31.12 |

The dataset contains a total of 4,782 samples, with an overall size of approximately 1.89 terabytes. These samples are organized into six main folders, each corresponding to a specific defect type and named accordingly. Each sample folder is labeled by its defect type followed by one or more

angle values that indicate the orientation of the defect. For example, names may include a single angle for single-layer defects or two to three angles for more complex configurations. Within each sample folder, there are three subfolders storing different types of data. The first subfolder contains surface temperature data sampled on a structured grid. The second provides full three-dimensional temperature and heat field data, also on a structured grid with a uniform spacing of one unit. The third subfolder includes similar 3D data, but sampled on an unstructured grid. The unstructured sampling strategy, which distributes 50,000 points per sample, is detailed in the main text. Table 6 summarizes the number of sampling points for each data type.

Table 6: Sampling point statistics for different types of data. Here, $T_{surf}$ denotes surface temperature measurements, $\{q(\mathbf{x}, t), u(\mathbf{x}, t)\}_{structured}$ represents paired heat source and temperature field data sampled on regular grids, and $\{q(\mathbf{x}, t), u(\mathbf{x}, t)\}_{unstructured}$ denotes the same sampled irregularly.

| Data type | x | y | z | Sampling points |
|---|---|---|---|---|
| $T_{surf}$ | 150 | 105 | - | 10750 |
| $\{q, u\}_{structured}$ | 50 | 30 | 30 | 45000 |
| $\{q, u\}_{unstructured}$ | - | - | - | 50000 |

Each sample is simulated under ten excitation frequencies ranging from 1 to 100 kHz, with each frequency corresponding to one VTU format file. Both surface and volumetric temperature data are time-resolved from 0 to 0.6 seconds, with a time step of 0.05 seconds, resulting in 13 time-series data stored in each VTU file. This structure ensures that the dataset supports diverse tasks involving spatial, temporal, and multi-frequency analysis.

## C  DEFECT DETAILS

Our dataset encompasses six distinct internal crack defect types, spanning single-layer, double-layer, and multi-layer configurations and including both open and closed crack cases. Six types of defects are shown in Figure 6 and the detailed parameters of each defect are shown in Table 7.

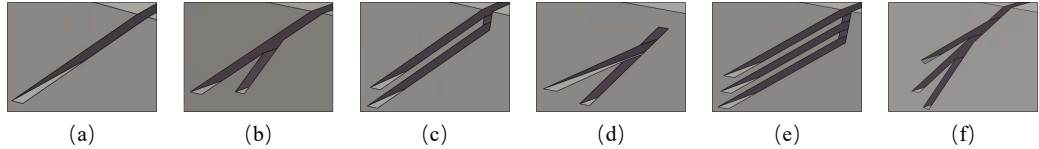

|     (a)     |     (b)     |     (c)     |     (d)     |     (e)     |     (f)     |

Figure 6: 6 types of defects: (a)Single layer defect (b)Type I double-layer defect (c)Type II double-layer defect (d)Type III double-layer defect (e)Type I multi-layer defect (f)Type II multi-layer defect

Table 7: Detailed parameters of 6 types of defects. The units of depth, width, and length are all in millimeters. In double-layer and multi-layer defects, the length of the defect refers to the length of the longest crack.

| Defect type | Sample size | Depth | Width | Length | Open/Closed |
|---|---|---|---|---|---|
| Single layer defect | 88 | 0.2~3.82 | 0.1~0.2 | 3.82 | Open |
| Type I double-layer defect | 1050 | 1.17~4.1 | 0.2 | 4.53 | Open |
| Type II double-layer defect | 69 | 0.24~4.02 | 0.1~0.2 | 4.22 | Open |
| Type III double-layer defect | 2285 | 1.40~3.75 | 0.2 | 3.29 | Closed |
| Type I multi-layer defect | 90 | 0.81~2.41 | 0.2 | 2.4 | Closed |
| Type II multi-layer defect | 1200 | 1.89~3.12 | 0.15 | 3.12 | Open |

## D   DEFECT RECONSTRUCTION FROM HEAT FIELD

We conducted a simple test using a basic 3D CNN to reconstruct the heat source field $Q(x, y, z)$. The results demonstrate that $Q$ can effectively capture defect morphology: high-Q-value dense zones align with defect surfaces, Q-value decay indicates defect propagation direction, and low-Q-value regions correspond to internal cavities. Leveraging a 3D CNN with a large receptive field, we extract global spatial features from the reconstructed Q-field and use a regression network to predict key defect parameters such as angle, length, and depth. Experimental results (Table 8) show reliable predictions for these parameters. The width parameter shows low variance in the dataset, resulting in negligible predictive signal rather than a model limitation. Therefore, as long as our neural operator can reconstruct $Q$ accurately, it can serve as a reliable basis for predicting multidimensional defect characteristics.

Table 8: Regression evaluation indicators for crack parameters

| Attributes | Model | MSE | RMSE | MAE |
|---|---|---|---|---|
| Crack angle | MLP | 0.23 | 0.48 | 0.38 |
| | FNO | 0.17 | 0.41 | 0.35 |
| | Transolver | 0.20 | 0.44 | 0.30 |
| Crack length | MLP | 0.0020 | 0.044 | 0.034 |
| | FNO | 0.0012 | 0.035 | 0.028 |
| | Transolver | 0.0011 | 0.033 | 0.020 |
| Crack depth | MLP | 1.24 | 1.11 | 0.91 |
| | FNO | 0.85 | 0.92 | 0.80 |
| | Transolver | 1.02 | 1.01 | 0.70 |
| Crack width | MLP | $1.00 \times 10^{-14}$ | $1.00 \times 10^{-7}$ | $7.86 \times 10^{-8}$ |
| | FNO | $1.26 \times 10^{-14}$ | $1.12 \times 10^{-7}$ | $1.10 \times 10^{-7}$ |
| | Transolver | $1.43 \times 10^{-14}$ | $1.19 \times 10^{-7}$ | $1.16 \times 10^{-7}$ |

## E   METRICS

Standard methods for calculating the root mean square error (RMSE) of test data fail to capture important optimization criteria in scientific machine learning. It is not enough to fit (usually sparse) data well if the physical laws of the underlying problem are seriously violated. Therefore, they must be evaluated using appropriate metrics. Furthermore, a single evaluation metric is not sufficient to compare differences in the ability of different methods to infer unseen time steps and parameters, which are important but not yet fully explored evaluation criteria for machine learning alternative models. We used the PDEBench evaluation metrics and the SSIM evaluation metrics, as shown in Table 9

The normalized RMSE is a variant of the RMSE to provide scale-independent information defined as:

$$\text{nRMSE} \equiv \frac{\|u_{\text{pred}} - u_{\text{true}}\|_2}{\|u_{\text{true}}\|_2}, \tag{3}$$

where $\|u\|_2$ is the $L_2$-norm of a (vector-valued) variable $u$, and $u_{\text{true}}, u_{\text{pred}}$ are true and predicted values, respectively. The maximum error measures the model's worst prediction, which quantifies both local performance and models' stability of their prediction.

cRMSE is defined as

$$\text{cRMSE} \equiv \frac{\|\sum u_{\text{pred}} - \sum u_{\text{true}}\|_2}{N}, \tag{4}$$

which measures the deviation of the prediction from some physically conserved value.

bRMSE measures the error at the boundary, indicating if the model understands the boundary condition properly.

Table 9: Evaluation indicators provided by PDEBench

| Scope | Acronym | Metric |
|---|---|---|
| Data view | RMSE | root-mean-squared-error |
| | nRMSE | normalized RMSE (ensuring scale independence) |
| | max error | maximum error (local worst case; also proxy for stability of time-stepping) |
| Physics view | cRMSE | RMSE of conserved value (deviation from conserved physical quantity) |
| | bRMSE | RMSE on boundary (whether boundary condition can be learned) |
| | fRMSE low | RMSE in Fourier space, low frequency regime (wavelength dependence) |
| | fRMSE mid | RMSE in Fourier space, medium frequency regime |
| | fRMSE high | RMSE in Fourier space, high frequency regime |

Finally, fRMSE measures the error in low/middle/high-frequency ranges defined as:

$$
\sqrt{\frac{\sum_{k_{\min}}^{k_{\max}} |\mathcal{F}(u_{\text{pred}}) - \mathcal{F}(u_{\text{true}})|^2}{k_{\max} - k_{\min} + 1}}, \tag{5}
$$

where $\mathcal{F}$ is a discrete Fourier transformation, and $k_{\min}, k_{\max}$ are the minimum and maximum indices in Fourier coordinates.

In PDEBench paper, the low/middle/high-frequency regions are defined as:

- Low: $k_{\min} = 0$, $k_{\max} = 4$
- Middle: $k_{\min} = 5$, $k_{\max} = 12$
- High: $k_{\min} = 13$, $k_{\max} = \infty$

This allows a quantitative discussion of the model performance's dependence on the wavelength. In the multidimensional cases, the $|\mathcal{F}(u_{\text{pred}} - u_{\text{true}})(k)|^2$ in the angular coordinate direction is first integrated and summed along the $k$ coordinate.

## F EXPERIMENTAL SETUP DETAILS

To ensure a fair comparison, this section details all hyperparameters and training configurations employed across the models. For the FNO family of models, given their similar parameter sets, these hyperparameters are presented in Table 10.

Table 10: Shared architectural settings for the FNO family baselines. All models use the same spectral bandwidth and channel widths to ensure fairness. $(m_x, m_y, m_z)$ denotes spectral modes, $Width$ denotes the linear transformation applied on the spatial domain, $Grid/scale$ denotes the resolution of the gird.

| Model | $(m_x, m_y, m_z)$ | $Width$ | $Grid/scale$ | Model Variant |
|---|---|---|---|---|
| FNO | $(12, 12, 8)$ | 32 | $(20, 20, 20)$ | Original Fourier Neural Operator |
| GeoFNO | $(12, 12, 8)$ | 32 | $(20, 20, 20)$ | Geometry-Aware FNO |
| FFNO | $(12, 12, 8)$ | 32 | $(20, 20, 20)$ | Factorized FNO |
| FCNO | $(12, 12, 8)$ | 32 | $(20, 20, 20)$ | Factorized Cosine Neural Operator |

**Transsolver.** We instantiate Transsolver with hidden size $n_{hidden} = 256$, depth $n_{layers} = 8$, and spatial dimension $space_{dim} = 3$. Multi-head attention uses $n_{head} = 8$ with an MLP expansion ratio of 2. We further use $slice_{num} = 32$ and disable positional unification. The same configuration is used for all tasks.

**LNO.** The LNO is consisting of $n_{block} = 4$ operator blocks with spectral resolution $n_{mode} = 256$ and hidden width $n_{dim} = 128$. Attention uses $n_{head} = 8$ and $n_{layer} = 2$ transformer layers with `GELU` activations and the vanilla attention kernel; temporal modeling is disabled.

**MLP.** We employ a point-wise multilayer perceptron with hidden width 32, and 4 layers in total, serving as a lightweight regression baseline.

**DeepONet.** The DeepONet baseline adopts a branch–trunk decomposition with width 32 in both sub-networks. The branch network has depth 2; the trunk network has depth 3, receiving the functional input of dimension 13 and producing a scalar output of dimension 1. All other settings follow the original implementation.

All models were trained under the same configuration: batch size of 4, learning rate of 0.0001, and 200 training epochs. Training employed the Adam optimizer alongside the OneCycleLR learning rate scheduler. All experiments were conducted on a single NVIDIA RTX 4090 GPU with 24GB of memory. The models were trained using mean squared error (MSE) loss, and evaluated using SSIM along with several PDEBench metrics. Regarding the part of evaluation indicators, lease refer to Appendix E for detailed definitions.

Detailed configurations reference Table 11. Apart from DeepONet, each model permits customisation of input and output channels. Due to DeepONet's constraint that its output channel must be limited to 1, DeepONet participated solely in the T2Q and S2Q tasks.

Table 11: Training configuration across tasks. Points denote the number of samples. *In → Out ch.* denotes input and output channels, *Sample Points* denotes the number of downsampling points for each data, *Epochs* denotes total training epochs, *LR* denotes initial learning rate, *Sample Data* denotes the number of simulated data used for training, *Optimizer* denotes the optimiser employed for training, *Batch* denotes the batch size used for training, *Training Times* denotes the number of repeated training sessions for each task

| Task | In → Out ch. | Sample Points | Epochs | LR | Sample Data | Optimizer | Batch | Training Times |
|------|------------|---------------|--------|-----|-------------|-----------|-------|----------------|
| T2Q | $13 \to 1$ | 8 000 (vol.) | | | | | | |
| S2Q | $13 \to 1$ | 8 000 (vol.) + 8 000 (surf.) | 200 | $1 \times 10^{-4}$ | 600 | Adam | 4 | 3 |
| T2T | $11 \to 2$ | 8 000 (vol.) | | | | | | |
| Q2T | $1 \to 13$ | 8 000 (vol.) | | | | | | |

## G  LLM USE DISCLOSURE

We used large language models (LLMs) only for paper grammar and wording edits, minor LaTeX formatting for tables and figures, lightweight source code checks and plotting assistance. LLMs were *not* used to generate scientific claims, design or run experiments, analyze results, create data or alter data, nor to draft substantive technical content. All scientific content, analyses, and conclusions were authored and verified by the authors, no confidential submission materials were provided to third-party LLM services. We take full responsibility for the submission.

## H  PHYSICS ATTENTION ANALYSIS

The S2Q task is governed by a smooth diffusive background with sharply localized anomalies at material interfaces and crack tips: the volumetric heat source is piecewise smooth in the interior but exhibits strong gradients near the upper surface and along the defect boundaries. Successful inversion therefore requires the model to selectively attend to these "physically active" regions, rather than treating all interior voxels uniformly. Transsolver is explicitly designed for this purpose through Physics-Attention, which first projects each mesh point feature into a low-entropy slice weight over learnable slices and then aggregates points within each slice into physics-aware tokens, followed by attention among tokens. This bottom-up decomposition pushes points with similar geometry

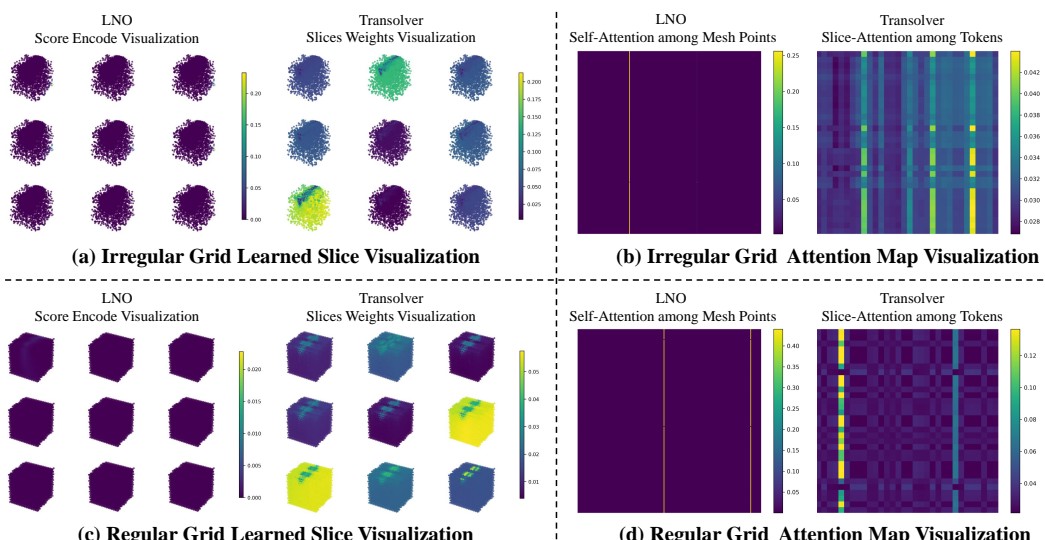

**Figure 7:** Attention visualization in the S2Q task for Type I double-layer defect with different grid: (a) slice weights in the last layer of Transolver and LNO for irregular grid, (b) attention maps of the last layer in Transolver and LNO for irregular grid, (c) slice weights in the last layer of Transolver and LNO for regular grid, (d) attention maps of the last layer in Transolver and LNO for regular grid.

and temperature patterns into the same slice, and the Softmax over the slice dimension sharpens the assignment so that slices specialize to distinct physical states.

To verify that this mechanism also emerges on Aletheia, we visualize in Figure 7(a, c), the learned slice weights in the last Transolver layer and compare them with the encoder scores of LNO on both irregular and regular grids for Type I double-layer defect. On the irregular grids, Transolver's slices form several compact 3D clusters that hug the vicinity of the defect source and the crack front, while assigning low weights to homogeneous interior regions. In contrast, the LNO score-encode fields are much more diffuse: the high-response regions spread over large portions of the volume and show weak alignment with the true source interfaces. On regular grids, this gap is also quite obvious. Transolver learns slice volumes that sharply align with the two latent source layers and their vertical transition zones, producing box-like clusters concentrated near the interfaces and near the top surface where the measured temperature is most informative for inversion. LNO again yields almost block-uniform scores inside each cube, indicating that its latent representation does not clearly separate boundary and background states.

We further compare the attention maps in Figure 7(b, d). For Transolver, attention is computed among physics-aware tokens, each summarizing one slice. The resulting slice–attention matrices exhibit structured, high-contrast patterns: a small subset of "boundary" slices receive high attention, and form blocks of strong interactions with their neighboring interior slices. This reflects a physically meaningful message-passing pattern: information is routed from the surface and boundary slices into nearby interior slices where the heat source lies. By contrast, LNO performs self-attention directly over all mesh points. The corresponding 2D attention maps are markedly closer to a degenerate pattern: most rows are near-uniform with a few narrow bright columns, and there is no clear block structure aligned with the defect geometry. This is consistent with the observation in the Transolver paper that attention over raw mesh points tends to be much flatter and closer to a uniform distribution than Physics-Attention over tokens.

Overall, these visualizations support the hypothesis that the adaptive, slice-based receptive field is well suited to S2Q. The learned slices provide an intermediate representation that (i) automatically segments the domain into physically coherent regions, (ii) concentrates model capacity on the thin boundary layers that encode most of the information about subsurface heat sources, and (iii) enables sharp, interpretable token-level attention patterns that capture long-range coupling between surface measurements and interior sources. In contrast, LNO's point-wise attention struggles to disentangle boundary anomalies from smooth background diffusion, leading to more diffuse score fields and

nearly uniform attention. These qualitative differences in physical attention align with the quantitative results in Tables 12 to 22, where Transolver achieves substantially lower S2Q error than LNO on both regular and irregular grids.

# I    NOISE ANALYSIS

Table 12: S2Q Surface-to-Source Reconstruction errors (MSE / RMSE) for geometric OOD (fsplit, train on defect types 1–5 and test on type 6) under different noise levels (Normal, Low-Noise, Mid-Noise, High-Noise), evaluated on irregular and regular grids.

| Model | Irregular Data | | | | | | | | Regular Data | | | | | | | |
|---|---|---|---|---|---|---|---|---|---|---|---|---|---|---|---|---|
| | Normal | | Low-Noise | | Mid-Noise | | High-Noise | | Normal | | Low-Noise | | Mid-Noise | | High-Noise | |
| | MSE | RMSE | MSE | RMSE | MSE | RMSE | MSE | RMSE | MSE | RMSE | MSE | RMSE | MSE | RMSE | MSE | RMSE |
| MLP | 0.605 | 0.775 | 0.652 | 0.805 | 0.696 | 0.832 | 1.019 | 1.007 | 0.469 | 0.682 | 0.465 | 0.672 | 0.470 | 0.681 | 0.547 | 0.737 |
| FNO | 0.804 | 0.894 | 0.812 | 0.887 | 0.821 | 0.903 | 0.842 | 0.915 | 0.184 | 0.385 | 0.186 | 0.389 | 0.188 | 0.392 | 0.205 | 0.428 |
| GeoFNO | 1.011 | 1.006 | 1.015 | 1.008 | 1.032 | 1.016 | 1.058 | 1.029 | 1.003 | 1.001 | 1.006 | 1.003 | 1.010 | 1.005 | 1.024 | 1.012 |
| FFNO | 1.017 | 1.006 | 1.018 | 1.007 | 1.034 | 1.015 | 1.112 | 1.051 | 0.186 | 0.396 | 0.192 | 0.403 | 0.206 | 0.422 | 0.266 | 0.498 |
| FCNO | 1.008 | 1.004 | 1.013 | 1.006 | 1.013 | 1.009 | 1.018 | 1.009 | 0.186 | 0.400 | 0.193 | 0.410 | 0.223 | 0.446 | 0.342 | 0.576 |
| LNO | 0.371 | 0.603 | 0.390 | 0.615 | 0.434 | 0.628 | 0.579 | 0.652 | 0.304 | 0.537 | 0.310 | 0.542 | 0.341 | 0.567 | 0.410 | 0.623 |
| Transolver | 0.218 | 0.442 | 0.253 | 0.478 | 0.281 | 0.511 | 0.373 | 0.597 | 0.197 | 0.400 | 0.201 | 0.403 | 0.212 | 0.418 | 0.256 | 0.485 |
| DeepONet | 0.600 | 0.772 | 0.611 | 0.775 | 0.610 | 0.778 | 0.623 | 0.737 | 0.560 | 0.746 | 0.560 | 0.738 | 0.561 | 0.745 | 0.572 | 0.763 |

As shown in Table 12, we further investigate the robustness of the S2Q task under different noise levels, in a setting that more closely reflects real industrial NDT conditions. We conduct a noise-robustness study in the geometric OOD regime: all models are trained on defect types 1–5 and evaluated on the unseen Type II multi-layer defect under multiple noise settings on both irregular and regular grids. Overall, injecting noise into the surface temperature measurements leads to a gradual degradation of performance rather than catastrophic failure. On the irregular grid, most baselines exhibit noticeable increases in mean squared error (MSE) from the Normal to the High-Noise regime; for example, the purely data-driven MLP rises from 0.605 to 1.019 ($\approx +68\%$), and LNO increases from 0.371 to 0.579 ($\approx +56\%$). On the regular grid, where the forward diffusion operator is better resolved, the same models show substantially smaller relative changes; MLP grows only from 0.469 to 0.547 ($\approx +17\%$), and LNO from 0.310 to 0.410 ($\approx +32\%$). Frequency-domain neural-operator baselines exhibit nearly flat error curves and modest error growth on both grids, indicating comparatively strong robustness.

FNO and Transolver display two distinct robustness patterns, consistent with their architectural biases. On the irregular grid, FNO's MSE increases only slightly from 0.804 (Normal) to 0.842 (High-Noise), a relative change of about $5\%$; Transolver shows mild degradation under Low-Noise and Mid-Noise conditions, and although its performance drop becomes larger in the High-Noise setting, it still maintains a clear advantage in absolute accuracy. This suggests that its physics-aware tokenization and attention mechanism can effectively filter high-frequency noise while preserving crack-induced boundary features. In the most challenging geometric OOD scenario, degradation across neural operators is therefore expected but far from catastrophic. On the regular grid, FNO achieves the lowest absolute errors overall, with MSE increasing only from 0.184 to 0.205 ($\approx +11\%$), which aligns with its spectral bias toward smooth diffusion on structured meshes. Transolver's absolute accuracy on the regular grid is slightly worse than that of FNO but remains competitive and significantly better than simpler baselines such as MLP and DeepONet.

Taken together, in a configuration that more faithfully approximates practical ECPT measurement conditions, all neural operators experience some performance loss under noise, but the degradation follows a structured and interpretable pattern. Regular grids are slightly more favorable than irregular point clouds, indicating that geometric and sampling uncertainty compounds the impact of noise; FNO exhibits particularly strong robustness and accuracy on structured meshes, whereas Transolver maintains strong robustness and low error under complex geometric OOD and varying noise levels. Benchmarking with ALETHEIA and analyzing how different neural-operator architectures trade off accuracy and robustness under realistic noise provides an evaluation setting that is much closer to

engineering practice, and can guide the design of future neural operators with explicit uncertainty awareness and noise-suppression capabilities.

## J MORE EXPERIMENTAL RESULTS

Here we report detailed quantitative comparisons of neural operator baselines on Forward Modeling (Q2T), Inverse Source Reconstruction (T2Q), Temporal Evolution Prediction (T2T), and Surface-to-Source Reconstruction (S2Q), spanning regular and irregular grids and both full-frequency and OOD settings. Experimental results are collected here, which cover a variety of models such as MLP, Transolver, FNO, and so on. For normal and frequency-OOD experiments we use MSE and RMSE as the primary metrics due to space constraints, while for the more challenging geometric OOD (fsplit) setting we additionally report the full set of seven error measures to more comprehensively characterize model behavior. Among these metrics, SSIM is the only one where higher values indicate better performance, whereas all other metrics are preferred to be smaller. Entries denoted by "/" correspond to cases where repeated trainings exhibited highly unstable behavior or consistently failed to produce meaningful results.

Table 13: Full metric breakdown of S2Q Surface-to-Source Reconstruction under geometric OOD (fsplit). Models are trained on 5 defect types and tested on the held-out Type II multi-layer defect, evaluated on irregular and regular grids. Reported metrics include MSE, SSIM, RMSE, normalized RMSE (nRMSE), conserved value RMSE (cRMSE), maximum error (Max), and boundary RMSE (bRMSE).

| Model | Irregular Data | | | | | | | Regular Data | | | | | | |
|---|---|---|---|---|---|---|---|---|---|---|---|---|---|---|
| | MSE | SSIM | RMSE | nRMSE | cRMSE | Max | bRMSE | MSE | SSIM | RMSE | nRMSE | cRMSE | Max | bRMSE |
| MLP | 0.605 | 0.414 | 0.775 | 0.775 | 0.015 | 26.183 | 0.204 | 0.469 | 0.652 | 0.682 | 0.682 | 0.007 | 27.144 | 0.050 |
| FNO | 0.804 | 0.407 | 0.894 | 0.894 | 0.040 | 27.408 | 0.129 | 0.184 | 0.953 | 0.385 | 0.385 | 0.003 | 23.723 | 0.011 |
| GeoFNO | 1.011 | -0.028 | 1.006 | 1.006 | 0.022 | 29.512 | 0.214 | 1.003 | 0.053 | 1.001 | 1.001 | 0.004 | 29.385 | 0.098 |
| FFNO | 1.017 | 0.346 | 1.006 | 1.006 | 0.028 | 30.898 | 0.151 | 0.186 | 0.881 | 0.396 | 0.396 | 0.010 | 23.176 | 0.033 |
| FCNO | 1.008 | 0.234 | 1.004 | 1.004 | 0.029 | 28.865 | 0.130 | 0.186 | 0.866 | 0.400 | 0.400 | 0.011 | 23.324 | 0.015 |
| LNO | 0.371 | 0.644 | 0.603 | 0.603 | 0.015 | 24.012 | 0.100 | 0.304 | 0.736 | 0.537 | 0.537 | 0.006 | 26.019 | 0.007 |
| Transolver | 0.218 | 0.705 | 0.442 | 0.442 | 0.006 | 20.878 | 0.053 | 0.197 | 0.944 | 0.400 | 0.400 | 0.002 | 23.240 | 0.013 |
| DeepONet | 0.600 | 0.228 | 0.772 | 0.772 | 0.013 | 26.312 | 0.210 | 0.560 | 0.144 | 0.746 | 0.746 | 0.002 | 26.578 | 0.712 |

Table 14: S2Q reconstruction performance (MSE / RMSE) for the single layer defect configuration under normal training and three frequency-band OOD regimes (High-OOD, Mid-OOD, Low-OOD), on irregular and regular grids.

| Model | Irregular Data | | | | | | | | Regular Data | | | | | | | |
|---|---|---|---|---|---|---|---|---|---|---|---|---|---|---|---|---|
| | Normal | | High-OOD | | Mid-OOD | | Low-OOD | | Normal | | High-OOD | | Mid-OOD | | Low-OOD | |
| | MSE | RMSE | MSE | RMSE | MSE | RMSE | MSE | RMSE | MSE | RMSE | MSE | RMSE | MSE | RMSE | MSE | RMSE |
| MLP | 0.446 | 0.664 | 0.428 | 0.651 | 0.440 | 0.659 | 0.426 | 0.650 | 0.338 | 0.578 | 0.331 | 0.572 | 0.333 | 0.574 | 0.337 | 0.577 |
| FNO | 0.009 | 0.088 | 0.008 | 0.087 | 0.008 | 0.086 | 0.009 | 0.086 | 0.002 | 0.040 | 0.002 | 0.040 | 0.002 | 0.041 | 0.002 | 0.039 |
| GeoFNO | 0.998 | 0.999 | 1.014 | 1.007 | 1.018 | 1.009 | 1.008 | 1.004 | 1.015 | 1.007 | 1.000 | 1.000 | 0.998 | 0.999 | 0.997 | 0.999 |
| FFNO | 0.052 | 0.218 | 0.048 | 0.213 | 0.053 | 0.221 | 0.046 | 0.209 | 0.015 | 0.115 | 0.014 | 0.112 | 0.015 | 0.116 | 0.015 | 0.118 |
| FCNO | 0.077 | 0.266 | 0.071 | 0.260 | 0.077 | 0.267 | 0.068 | 0.253 | 0.022 | 0.137 | 0.019 | 0.131 | 0.021 | 0.134 | 0.021 | 0.136 |
| LNO | 0.249 | 0.494 | 0.244 | 0.491 | 0.992 | 0.996 | 0.259 | 0.506 | 0.097 | 0.308 | 0.093 | 0.301 | 0.095 | 0.304 | 0.096 | 0.307 |
| Transolver | 0.052 | 0.222 | 0.050 | 0.219 | 0.054 | 0.225 | 0.045 | 0.208 | 0.035 | 0.176 | 0.030 | 0.157 | 0.032 | 0.168 | 0.024 | 0.148 |
| DeepONet | 0.520 | 0.713 | 0.499 | 0.699 | 0.515 | 0.710 | 0.485 | 0.690 | 0.521 | 0.720 | 0.504 | 0.709 | 0.508 | 0.712 | 0.511 | 0.714 |

Table 15: S2Q reconstruction performance (MSE / RMSE) for the Type I double-layer defect under normal training and three frequency-band OOD regimes (High-OOD, Mid-OOD, Low-OOD), on irregular and regular grids.

| Model | Irregular Data | | | | | | | | Regular Data | | | | | | | |
|---|---|---|---|---|---|---|---|---|---|---|---|---|---|---|---|---|
| | Normal | | High-OOD | | Mid-OOD | | Low-OOD | | Normal | | High-OOD | | Mid-OOD | | Low-OOD | |
| | MSE | RMSE | MSE | RMSE | MSE | RMSE | MSE | RMSE | MSE | RMSE | MSE | RMSE | MSE | RMSE | MSE | RMSE |
| MLP | 0.599 | 0.772 | 0.609 | 0.778 | 0.602 | 0.774 | 0.603 | 0.774 | 0.629 | 0.790 | 0.605 | 0.776 | 0.626 | 0.789 | 0.635 | 0.795 |
| FNO | 0.029 | 0.159 | 0.025 | 0.147 | 0.028 | 0.156 | 0.028 | 0.154 | 0.006 | 0.068 | 0.007 | 0.070 | 0.007 | 0.070 | 0.006 | 0.066 |
| GeoFNO | 1.001 | 1.001 | 1.005 | 1.003 | 1.007 | 1.003 | 1.047 | 1.023 | 0.998 | 0.999 | 0.999 | 1.000 | 0.993 | 0.996 | 0.999 | 1.000 |
| FFNO | 0.161 | 0.397 | 0.162 | 0.400 | 0.162 | 0.398 | 0.157 | 0.393 | 0.055 | 0.229 | 0.054 | 0.226 | 0.057 | 0.232 | 0.051 | 0.221 |
| FCNO | 0.235 | 0.479 | 0.251 | 0.493 | 0.245 | 0.489 | 0.239 | 0.482 | 0.085 | 0.284 | 0.085 | 0.283 | 0.084 | 0.282 | 0.079 | 0.276 |
| LNO | 0.493 | 0.695 | 0.507 | 0.705 | 0.500 | 0.700 | 0.501 | 0.700 | 0.231 | 0.459 | 0.338 | 0.563 | 0.167 | 0.393 | 0.154 | 0.373 |
| Transolver | 0.103 | 0.313 | 0.114 | 0.329 | 0.105 | 0.314 | 0.104 | 0.314 | 0.079 | 0.276 | 0.068 | 0.256 | 0.076 | 0.269 | 0.064 | 0.248 |
| DeepONet | 0.662 | 0.811 | 0.648 | 0.803 | 0.646 | 0.802 | 0.644 | 0.800 | 0.724 | 0.849 | 0.752 | 0.866 | 0.768 | 0.875 | 0.767 | 0.874 |

Table 16: S2Q reconstruction performance (MSE / RMSE) for the Type II double-layer defect under normal training and three frequency-band OOD regimes (High-OOD, Mid-OOD, Low-OOD), on irregular and regular grids.

| Model | Irregular Data | | | | | | | | Regular Data | | | | | | | |
|---|---|---|---|---|---|---|---|---|---|---|---|---|---|---|---|---|
| | Normal | | High-OOD | | Mid-OOD | | Low-OOD | | Normal | | High-OOD | | Mid-OOD | | Low-OOD | |
| | MSE | RMSE | MSE | RMSE | MSE | RMSE | MSE | RMSE | MSE | RMSE | MSE | RMSE | MSE | RMSE | MSE | RMSE |
| MLP | 0.543 | 0.733 | 0.527 | 0.722 | 0.546 | 0.735 | 0.536 | 0.729 | 0.388 | 0.621 | 0.380 | 0.615 | 0.385 | 0.619 | 0.377 | 0.612 |
| FNO | 0.010 | 0.077 | 0.012 | 0.085 | 0.010 | 0.078 | 0.011 | 0.082 | 0.002 | 0.038 | 0.002 | 0.038 | 0.002 | 0.041 | 0.002 | 0.037 |
| GeoFNO | 1.030 | 1.013 | 1.007 | 1.004 | 1.015 | 1.007 | 1.012 | 1.006 | 1.026 | 1.013 | 0.999 | 0.999 | 1.016 | 1.008 | 1.004 | 1.002 |
| FFNO | 0.129 | 0.339 | 0.134 | 0.347 | 0.125 | 0.334 | 0.137 | 0.352 | 0.018 | 0.130 | 0.016 | 0.124 | 0.018 | 0.129 | 0.016 | 0.125 |
| FCNO | 0.421 | 0.635 | 0.433 | 0.648 | 0.407 | 0.623 | 0.438 | 0.650 | 0.029 | 0.166 | 0.031 | 0.170 | 0.034 | 0.176 | 0.028 | 0.163 |
| LNO | 0.368 | 0.597 | 0.323 | 0.556 | 0.362 | 0.588 | 0.336 | 0.571 | 0.186 | 0.427 | 0.172 | 0.407 | 0.181 | 0.419 | 0.189 | 0.429 |
| Transolver | 0.076 | 0.270 | 0.076 | 0.268 | 0.078 | 0.271 | 0.065 | 0.249 | 0.020 | 0.132 | 0.021 | 0.138 | 0.029 | 0.161 | 0.022 | 0.139 |
| DeepONet | 0.542 | 0.732 | 0.528 | 0.723 | 0.556 | 0.741 | 0.542 | 0.732 | 0.519 | 0.718 | 0.516 | 0.716 | 0.524 | 0.722 | 0.522 | 0.721 |

Table 17: S2Q reconstruction performance (MSE / RMSE) for the Type III double-layer defect under normal training and three frequency-band OOD regimes (High-OOD, Mid-OOD, Low-OOD), on irregular and regular grids.

| Model | Irregular Data | | | | | | | | Regular Data | | | | | | | |
|---|---|---|---|---|---|---|---|---|---|---|---|---|---|---|---|---|
| | Normal | | High-OOD | | Mid-OOD | | Low-OOD | | Normal | | High-OOD | | Mid-OOD | | Low-OOD | |
| | MSE | RMSE | MSE | RMSE | MSE | RMSE | MSE | RMSE | MSE | RMSE | MSE | RMSE | MSE | RMSE | MSE | RMSE |
| MLP | 0.561 | 0.744 | 0.548 | 0.736 | 0.519 | 0.717 | 0.502 | 0.706 | 0.473 | 0.682 | 0.447 | 0.664 | 0.433 | 0.654 | 0.439 | 0.658 |
| FNO | 0.016 | 0.098 | 0.010 | 0.084 | 0.232 | 0.417 | 0.242 | 0.437 | 0.005 | 0.062 | 0.063 | 0.233 | 0.069 | 0.247 | 0.062 | 0.234 |
| GeoFNO | 1.013 | 1.006 | 1.015 | 1.007 | 1.007 | 1.003 | 1.009 | 1.004 | 0.991 | 0.995 | 1.005 | 1.003 | 0.994 | 0.997 | 0.999 | 1.000 |
| FFNO | 0.170 | 0.382 | 0.140 | 0.352 | 0.530 | 0.705 | 0.574 | 0.738 | 0.041 | 0.199 | 0.074 | 0.267 | 0.077 | 0.270 | 0.076 | 0.270 |
| FCNO | 0.479 | 0.675 | 0.464 | 0.668 | 0.800 | 0.885 | 0.816 | 0.895 | 0.069 | 0.259 | 0.088 | 0.291 | 0.093 | 0.298 | 0.092 | 0.298 |
| LNO | 0.397 | 0.622 | 0.314 | 0.550 | 0.592 | 0.642 | 0.321 | 0.561 | 0.128 | 0.354 | 0.145 | 0.376 | 0.160 | 0.395 | 0.171 | 0.408 |
| Transolver | 0.055 | 0.230 | 0.051 | 0.223 | 0.132 | 0.356 | 0.136 | 0.363 | 0.047 | 0.214 | 0.103 | 0.310 | 0.101 | 0.309 | 0.093 | 0.297 |
| DeepONet | 0.580 | 0.756 | 0.569 | 0.750 | 0.551 | 0.739 | 0.538 | 0.731 | 0.675 | 0.820 | 0.623 | 0.787 | 0.613 | 0.781 | 0.611 | 0.779 |

Table 18: S2Q reconstruction performance (MSE / RMSE) for the Type I multi-layer defect under normal training and three frequency-band OOD regimes (High-OOD, Mid-OOD, Low-OOD), on irregular and regular grids.

| Model | Irregular Data | | | | | | | | Regular Data | | | | | | | |
|---|---|---|---|---|---|---|---|---|---|---|---|---|---|---|---|---|
| | Normal | | High-OOD | | Mid-OOD | | Low-OOD | | Normal | | High-OOD | | Mid-OOD | | Low-OOD | |
| | MSE | RMSE | MSE | RMSE | MSE | RMSE | MSE | RMSE | MSE | RMSE | MSE | RMSE | MSE | RMSE | MSE | RMSE |
| MLP | 0.550 | 0.737 | 0.539 | 0.730 | 0.541 | 0.731 | 0.541 | 0.731 | 0.438 | 0.657 | 0.427 | 0.649 | 0.409 | 0.636 | 0.422 | 0.645 |
| FNO | 0.018 | 0.094 | 0.015 | 0.089 | 0.017 | 0.095 | 0.012 | 0.086 | 0.001 | 0.030 | 0.002 | 0.034 | 0.003 | 0.042 | 0.002 | 0.036 |
| GeoFNO | 1.023 | 1.010 | 1.066 | 1.021 | 1.023 | 1.011 | 1.029 | 1.014 | 1.004 | 1.002 | 1.011 | 1.006 | 1.018 | 1.009 | 1.007 | 1.004 |
| FFNO | 0.138 | 0.345 | 0.128 | 0.333 | 0.135 | 0.339 | 0.122 | 0.329 | 0.019 | 0.133 | 0.019 | 0.135 | 0.023 | 0.146 | 0.021 | 0.139 |
| FCNO | 0.379 | 0.598 | 0.399 | 0.617 | 0.411 | 0.621 | 0.405 | 0.622 | 0.036 | 0.184 | 0.040 | 0.190 | 0.052 | 0.214 | 0.038 | 0.188 |
| LNO | 0.289 | 0.522 | 0.311 | 0.546 | 0.304 | 0.542 | 0.273 | 0.513 | 0.295 | 0.529 | 0.284 | 0.519 | 0.264 | 0.500 | 0.274 | 0.509 |
| Transolver | 0.190 | 0.421 | 0.095 | 0.301 | 0.071 | 0.259 | 0.077 | 0.273 | 0.011 | 0.099 | 0.011 | 0.097 | 0.018 | 0.119 | 0.010 | 0.090 |
| DeepONet | 0.541 | 0.731 | 0.525 | 0.721 | 0.531 | 0.724 | 0.526 | 0.721 | 0.576 | 0.756 | 0.565 | 0.749 | 0.551 | 0.740 | 0.562 | 0.747 |

Table 19: S2Q reconstruction performance (MSE / RMSE) for the Type II multi-layer defect under normal training and three frequency-band OOD regimes (High-OOD, Mid-OOD, Low-OOD), on irregular and regular grids.

| Model | Irregular Data | | | | | | | | Regular Data | | | | | | | |
|---|---|---|---|---|---|---|---|---|---|---|---|---|---|---|---|---|
| | Normal | | High-OOD | | Mid-OOD | | Low-OOD | | Normal | | High-OOD | | Mid-OOD | | Low-OOD | |
| | MSE | RMSE | MSE | RMSE | MSE | RMSE | MSE | RMSE | MSE | RMSE | MSE | RMSE | MSE | RMSE | MSE | RMSE |
| MLP | 0.522 | 0.720 | 0.507 | 0.709 | 0.520 | 0.718 | 0.499 | 0.704 | 0.434 | 0.656 | 0.422 | 0.647 | 0.438 | 0.658 | 0.431 | 0.653 |
| FNO | 0.010 | 0.077 | 0.094 | 0.265 | 0.111 | 0.280 | 0.102 | 0.277 | 0.014 | 0.105 | 0.015 | 0.107 | 0.017 | 0.114 | 0.018 | 0.114 |
| GeoFNO | 1.018 | 1.009 | 1.018 | 1.009 | 1.027 | 1.014 | 1.005 | 1.002 | 0.994 | 0.997 | 0.995 | 0.998 | 0.996 | 0.998 | 0.998 | 0.999 |
| FFNO | 0.123 | 0.334 | 0.387 | 0.601 | 0.378 | 0.591 | 0.395 | 0.608 | 0.024 | 0.148 | 0.023 | 0.142 | 0.027 | 0.153 | 0.026 | 0.147 |
| FCNO | 0.416 | 0.637 | 0.653 | 0.800 | 0.675 | 0.813 | 0.669 | 0.809 | 0.031 | 0.166 | 0.028 | 0.157 | 0.032 | 0.168 | 0.031 | 0.164 |
| LNO | 0.284 | 0.524 | 0.324 | 0.560 | 0.314 | 0.550 | 0.289 | 0.528 | 0.202 | 0.445 | 0.168 | 0.407 | 0.219 | 0.461 | 0.206 | 0.449 |
| Transolver | 0.126 | 0.347 | 0.095 | 0.296 | 0.225 | 0.458 | 0.079 | 0.273 | 0.042 | 0.184 | 0.038 | 0.179 | 0.046 | 0.193 | 0.047 | 0.195 |
| DeepONet | 0.490 | 0.698 | 0.501 | 0.705 | 0.518 | 0.717 | 0.493 | 0.699 | 0.563 | 0.748 | 0.568 | 0.752 | 0.575 | 0.756 | 0.553 | 0.741 |

Table 20: S2Q reconstruction performance (MSE / RMSE) for the Type I double-layer defect under single-frequency OOD (SFO). Models are trained on all frequencies (Normal) or on a single excitation frequency at 9, 36, or 81 kHz, and evaluated on irregular and regular grids.

| Model | Irregular Data | | | | | | | | Regular Data | | | | | | | |
|---|---|---|---|---|---|---|---|---|---|---|---|---|---|---|---|---|
| | Normal | | SFO-9kHz | | SFO-36kHz | | SFO-81kHz | | Normal | | SFO-9kHz | | SFO-36kHz | | SFO-81kHz | |
| | MSE | RMSE | MSE | RMSE | MSE | RMSE | MSE | RMSE | MSE | RMSE | MSE | RMSE | MSE | RMSE | MSE | RMSE |
| MLP | 0.599 | 0.772 | 0.725 | 0.846 | 0.746 | 0.862 | 0.758 | 0.870 | 0.629 | 0.790 | 0.823 | 0.882 | 0.597 | 0.770 | 0.693 | 0.831 |
| FNO | 0.029 | 0.159 | 0.767 | 0.833 | 0.615 | 0.731 | 0.870 | 0.870 | 0.006 | 0.068 | 0.604 | 0.699 | 0.431 | 0.582 | 0.545 | 0.669 |
| GeoFNO | 1.001 | 1.001 | / | / | / | / | / | / | 0.998 | 0.999 | 1.069 | 1.032 | 1.036 | 1.018 | 1.088 | 1.043 |
| FFNO | 0.161 | 0.397 | 0.553 | 0.728 | 0.519 | 0.709 | 0.695 | 0.814 | 0.055 | 0.229 | 0.502 | 0.653 | 0.394 | 0.598 | 0.471 | 0.641 |
| FCNO | 0.235 | 0.479 | 0.607 | 0.763 | 0.573 | 0.743 | 0.647 | 0.791 | 0.085 | 0.284 | 0.394 | 0.602 | 0.465 | 0.640 | 0.418 | 0.618 |
| LNO | 0.493 | 0.695 | 0.999 | 0.953 | 0.523 | 0.717 | 1.780 | 1.152 | 0.231 | 0.459 | 0.661 | 0.728 | / | / | 1.063 | 1.001 |
| Transolver | 0.103 | 0.313 | 0.585 | 0.748 | 0.565 | 0.740 | 0.573 | 0.746 | 0.079 | 0.276 | 0.355 | 0.563 | 0.419 | 0.601 | 0.393 | 0.607 |
| DeepONet | 0.662 | 0.811 | 0.654 | 0.803 | 0.666 | 0.813 | 0.643 | 0.800 | 0.724 | 0.849 | 0.741 | 0.856 | 0.767 | 0.875 | 0.789 | 0.888 |

Table 21: S2Q reconstruction performance (MSE / RMSE) for the Type III double-layer defect under single-frequency OOD (SFO). Models are trained on all frequencies (Normal) or on a single excitation frequency at 9, 36, or 81 kHz, and evaluated on irregular and regular grids.

| Model | Irregular Data | | | | | | | | Regular Data | | | | | | | |
|---|---|---|---|---|---|---|---|---|---|---|---|---|---|---|---|---|
| | Normal | | SFO-9kHz | | SFO-36kHz | | SFO-81kHz | | Normal | | SFO-9kHz | | SFO-36kHz | | SFO-81kHz | |
| | MSE | RMSE | MSE | RMSE | MSE | RMSE | MSE | RMSE | MSE | RMSE | MSE | RMSE | MSE | RMSE | MSE | RMSE |
| MLP | 0.561 | 0.744 | 0.556 | 0.739 | 0.566 | 0.748 | 0.555 | 0.744 | 0.473 | 0.682 | 0.498 | 0.695 | 0.456 | 0.671 | 0.529 | 0.720 |
| FNO | 0.016 | 0.098 | 0.269 | 0.484 | 0.183 | 0.375 | 0.234 | 0.417 | 0.005 | 0.062 | 0.322 | 0.531 | 0.185 | 0.374 | 0.268 | 0.462 |
| GeoFNO | 1.013 | 1.006 | 1.148 | 1.062 | 1.049 | 1.024 | 1.182 | 1.081 | 0.991 | 0.995 | 1.644 | 1.219 | 1.170 | 1.075 | 1.028 | 1.014 |
| FFNO | 0.170 | 0.382 | 0.540 | 0.724 | 0.450 | 0.651 | 0.511 | 0.696 | 0.041 | 0.199 | 0.233 | 0.463 | 0.153 | 0.370 | 0.200 | 0.425 |
| FCNO | 0.479 | 0.675 | 0.818 | 0.900 | 0.730 | 0.843 | 0.793 | 0.883 | 0.069 | 0.259 | 0.231 | 0.461 | 0.191 | 0.407 | 0.240 | 0.455 |
| LNO | 0.397 | 0.622 | 0.687 | 0.689 | 0.394 | 0.619 | 0.386 | 0.617 | 0.128 | 0.354 | 0.440 | 0.638 | 0.334 | 0.553 | 0.518 | 0.597 |
| Transolver | 0.055 | 0.230 | 0.261 | 0.494 | 0.205 | 0.443 | 0.259 | 0.493 | 0.047 | 0.214 | 0.240 | 0.477 | 0.188 | 0.411 | 0.240 | 0.454 |
| DeepONet | 0.580 | 0.756 | 0.558 | 0.742 | 0.564 | 0.747 | 0.541 | 0.734 | 0.675 | 0.820 | 0.622 | 0.784 | 0.619 | 0.785 | 0.660 | 0.812 |

Table 22: S2Q reconstruction performance (MSE / RMSE) for the Type II multi-layer defect under single-frequency OOD (SFO). Models are trained on all frequencies (Normal) or on a single excitation frequency at 9, 36, or 81 kHz, and evaluated on irregular and regular grids.

| Model | Irregular Data | | | | | | | | Regular Data | | | | | | | |
|---|---|---|---|---|---|---|---|---|---|---|---|---|---|---|---|---|
| | Normal | | SFO-9kHz | | SFO-36kHz | | SFO-81kHz | | Normal | | SFO-9kHz | | SFO-36kHz | | SFO-81kHz | |
| | MSE | RMSE | MSE | RMSE | MSE | RMSE | MSE | RMSE | MSE | RMSE | MSE | RMSE | MSE | RMSE | MSE | RMSE |
| MLP | 0.522 | 0.720 | 0.484 | 0.692 | 0.541 | 0.733 | 0.580 | 0.760 | 0.434 | 0.656 | 0.507 | 0.694 | 0.412 | 0.638 | 0.519 | 0.716 |
| FNO | 0.010 | 0.077 | 0.267 | 0.472 | 0.162 | 0.345 | 0.237 | 0.390 | 0.014 | 0.105 | 0.213 | 0.416 | 0.124 | 0.319 | 0.240 | 0.432 |
| GeoFNO | 1.018 | 1.009 | 1.217 | 1.100 | 1.020 | 1.010 | 1.056 | 1.027 | 0.994 | 0.997 | / | / | / | / | 1.114 | 1.052 |
| FFNO | 0.123 | 0.334 | 0.597 | 0.752 | 0.439 | 0.637 | 0.496 | 0.669 | 0.024 | 0.148 | 0.237 | 0.440 | 0.123 | 0.321 | 0.212 | 0.416 |
| FCNO | 0.416 | 0.637 | 0.851 | 0.917 | 0.738 | 0.850 | 0.783 | 0.872 | 0.031 | 0.166 | 0.181 | 0.397 | 0.135 | 0.340 | 0.268 | 0.474 |
| LNO | 0.284 | 0.524 | 0.394 | 0.599 | 0.408 | 0.588 | 0.394 | 0.619 | 0.202 | 0.445 | 0.384 | 0.601 | 0.304 | 0.539 | 0.521 | 0.663 |
| Transolver | 0.126 | 0.347 | 0.232 | 0.454 | 0.247 | 0.481 | 0.292 | 0.507 | 0.042 | 0.184 | 0.260 | 0.475 | 0.156 | 0.373 | 0.319 | 0.536 |
| DeepONet | 0.490 | 0.698 | 0.517 | 0.716 | 0.534 | 0.728 | 0.530 | 0.726 | 0.563 | 0.748 | 0.537 | 0.726 | 0.565 | 0.750 | 0.610 | 0.780 |

Table 23: Full metric breakdown of T2Q Inverse Source Reconstruction under geometric OOD (fsplit). Models are trained on 5 defect types and tested on the held-out Type II multi-layer defect, evaluated on irregular and regular grids. Reported metrics include MSE, SSIM, RMSE, normalized RMSE (nRMSE), conserved value RMSE (cRMSE), maximum error (Max), and boundary RMSE (bRMSE).

| Model | Irregular Data | | | | | | | Regular Data | | | | | | |
|---|---|---|---|---|---|---|---|---|---|---|---|---|---|---|
| | MSE | SSIM | RMSE | nRMSE | cRMSE | Max | bRMSE | MSE | SSIM | RMSE | nRMSE | cRMSE | Max | bRMSE |
| MLP | 0.192 | 0.694 | 0.434 | 0.434 | 0.009 | 12.405 | 0.178 | 0.018 | 0.942 | 0.130 | 0.130 | 0.003 | 5.516 | 0.012 |
| FNO | 0.390 | 0.631 | 0.622 | 0.622 | 0.006 | 16.761 | 0.144 | 0.091 | 0.948 | 0.272 | 0.272 | 0.003 | 16.975 | 0.010 |
| GeoFNO | 0.210 | 0.545 | 0.455 | 0.455 | 0.010 | 12.422 | 0.166 | 0.018 | 0.900 | 0.130 | 0.130 | 0.007 | 4.554 | 0.014 |
| FFNO | 0.314 | 0.653 | 0.557 | 0.557 | 0.009 | 17.725 | 0.173 | 0.089 | 0.947 | 0.252 | 0.252 | 0.008 | 16.867 | 0.034 |
| FCNO | 0.264 | 0.655 | 0.511 | 0.511 | 0.008 | 15.235 | 0.318 | 0.086 | 0.947 | 0.255 | 0.255 | 0.008 | 16.368 | 0.048 |
| LNO | 0.346 | 0.702 | 0.568 | 0.568 | 0.023 | 23.519 | 0.100 | 0.247 | 0.818 | 0.482 | 0.482 | 0.004 | 24.960 | 0.008 |
| Transolver | 0.106 | 0.817 | 0.307 | 0.307 | 0.007 | 15.711 | 0.111 | 0.012 | 0.983 | 0.104 | 0.104 | 0.002 | 5.499 | 0.006 |
| DeepONet | 0.172 | 0.701 | 0.412 | 0.412 | 0.011 | 11.465 | 0.198 | 0.017 | 0.946 | 0.126 | 0.126 | 0.002 | 5.767 | 0.007 |

Table 24: T2Q reconstruction performance (MSE / RMSE) for the single layer defect configuration under normal training and three frequency-band OOD regimes (High-OOD, Mid-OOD, Low-OOD), on irregular and regular grids.

| Model | Irregular Data | | | | | | | | Regular Data | | | | | | | |
|---|---|---|---|---|---|---|---|---|---|---|---|---|---|---|---|---|
| | Normal | | High-OOD | | Mid-OOD | | Low-OOD | | Normal | | High-OOD | | Mid-OOD | | Low-OOD | |
| | MSE | RMSE | MSE | RMSE | MSE | RMSE | MSE | RMSE | MSE | RMSE | MSE | RMSE | MSE | RMSE | MSE | RMSE |
| MLP | 0.137 | 0.358 | 0.132 | 0.354 | 0.140 | 0.363 | 0.124 | 0.343 | 0.011 | 0.097 | 0.010 | 0.094 | 0.011 | 0.098 | 0.011 | 0.099 |
| FNO | 0.006 | 0.073 | 0.006 | 0.072 | 0.006 | 0.074 | 0.005 | 0.068 | 0.002 | 0.042 | 0.002 | 0.039 | 0.002 | 0.040 | 0.002 | 0.039 |
| GeoFNO | 0.181 | 0.419 | 0.176 | 0.413 | 0.175 | 0.411 | 0.169 | 0.404 | 0.015 | 0.115 | 0.011 | 0.103 | 0.011 | 0.099 | 0.015 | 0.113 |
| FFNO | 0.020 | 0.138 | 0.021 | 0.143 | 0.023 | 0.148 | 0.020 | 0.137 | 0.004 | 0.057 | 0.003 | 0.057 | 0.004 | 0.058 | 0.003 | 0.056 |
| FCNO | 0.029 | 0.166 | 0.030 | 0.171 | 0.034 | 0.178 | 0.028 | 0.163 | 0.004 | 0.063 | 0.004 | 0.062 | 0.004 | 0.063 | 0.004 | 0.062 |
| LNO | 0.160 | 0.396 | 0.181 | 0.421 | 0.166 | 0.403 | 0.140 | 0.371 | 0.081 | 0.281 | 0.079 | 0.277 | 0.088 | 0.291 | 0.079 | 0.274 |
| Transolver | 0.009 | 0.089 | 0.009 | 0.090 | 0.010 | 0.094 | 0.008 | 0.088 | 0.003 | 0.052 | 0.003 | 0.052 | 0.005 | 0.055 | 0.004 | 0.051 |
| DeepONet | 0.110 | 0.322 | 0.109 | 0.322 | 0.114 | 0.330 | 0.103 | 0.313 | 0.010 | 0.094 | 0.009 | 0.092 | 0.010 | 0.095 | 0.010 | 0.096 |

Table 25: T2Q reconstruction performance (MSE / RMSE) for the Type I double-layer defect under normal training and three frequency-band OOD regimes (High-OOD, Mid-OOD, Low-OOD), on irregular and regular grids.

| Model | Irregular Data | | | | | | | | Regular Data | | | | | | | |
|---|---|---|---|---|---|---|---|---|---|---|---|---|---|---|---|---|
| | Normal | | High-OOD | | Mid-OOD | | Low-OOD | | Normal | | High-OOD | | Mid-OOD | | Low-OOD | |
| | MSE | RMSE | MSE | RMSE | MSE | RMSE | MSE | RMSE | MSE | RMSE | MSE | RMSE | MSE | RMSE | MSE | RMSE |
| MLP | 0.308 | 0.551 | 0.320 | 0.560 | 0.322 | 0.563 | 0.309 | 0.552 | 0.090 | 0.277 | 0.083 | 0.271 | 0.103 | 0.286 | 0.099 | 0.287 |
| FNO | 0.019 | 0.128 | 0.023 | 0.139 | 0.021 | 0.135 | 0.017 | 0.122 | 0.010 | 0.089 | 0.007 | 0.080 | 0.010 | 0.092 | 0.010 | 0.091 |
| GeoFNO | 0.374 | 0.607 | 0.394 | 0.622 | 0.384 | 0.614 | 0.368 | 0.600 | 0.105 | 0.298 | 0.094 | 0.287 | 0.125 | 0.317 | 0.112 | 0.306 |
| FFNO | 0.091 | 0.299 | 0.095 | 0.305 | 0.100 | 0.313 | 0.089 | 0.296 | 0.052 | 0.203 | 0.042 | 0.192 | 0.062 | 0.213 | 0.050 | 0.205 |
| FCNO | 0.132 | 0.360 | 0.138 | 0.369 | 0.142 | 0.373 | 0.131 | 0.359 | 0.057 | 0.214 | 0.046 | 0.200 | 0.068 | 0.223 | 0.057 | 0.218 |
| LNO | 0.460 | 0.670 | 0.453 | 0.666 | 0.445 | 0.660 | 0.468 | 0.677 | 0.143 | 0.360 | 0.112 | 0.322 | 0.139 | 0.348 | 0.167 | 0.393 |
| Transolver | 0.042 | 0.197 | 0.046 | 0.205 | 0.044 | 0.201 | 0.035 | 0.181 | 0.046 | 0.200 | 0.031 | 0.172 | 0.041 | 0.192 | 0.037 | 0.186 |
| DeepONet | 0.287 | 0.531 | 0.302 | 0.543 | 0.301 | 0.543 | 0.295 | 0.537 | 0.080 | 0.258 | 0.071 | 0.250 | 0.093 | 0.269 | 0.088 | 0.269 |

Table 26: T2Q reconstruction performance (MSE / RMSE) for the Type II double-layer defect under normal training and three frequency-band OOD regimes (High-OOD, Mid-OOD, Low-OOD), on irregular and regular grids.

| Model | Irregular Data | | | | | | | | Regular Data | | | | | | | |
|---|---|---|---|---|---|---|---|---|---|---|---|---|---|---|---|---|
| | Normal | | High-OOD | | Mid-OOD | | Low-OOD | | Normal | | High-OOD | | Mid-OOD | | Low-OOD | |
| | MSE | RMSE | MSE | RMSE | MSE | RMSE | MSE | RMSE | MSE | RMSE | MSE | RMSE | MSE | RMSE | MSE | RMSE |
| MLP | 0.167 | 0.401 | 0.179 | 0.413 | 0.182 | 0.418 | 0.187 | 0.422 | 0.013 | 0.108 | 0.015 | 0.113 | 0.016 | 0.114 | 0.016 | 0.118 |
| FNO | 0.009 | 0.074 | 0.010 | 0.075 | 0.007 | 0.064 | 0.010 | 0.077 | 0.001 | 0.037 | 0.002 | 0.038 | 0.002 | 0.040 | 0.002 | 0.039 |
| GeoFNO | 0.176 | 0.414 | 0.239 | 0.482 | 0.217 | 0.459 | 0.203 | 0.440 | 0.030 | 0.165 | 0.017 | 0.121 | 0.026 | 0.147 | 0.024 | 0.143 |
| FFNO | 0.040 | 0.183 | 0.042 | 0.187 | 0.035 | 0.173 | 0.044 | 0.192 | 0.004 | 0.062 | 0.005 | 0.066 | 0.006 | 0.067 | 0.006 | 0.069 |
| FCNO | 0.074 | 0.258 | 0.076 | 0.259 | 0.065 | 0.243 | 0.078 | 0.263 | 0.006 | 0.071 | 0.006 | 0.073 | 0.007 | 0.075 | 0.007 | 0.077 |
| LNO | 0.302 | 0.541 | 0.306 | 0.543 | 0.310 | 0.549 | 0.304 | 0.540 | 0.217 | 0.460 | 0.172 | 0.407 | 0.186 | 0.424 | 0.238 | 0.480 |
| Transolver | 0.016 | 0.118 | 0.015 | 0.115 | 0.014 | 0.113 | 0.015 | 0.119 | 0.003 | 0.052 | 0.003 | 0.055 | 0.004 | 0.057 | 0.004 | 0.056 |
| DeepONet | 0.140 | 0.367 | 0.152 | 0.380 | 0.151 | 0.380 | 0.160 | 0.390 | 0.012 | 0.104 | 0.014 | 0.108 | 0.015 | 0.109 | 0.016 | 0.114 |

Table 27: T2Q reconstruction performance (MSE / RMSE) for the Type III double-layer defect under normal training and three frequency-band OOD regimes (High-OOD, Mid-OOD, Low-OOD), on irregular and regular grids.

| Model | Irregular Data | | | | | | | | Regular Data | | | | | | | |
|---|---|---|---|---|---|---|---|---|---|---|---|---|---|---|---|---|
| | Normal | | High-OOD | | Mid-OOD | | Low-OOD | | Normal | | High-OOD | | Mid-OOD | | Low-OOD | |
| | MSE | RMSE | MSE | RMSE | MSE | RMSE | MSE | RMSE | MSE | RMSE | MSE | RMSE | MSE | RMSE | MSE | RMSE |
| MLP | 0.221 | 0.463 | 0.225 | 0.466 | 0.222 | 0.467 | 0.207 | 0.452 | 0.075 | 0.466 | 0.072 | 0.262 | 0.069 | 0.259 | 0.064 | 0.249 |
| FNO | 0.008 | 0.067 | 0.007 | 0.066 | 0.126 | 0.311 | 0.133 | 0.323 | 0.009 | 0.066 | 0.051 | 0.219 | 0.048 | 0.212 | 0.050 | 0.215 |
| GeoFNO | 0.267 | 0.508 | 0.260 | 0.501 | 0.259 | 0.505 | 0.458 | 0.675 | 0.084 | 0.501 | 0.084 | 0.286 | 0.081 | 0.282 | 0.076 | 0.272 |
| FFNO | 0.057 | 0.226 | 0.058 | 0.229 | 0.185 | 0.421 | 0.180 | 0.416 | 0.043 | 0.229 | 0.050 | 0.217 | 0.047 | 0.212 | 0.046 | 0.210 |
| FCNO | 0.108 | 0.320 | 0.111 | 0.324 | 0.202 | 0.446 | 0.193 | 0.435 | 0.049 | 0.324 | 0.051 | 0.219 | 0.048 | 0.214 | 0.046 | 0.210 |
| LNO | 0.336 | 0.566 | 0.340 | 0.570 | 0.294 | 0.533 | 0.288 | 0.526 | 0.118 | 0.570 | 0.131 | 0.351 | 0.127 | 0.351 | 0.140 | 0.369 |
| Transolver | 0.038 | 0.188 | 0.039 | 0.190 | 0.093 | 0.300 | 0.085 | 0.288 | 0.045 | 0.190 | 0.052 | 0.222 | 0.048 | 0.215 | 0.048 | 0.214 |
| DeepONet | 0.213 | 0.454 | 0.212 | 0.452 | 0.206 | 0.449 | 0.192 | 0.434 | 0.065 | 0.452 | 0.063 | 0.245 | 0.060 | 0.241 | 0.056 | 0.232 |

Table 28: T2Q reconstruction performance (MSE / RMSE) for the Type I multi-layer defect under normal training and three frequency-band OOD regimes (High-OOD, Mid-OOD, Low-OOD), on irregular and regular grids.

| Model | Irregular Data | | | | | | | | Regular Data | | | | | | | |
|---|---|---|---|---|---|---|---|---|---|---|---|---|---|---|---|---|
| | Normal | | High-OOD | | Mid-OOD | | Low-OOD | | Normal | | High-OOD | | Mid-OOD | | Low-OOD | |
| | MSE | RMSE | MSE | RMSE | MSE | RMSE | MSE | RMSE | MSE | RMSE | MSE | RMSE | MSE | RMSE | MSE | RMSE |
| MLP | 0.142 | 0.370 | 0.149 | 0.379 | 0.145 | 0.375 | 0.141 | 0.369 | 0.027 | 0.157 | 0.028 | 0.159 | 0.033 | 0.170 | 0.030 | 0.161 |
| FNO | 0.008 | 0.066 | 0.012 | 0.082 | 0.010 | 0.069 | 0.009 | 0.071 | 0.002 | 0.042 | 0.003 | 0.046 | 0.003 | 0.044 | 0.003 | 0.045 |
| GeoFNO | 0.155 | 0.388 | 0.151 | 0.382 | 0.154 | 0.387 | 0.151 | 0.383 | 0.036 | 0.185 | 0.029 | 0.162 | 0.041 | 0.193 | 0.036 | 0.181 |
| FFNO | 0.032 | 0.162 | 0.039 | 0.177 | 0.032 | 0.159 | 0.032 | 0.161 | 0.010 | 0.099 | 0.010 | 0.099 | 0.012 | 0.104 | 0.010 | 0.097 |
| FCNO | 0.054 | 0.216 | 0.063 | 0.232 | 0.053 | 0.213 | 0.054 | 0.215 | 0.015 | 0.118 | 0.015 | 0.118 | 0.018 | 0.127 | 0.015 | 0.118 |
| LNO | 0.308 | 0.543 | 0.290 | 0.526 | 0.290 | 0.526 | 0.293 | 0.530 | 0.272 | 0.506 | 0.256 | 0.493 | 0.289 | 0.521 | 0.246 | 0.482 |
| Transolver | 0.014 | 0.114 | 0.018 | 0.126 | 0.014 | 0.111 | 0.015 | 0.116 | 0.016 | 0.118 | 0.013 | 0.109 | 0.017 | 0.117 | 0.014 | 0.112 |
| DeepONet | 0.130 | 0.355 | 0.136 | 0.361 | 0.131 | 0.356 | 0.127 | 0.351 | 0.026 | 0.152 | 0.026 | 0.154 | 0.032 | 0.165 | 0.028 | 0.157 |

Table 29: T2Q reconstruction performance (MSE / RMSE) for the Type II multi-layer defect under normal training and three frequency-band OOD regimes (High-OOD, Mid-OOD, Low-OOD), on irregular and regular grids.

| Model | Irregular Data | | | | | | | | Regular Data | | | | | | | |
|---|---|---|---|---|---|---|---|---|---|---|---|---|---|---|---|---|
| | Normal | | High-OOD | | Mid-OOD | | Low-OOD | | Normal | | High-OOD | | Mid-OOD | | Low-OOD | |
| | MSE | RMSE | MSE | RMSE | MSE | RMSE | MSE | RMSE | MSE | RMSE | MSE | RMSE | MSE | RMSE | MSE | RMSE |
| MLP | 0.154 | 0.387 | 0.152 | 0.385 | 0.146 | 0.376 | 0.148 | 0.380 | 0.010 | 0.096 | 0.010 | 0.096 | 0.010 | 0.096 | 0.011 | 0.100 |
| FNO | 0.047 | 0.190 | 0.054 | 0.214 | 0.041 | 0.180 | 0.046 | 0.190 | 0.003 | 0.046 | 0.002 | 0.047 | 0.002 | 0.045 | 0.003 | 0.051 |
| GeoFNO | 0.177 | 0.418 | 0.173 | 0.412 | 0.159 | 0.394 | 0.175 | 0.415 | 0.013 | 0.112 | 0.017 | 0.127 | 0.018 | 0.133 | 0.016 | 0.123 |
| FFNO | 0.091 | 0.287 | 0.098 | 0.302 | 0.084 | 0.280 | 0.085 | 0.280 | 0.003 | 0.052 | 0.003 | 0.052 | 0.003 | 0.050 | 0.004 | 0.055 |
| FCNO | 0.105 | 0.314 | 0.106 | 0.316 | 0.098 | 0.306 | 0.095 | 0.300 | 0.004 | 0.056 | 0.003 | 0.055 | 0.003 | 0.054 | 0.004 | 0.059 |
| LNO | 0.222 | 0.457 | 0.219 | 0.456 | 0.257 | 0.498 | 0.226 | 0.464 | 0.200 | 0.444 | 0.173 | 0.413 | 0.207 | 0.451 | 0.215 | 0.460 |
| Transolver | 0.021 | 0.131 | 0.021 | 0.137 | 0.020 | 0.134 | 0.019 | 0.131 | 0.003 | 0.047 | 0.002 | 0.045 | 0.002 | 0.044 | 0.004 | 0.051 |
| DeepONet | 0.120 | 0.339 | 0.118 | 0.338 | 0.113 | 0.330 | 0.114 | 0.332 | 0.008 | 0.084 | 0.008 | 0.084 | 0.008 | 0.084 | 0.008 | 0.088 |

Table 30: T2Q reconstruction performance (MSE / RMSE) for the Type I double-layer defect under single-frequency OOD (SFO). Models are trained on all frequencies (Normal) or on a single excitation frequency at 9, 36, or 81 kHz, and evaluated on irregular and regular grids.

| Model | Irregular Data | | | | | | | | Regular Data | | | | | | | |
|---|---|---|---|---|---|---|---|---|---|---|---|---|---|---|---|---|
| | Normal | | SFO-9kHz | | SFO-36kHz | | SFO-81kHz | | Normal | | SFO-9kHz | | SFO-36kHz | | SFO-81kHz | |
| | MSE | RMSE | MSE | RMSE | MSE | RMSE | MSE | RMSE | MSE | RMSE | MSE | RMSE | MSE | RMSE | MSE | RMSE |
| MLP | 0.308 | 0.551 | 0.347 | 0.581 | 0.305 | 0.548 | 0.331 | 0.568 | 0.090 | 0.277 | 0.151 | 0.364 | 0.168 | 0.384 | 0.163 | 0.375 |
| FNO | 0.019 | 0.128 | 0.217 | 0.446 | 0.207 | 0.427 | 0.202 | 0.419 | 0.010 | 0.089 | 0.329 | 0.520 | 0.120 | 0.324 | 0.206 | 0.412 |
| GeoFNO | 0.374 | 0.607 | 0.398 | 0.624 | 0.417 | 0.642 | 0.462 | 0.669 | 0.105 | 0.298 | 0.243 | 0.468 | 0.264 | 0.466 | 0.231 | 0.450 |
| FFNO | 0.091 | 0.299 | 0.237 | 0.473 | 0.195 | 0.435 | 0.214 | 0.454 | 0.052 | 0.203 | 0.122 | 0.326 | 0.119 | 0.321 | 0.150 | 0.353 |
| FCNO | 0.132 | 0.360 | 0.239 | 0.477 | 0.214 | 0.456 | 0.230 | 0.475 | 0.057 | 0.214 | 0.127 | 0.336 | 0.170 | 0.373 | 0.255 | 0.466 |
| LNO | 0.460 | 0.670 | / | / | 0.569 | 0.749 | 1.273 | 1.022 | 0.143 | 0.360 | 0.462 | 0.662 | 0.409 | 0.630 | 0.433 | 0.647 |
| Transolver | 0.042 | 0.197 | 0.286 | 0.513 | 0.226 | 0.464 | 0.216 | 0.448 | 0.046 | 0.200 | 0.209 | 0.425 | 0.166 | 0.357 | 0.122 | 0.318 |
| DeepONet | 0.287 | 0.531 | 0.329 | 0.567 | 0.293 | 0.536 | 0.322 | 0.560 | 0.080 | 0.258 | 0.141 | 0.347 | 0.149 | 0.356 | 0.154 | 0.363 |

Table 31: T2Q reconstruction performance (MSE / RMSE) for the Type III double-layer defect under single-frequency OOD (SFO). Models are trained on all frequencies (Normal) or on a single excitation frequency at 9, 36, or 81 kHz, and evaluated on irregular and regular grids.

| Model | Irregular Data | | | | | | | | Regular Data | | | | | | | |
|---|---|---|---|---|---|---|---|---|---|---|---|---|---|---|---|---|
| | Normal | | SFO-9kHz | | SFO-36kHz | | SFO-81kHz | | Normal | | SFO-9kHz | | SFO-36kHz | | SFO-81kHz | |
| | MSE | RMSE | MSE | RMSE | MSE | RMSE | MSE | RMSE | MSE | RMSE | MSE | RMSE | MSE | RMSE | MSE | RMSE |
| MLP | 0.221 | 0.463 | 0.226 | 0.470 | 0.218 | 0.461 | 0.213 | 0.457 | 0.075 | 0.466 | 0.077 | 0.272 | 0.069 | 0.258 | 0.074 | 0.269 |
| FNO | 0.008 | 0.067 | 0.116 | 0.318 | 0.085 | 0.265 | 0.099 | 0.285 | 0.009 | 0.066 | 0.150 | 0.373 | 0.107 | 0.304 | 0.111 | 0.315 |
| GeoFNO | 0.267 | 0.508 | 0.274 | 0.518 | 0.262 | 0.506 | 0.265 | 0.510 | 0.084 | 0.501 | / | / | / | / | / | / |
| FFNO | 0.057 | 0.226 | 0.199 | 0.438 | 0.164 | 0.397 | 0.173 | 0.407 | 0.043 | 0.229 | 0.061 | 0.241 | 0.051 | 0.222 | 0.059 | 0.239 |
| FCNO | 0.108 | 0.320 | 0.211 | 0.455 | 0.193 | 0.434 | 0.201 | 0.442 | 0.049 | 0.324 | 0.061 | 0.241 | 0.056 | 0.232 | 0.062 | 0.243 |
| LNO | 0.336 | 0.566 | 0.360 | 0.586 | 0.354 | 0.586 | 0.317 | 0.555 | 0.118 | 0.570 | 0.399 | 0.620 | 0.277 | 0.507 | 0.224 | 0.458 |
| Transolver | 0.038 | 0.188 | 0.138 | 0.358 | 0.116 | 0.334 | 0.106 | 0.320 | 0.045 | 0.190 | 0.109 | 0.316 | 0.065 | 0.250 | 0.068 | 0.257 |
| DeepONet | 0.213 | 0.454 | 0.207 | 0.449 | 0.210 | 0.453 | 0.210 | 0.454 | 0.065 | 0.452 | 0.065 | 0.250 | 0.065 | 0.250 | 0.073 | 0.264 |

Table 32: T2Q reconstruction performance (MSE / RMSE) for the Type II multi-layer defect under single-frequency OOD (SFO). Models are trained on all frequencies (Normal) or on a single excitation frequency at 9, 36, or 81 kHz, and evaluated on irregular and regular grids.

| Model | Irregular Data | | | | | | | | Regular Data | | | | | | | |
|---|---|---|---|---|---|---|---|---|---|---|---|---|---|---|---|---|
| | Normal | | SFO-9kHz | | SFO-36kHz | | SFO-81kHz | | Normal | | SFO-9kHz | | SFO-36kHz | | SFO-81kHz | |
| | MSE | RMSE | MSE | RMSE | MSE | RMSE | MSE | RMSE | MSE | RMSE | MSE | RMSE | MSE | RMSE | MSE | RMSE |
| MLP | 0.154 | 0.387 | 0.166 | 0.395 | 0.168 | 0.396 | 0.194 | 0.430 | 0.010 | 0.096 | 0.021 | 0.137 | 0.016 | 0.116 | 0.019 | 0.125 |
| FNO | 0.047 | 0.190 | 0.063 | 0.235 | 0.058 | 0.205 | 0.087 | 0.255 | 0.003 | 0.046 | 0.031 | 0.166 | 0.020 | 0.126 | 0.026 | 0.145 |
| GeoFNO | 0.177 | 0.418 | 0.936 | 0.860 | 0.232 | 0.472 | 0.224 | 0.465 | 0.013 | 0.112 | / | / | 0.118 | 0.260 | 0.044 | 0.186 |
| FFNO | 0.091 | 0.287 | 0.108 | 0.314 | 0.106 | 0.299 | 0.155 | 0.363 | 0.003 | 0.052 | 0.017 | 0.125 | 0.013 | 0.098 | 0.018 | 0.117 |
| FCNO | 0.105 | 0.314 | 0.126 | 0.343 | 0.127 | 0.337 | 0.158 | 0.375 | 0.004 | 0.056 | 0.025 | 0.150 | 0.017 | 0.116 | 0.022 | 0.130 |
| LNO | 0.222 | 0.457 | 0.385 | 0.599 | 0.349 | 0.580 | 0.384 | 0.612 | 0.200 | 0.444 | 0.512 | 0.683 | 0.323 | 0.551 | 0.468 | 0.642 |
| Transolver | 0.021 | 0.131 | 0.116 | 0.314 | 0.096 | 0.283 | 0.146 | 0.341 | 0.003 | 0.047 | 0.072 | 0.238 | 0.032 | 0.162 | 0.033 | 0.171 |
| DeepONet | 0.120 | 0.339 | 0.135 | 0.354 | 0.161 | 0.381 | 0.171 | 0.393 | 0.008 | 0.084 | 0.014 | 0.111 | 0.012 | 0.099 | 0.017 | 0.116 |

Table 33: Full metric breakdown of Q2T Forward Modeling under geometric OOD (fsplit). Models are trained on 5 defect types and tested on the held-out Type II multi-layer defect, evaluated on irregular and regular grids. Reported metrics include MSE, SSIM, RMSE, normalized RMSE (nRMSE), conserved value RMSE (cRMSE), maximum error (Max), and boundary RMSE (bRMSE).

| Model | Irregular Data | | | | | | | Regular Data | | | | | | |
|---|---|---|---|---|---|---|---|---|---|---|---|---|---|---|
| | MSE | SSIM | RMSE | nRMSE | cRMSE | Max | bRMSE | MSE | SSIM | RMSE | nRMSE | cRMSE | Max | bRMSE |
| MLP | 0.189 | 0.739 | 0.432 | 0.432 | 0.007 | 7.711 | 0.258 | 0.108 | 0.887 | 0.325 | 0.325 | 0.006 | 9.253 | 0.031 |
| FNO | 0.428 | 0.666 | 0.652 | 0.653 | 0.010 | 9.500 | 0.423 | 0.034 | 0.964 | 0.174 | 0.174 | 0.002 | 8.270 | 0.014 |
| GeoFNO | 0.398 | 0.463 | 0.626 | 0.626 | 0.037 | 8.234 | 0.378 | 0.232 | 0.731 | 0.466 | 0.466 | 0.006 | 9.574 | 0.114 |
| FFNO | 0.353 | 0.647 | 0.592 | 0.592 | 0.013 | 9.079 | 0.288 | 0.022 | 0.939 | 0.145 | 0.145 | 0.014 | 6.429 | 0.026 |
| FCNO | 0.487 | 0.460 | 0.693 | 0.693 | 0.014 | 8.741 | 0.333 | 0.036 | 0.930 | 0.182 | 0.182 | 0.013 | 8.599 | 0.174 |
| LNO | 0.094 | 0.862 | 0.285 | 0.285 | 0.023 | 7.515 | 0.131 | 0.101 | 0.936 | 0.294 | 0.294 | 0.006 | 15.411 | 0.013 |
| Transolver | 0.098 | 0.921 | 0.296 | 0.296 | 0.005 | 10.585 | 0.138 | 0.037 | 0.968 | 0.175 | 0.175 | 0.004 | 8.700 | 0.016 |

Table 34: Q2T forward modeling performance (MSE / RMSE) for the single layer defect configuration under normal training and three frequency-band OOD regimes (High-OOD, Mid-OOD, Low-OOD), on irregular and regular grids.

| Model | Irregular Data | | | | | | | | Regular Data | | | | | | | |
|---|---|---|---|---|---|---|---|---|---|---|---|---|---|---|---|---|
| | Normal | | High-OOD | | Mid-OOD | | Low-OOD | | Normal | | High-OOD | | Mid-OOD | | Low-OOD | |
| | MSE | RMSE | MSE | RMSE | MSE | RMSE | MSE | RMSE | MSE | RMSE | MSE | RMSE | MSE | RMSE | MSE | RMSE |
| MLP | 0.186 | 0.423 | 0.172 | 0.408 | 0.167 | 0.404 | 0.172 | 0.409 | 0.111 | 0.324 | 0.112 | 0.326 | 0.113 | 0.326 | 0.108 | 0.320 |
| FNO | 0.006 | 0.068 | 0.006 | 0.066 | 0.006 | 0.067 | 0.006 | 0.066 | 0.003 | 0.047 | 0.003 | 0.049 | 0.003 | 0.048 | 0.003 | 0.048 |
| GeoFNO | 0.377 | 0.608 | 0.362 | 0.596 | 0.349 | 0.586 | 0.360 | 0.595 | 0.241 | 0.474 | 0.248 | 0.481 | 0.243 | 0.475 | 0.241 | 0.473 |
| FFNO | 0.032 | 0.171 | 0.031 | 0.168 | 0.031 | 0.168 | 0.030 | 0.167 | 0.011 | 0.100 | 0.011 | 0.100 | 0.011 | 0.101 | 0.011 | 0.099 |
| FCNO | 0.050 | 0.217 | 0.048 | 0.213 | 0.049 | 0.215 | 0.048 | 0.213 | 0.014 | 0.115 | 0.015 | 0.116 | 0.014 | 0.116 | 0.014 | 0.114 |
| LNO | 0.069 | 0.243 | 0.056 | 0.216 | 0.069 | 0.242 | 0.068 | 0.240 | 0.139 | 0.364 | 0.127 | 0.348 | 0.135 | 0.359 | 0.127 | 0.348 |
| Transolver | 0.007 | 0.067 | 0.006 | 0.067 | 0.006 | 0.067 | 0.006 | 0.066 | 0.003 | 0.051 | 0.005 | 0.063 | 0.004 | 0.054 | 0.004 | 0.052 |

Table 35: Q2T forward modeling performance (MSE / RMSE) for the Type I double-layer defect under normal training and three frequency-band OOD regimes (High-OOD, Mid-OOD, Low-OOD), on irregular and regular grids.

| Model | Irregular Data | | | | | | | | Regular Data | | | | | | | |
|---|---|---|---|---|---|---|---|---|---|---|---|---|---|---|---|---|
| | Normal | | High-OOD | | Mid-OOD | | Low-OOD | | Normal | | High-OOD | | Mid-OOD | | Low-OOD | |
| | MSE | RMSE | MSE | RMSE | MSE | RMSE | MSE | RMSE | MSE | RMSE | MSE | RMSE | MSE | RMSE | MSE | RMSE |
| MLP | 0.203 | 0.443 | 0.196 | 0.435 | 0.203 | 0.443 | 0.194 | 0.433 | 0.160 | 0.388 | 0.143 | 0.371 | 0.153 | 0.380 | 0.155 | 0.384 |
| FNO | 0.011 | 0.081 | 0.011 | 0.078 | 0.012 | 0.082 | 0.011 | 0.078 | 0.008 | 0.075 | 0.007 | 0.072 | 0.008 | 0.074 | 0.008 | 0.074 |
| GeoFNO | 0.455 | 0.670 | 0.468 | 0.681 | 0.459 | 0.673 | 0.478 | 0.688 | 0.337 | 0.568 | 0.310 | 0.545 | 0.323 | 0.555 | 0.330 | 0.563 |
| FFNO | 0.059 | 0.224 | 0.057 | 0.220 | 0.059 | 0.224 | 0.058 | 0.221 | 0.028 | 0.152 | 0.024 | 0.144 | 0.029 | 0.154 | 0.030 | 0.156 |
| FCNO | 0.086 | 0.276 | 0.084 | 0.272 | 0.085 | 0.274 | 0.083 | 0.271 | 0.033 | 0.166 | 0.028 | 0.158 | 0.033 | 0.167 | 0.034 | 0.170 |
| LNO | 0.075 | 0.241 | 0.125 | 0.324 | 0.087 | 0.260 | 0.070 | 0.229 | 0.088 | 0.274 | 0.105 | 0.309 | 0.075 | 0.253 | 0.114 | 0.321 |
| Transolver | 0.015 | 0.100 | 0.015 | 0.100 | 0.015 | 0.101 | 0.015 | 0.099 | 0.011 | 0.086 | 0.010 | 0.084 | 0.011 | 0.089 | 0.010 | 0.085 |

Table 36: Q2T forward modeling performance (MSE / RMSE) for the Type II double-layer defect under normal training and three frequency-band OOD regimes (High-OOD, Mid-OOD, Low-OOD), on irregular and regular grids.

| Model | Irregular Data | | | | | | | | Regular Data | | | | | | | |
| | Normal | | High-OOD | | Mid-OOD | | Low-OOD | | Normal | | High-OOD | | Mid-OOD | | Low-OOD | |
| | MSE | RMSE | MSE | RMSE | MSE | RMSE | MSE | RMSE | MSE | RMSE | MSE | RMSE | MSE | RMSE | MSE | RMSE |
|---|---|---|---|---|---|---|---|---|---|---|---|---|---|---|---|---|
| MLP | 0.226 | 0.469 | 0.228 | 0.472 | 0.234 | 0.477 | 0.232 | 0.475 | 0.155 | 0.377 | 0.143 | 0.365 | 0.167 | 0.392 | 0.171 | 0.395 |
| FNO | 0.027 | 0.147 | 0.025 | 0.141 | 0.025 | 0.141 | 0.027 | 0.146 | 0.003 | 0.050 | 0.004 | 0.052 | 0.003 | 0.051 | 0.003 | 0.052 |
| GeoFNO | 0.416 | 0.639 | 0.421 | 0.643 | 0.430 | 0.649 | 0.425 | 0.646 | 0.281 | 0.510 | 0.263 | 0.495 | 0.291 | 0.520 | 0.291 | 0.519 |
| FFNO | 0.158 | 0.394 | 0.154 | 0.390 | 0.156 | 0.391 | 0.158 | 0.394 | 0.020 | 0.135 | 0.020 | 0.135 | 0.022 | 0.140 | 0.023 | 0.141 |
| FCNO | 0.344 | 0.584 | 0.348 | 0.588 | 0.347 | 0.587 | 0.347 | 0.586 | 0.024 | 0.150 | 0.025 | 0.152 | 0.027 | 0.156 | 0.028 | 0.159 |
| LNO | 0.152 | 0.369 | 0.156 | 0.375 | 0.160 | 0.373 | 0.177 | 0.402 | 0.189 | 0.421 | 0.181 | 0.413 | 0.196 | 0.427 | 0.187 | 0.416 |
| Transolver | 0.007 | 0.074 | 0.007 | 0.074 | 0.008 | 0.077 | 0.007 | 0.075 | 0.005 | 0.060 | 0.005 | 0.061 | 0.006 | 0.063 | 0.004 | 0.059 |

Table 37: Q2T forward modeling performance (MSE / RMSE) for the Type III double-layer defect under normal training and three frequency-band OOD regimes (High-OOD, Mid-OOD, Low-OOD), on irregular and regular grids.

| Model | Irregular Data | | | | | | | | Regular Data | | | | | | | |
| | Normal | | High-OOD | | Mid-OOD | | Low-OOD | | Normal | | High-OOD | | Mid-OOD | | Low-OOD | |
| | MSE | RMSE | MSE | RMSE | MSE | RMSE | MSE | RMSE | MSE | RMSE | MSE | RMSE | MSE | RMSE | MSE | RMSE |
|---|---|---|---|---|---|---|---|---|---|---|---|---|---|---|---|---|
| MLP | 0.185 | 0.424 | 0.181 | 0.420 | 0.161 | 0.395 | 0.158 | 0.392 | 0.120 | 0.344 | 0.119 | 0.341 | 0.114 | 0.334 | 0.114 | 0.333 |
| FNO | 0.041 | 0.169 | 0.028 | 0.140 | 0.157 | 0.357 | 0.153 | 0.356 | 0.007 | 0.066 | 0.013 | 0.101 | 0.016 | 0.111 | 0.017 | 0.115 |
| GeoFNO | 0.377 | 0.611 | 0.387 | 0.619 | 0.371 | 0.607 | 0.364 | 0.601 | 0.231 | 0.474 | 0.231 | 0.473 | 0.228 | 0.470 | 0.232 | 0.474 |
| FFNO | 0.155 | 0.387 | 0.135 | 0.361 | 0.186 | 0.427 | 0.184 | 0.426 | 0.023 | 0.142 | 0.023 | 0.142 | 0.024 | 0.145 | 0.025 | 0.147 |
| FCNO | 0.324 | 0.568 | 0.316 | 0.561 | 0.369 | 0.606 | 0.365 | 0.603 | 0.026 | 0.151 | 0.026 | 0.152 | 0.027 | 0.154 | 0.028 | 0.155 |
| LNO | 0.110 | 0.312 | 0.061 | 0.226 | 0.060 | 0.226 | 0.049 | 0.205 | 0.045 | 0.201 | 0.049 | 0.208 | 0.047 | 0.205 | 0.058 | 0.229 |
| Transolver | 0.017 | 0.106 | 0.016 | 0.105 | 0.023 | 0.130 | 0.023 | 0.131 | 0.010 | 0.088 | 0.014 | 0.105 | 0.015 | 0.109 | 0.016 | 0.111 |

Table 38: Q2T forward modeling performance (MSE / RMSE) for the Type I multi-layer defect under normal training and three frequency-band OOD regimes (High-OOD, Mid-OOD, Low-OOD), on irregular and regular grids.

| Model | Irregular Data | | | | | | | | Regular Data | | | | | | | |
| | Normal | | High-OOD | | Mid-OOD | | Low-OOD | | Normal | | High-OOD | | Mid-OOD | | Low-OOD | |
| | MSE | RMSE | MSE | RMSE | MSE | RMSE | MSE | RMSE | MSE | RMSE | MSE | RMSE | MSE | RMSE | MSE | RMSE |
|---|---|---|---|---|---|---|---|---|---|---|---|---|---|---|---|---|
| MLP | 0.153 | 0.384 | 0.156 | 0.389 | 0.162 | 0.394 | 0.158 | 0.391 | 0.107 | 0.324 | 0.105 | 0.321 | 0.106 | 0.322 | 0.105 | 0.321 |
| FNO | 0.025 | 0.141 | 0.024 | 0.139 | 0.028 | 0.149 | 0.029 | 0.150 | 0.004 | 0.057 | 0.004 | 0.059 | 0.004 | 0.053 | 0.005 | 0.061 |
| GeoFNO | 0.381 | 0.612 | 0.383 | 0.614 | 0.388 | 0.617 | 0.383 | 0.613 | 0.242 | 0.478 | 0.248 | 0.484 | 0.246 | 0.483 | 0.245 | 0.481 |
| FFNO | 0.147 | 0.382 | 0.150 | 0.385 | 0.155 | 0.391 | 0.153 | 0.389 | 0.023 | 0.144 | 0.021 | 0.140 | 0.021 | 0.138 | 0.021 | 0.138 |
| FCNO | 0.351 | 0.589 | 0.356 | 0.593 | 0.358 | 0.595 | 0.352 | 0.590 | 0.026 | 0.155 | 0.024 | 0.150 | 0.024 | 0.150 | 0.024 | 0.149 |
| LNO | 0.104 | 0.301 | 0.088 | 0.277 | 0.102 | 0.298 | 0.099 | 0.295 | 0.131 | 0.350 | 0.130 | 0.348 | 0.152 | 0.378 | 0.149 | 0.373 |
| Transolver | 0.012 | 0.100 | 0.012 | 0.101 | 0.014 | 0.106 | 0.013 | 0.104 | 0.008 | 0.081 | 0.008 | 0.082 | 0.007 | 0.077 | 0.008 | 0.083 |

Table 39: Q2T forward modeling performance (MSE / RMSE) for the Type II multi-layer defect under normal training and three frequency-band OOD regimes (High-OOD, Mid-OOD, Low-OOD), on irregular and regular grids.

| Model | Irregular Data | | | | | | | | Regular Data | | | | | | | |
|---|---|---|---|---|---|---|---|---|---|---|---|---|---|---|---|---|
| | Normal | | High-OOD | | Mid-OOD | | Low-OOD | | Normal | | High-OOD | | Mid-OOD | | Low-OOD | |
| | MSE | RMSE | MSE | RMSE | MSE | RMSE | MSE | RMSE | MSE | RMSE | MSE | RMSE | MSE | RMSE | MSE | RMSE |
| MLP | 0.175 | 0.414 | 0.180 | 0.421 | 0.180 | 0.420 | 0.173 | 0.411 | 0.088 | 0.291 | 0.089 | 0.294 | 0.090 | 0.296 | 0.088 | 0.292 |
| FNO | 0.096 | 0.285 | 0.109 | 0.309 | 0.093 | 0.281 | 0.108 | 0.301 | 0.002 | 0.041 | 0.002 | 0.043 | 0.002 | 0.043 | 0.002 | 0.040 |
| GeoFNO | 0.389 | 0.619 | 0.395 | 0.624 | 0.398 | 0.626 | 0.389 | 0.619 | 0.215 | 0.447 | 0.216 | 0.449 | 0.218 | 0.450 | 0.217 | 0.449 |
| FFNO | 0.170 | 0.410 | 0.180 | 0.422 | 0.165 | 0.404 | 0.169 | 0.409 | 0.005 | 0.068 | 0.006 | 0.071 | 0.006 | 0.072 | 0.005 | 0.069 |
| FCNO | 0.370 | 0.605 | 0.377 | 0.610 | 0.374 | 0.608 | 0.373 | 0.607 | 0.007 | 0.079 | 0.008 | 0.082 | 0.008 | 0.083 | 0.007 | 0.079 |
| LNO | 0.083 | 0.262 | 0.097 | 0.288 | 0.091 | 0.281 | 0.054 | 0.215 | 0.126 | 0.350 | 0.101 | 0.313 | 0.118 | 0.336 | 0.120 | 0.340 |
| Transolver | 0.008 | 0.076 | 0.008 | 0.077 | 0.008 | 0.076 | 0.008 | 0.077 | 0.002 | 0.044 | 0.003 | 0.047 | 0.003 | 0.047 | 0.002 | 0.044 |

Table 40: Q2T forward modeling performance (MSE / RMSE) for the Type I double-layer defect under single-frequency OOD (SFO). Models are trained on all frequencies (Normal) or on a single excitation frequency at 9, 36, or 81 kHz, and evaluated on irregular and regular grids.

| Model | Irregular Data | | | | | | | | Regular Data | | | | | | | |
|---|---|---|---|---|---|---|---|---|---|---|---|---|---|---|---|---|
| | Normal | | SFO-9kHz | | SFO-36kHz | | SFO-81kHz | | Normal | | SFO-9kHz | | SFO-36kHz | | SFO-81kHz | |
| | MSE | RMSE | MSE | RMSE | MSE | RMSE | MSE | RMSE | MSE | RMSE | MSE | RMSE | MSE | RMSE | MSE | RMSE |
| MLP | 0.203 | 0.443 | 0.207 | 0.436 | 0.186 | 0.412 | 0.225 | 0.448 | 0.160 | 0.388 | 0.225 | 0.454 | 0.196 | 0.426 | 0.222 | 0.451 |
| FNO | 0.011 | 0.081 | 0.140 | 0.348 | 0.103 | 0.282 | 0.133 | 0.309 | 0.008 | 0.075 | 0.163 | 0.375 | 0.135 | 0.329 | 0.203 | 0.386 |
| GeoFNO | 0.455 | 0.670 | 0.488 | 0.695 | 0.739 | 0.822 | 0.699 | 0.808 | 0.337 | 0.568 | / | / | / | / | 1.899 | 1.034 |
| FFNO | 0.059 | 0.224 | 0.165 | 0.384 | 0.136 | 0.338 | 0.179 | 0.380 | 0.028 | 0.152 | 0.166 | 0.381 | 0.130 | 0.327 | 0.202 | 0.393 |
| FCNO | 0.086 | 0.276 | 0.176 | 0.399 | 0.149 | 0.359 | 0.189 | 0.399 | 0.033 | 0.166 | 0.163 | 0.377 | 0.125 | 0.323 | 0.173 | 0.370 |
| LNO | 0.075 | 0.241 | 0.288 | 0.489 | 0.198 | 0.406 | 1.778 | 0.731 | 0.088 | 0.274 | 0.284 | 0.515 | 0.274 | 0.490 | 1.000 | 1.000 |
| Transolver | 0.015 | 0.100 | 0.143 | 0.354 | 0.111 | 0.292 | 0.147 | 0.320 | 0.011 | 0.086 | 0.158 | 0.369 | 0.128 | 0.322 | 0.166 | 0.354 |

Table 41: Q2T forward modeling performance (MSE / RMSE) for the Type III double-layer defect under single-frequency OOD (SFO). Models are trained on all frequencies (Normal) or on a single excitation frequency at 9, 36, or 81 kHz, and evaluated on irregular and regular grids.

| Model | Irregular Data | | | | | | | | Regular Data | | | | | | | |
|---|---|---|---|---|---|---|---|---|---|---|---|---|---|---|---|---|
| | Normal | | SFO-9kHz | | SFO-36kHz | | SFO-81kHz | | Normal | | SFO-9kHz | | SFO-36kHz | | SFO-81kHz | |
| | MSE | RMSE | MSE | RMSE | MSE | RMSE | MSE | RMSE | MSE | RMSE | MSE | RMSE | MSE | RMSE | MSE | RMSE |
| MLP | 0.185 | 0.424 | 0.189 | 0.425 | 0.136 | 0.348 | 0.166 | 0.382 | 0.120 | 0.344 | 0.132 | 0.360 | 0.120 | 0.339 | 0.128 | 0.349 |
| FNO | 0.041 | 0.169 | 0.117 | 0.323 | 0.103 | 0.271 | 0.132 | 0.303 | 0.007 | 0.066 | 0.094 | 0.290 | 0.069 | 0.226 | 0.095 | 0.270 |
| GeoFNO | 0.377 | 0.611 | 0.400 | 0.629 | 0.412 | 0.637 | 0.414 | 0.639 | 0.231 | 0.474 | 0.702 | 0.681 | / | / | / | / |
| FFNO | 0.155 | 0.387 | 0.196 | 0.437 | 0.185 | 0.422 | 0.219 | 0.456 | 0.023 | 0.142 | 0.098 | 0.298 | 0.077 | 0.241 | 0.114 | 0.299 |
| FCNO | 0.324 | 0.568 | 0.374 | 0.610 | 0.364 | 0.602 | 0.385 | 0.619 | 0.026 | 0.151 | 0.098 | 0.299 | 0.075 | 0.241 | 0.104 | 0.290 |
| LNO | 0.110 | 0.312 | 0.142 | 0.354 | 0.135 | 0.329 | 0.176 | 0.371 | 0.045 | 0.201 | 0.202 | 0.437 | 0.163 | 0.346 | 0.185 | 0.386 |
| Transolver | 0.017 | 0.106 | 0.115 | 0.321 | 0.098 | 0.270 | 0.133 | 0.310 | 0.010 | 0.088 | 0.102 | 0.303 | 0.073 | 0.230 | 0.102 | 0.280 |

Table 42: Q2T forward modeling performance (MSE / RMSE) for the Type II multi-layer defect under single-frequency OOD (SFO). Models are trained on all frequencies (Normal) or on a single excitation frequency at 9, 36, or 81 kHz, and evaluated on irregular and regular grids.

| Model | Irregular Data | | | | | | | | Regular Data | | | | | | | |
| | Normal | | SFO-9kHz | | SFO-36kHz | | SFO-81kHz | | Normal | | SFO-9kHz | | SFO-36kHz | | SFO-81kHz | |
| | MSE | RMSE | MSE | RMSE | MSE | RMSE | MSE | RMSE | MSE | RMSE | MSE | RMSE | MSE | RMSE | MSE | RMSE |
|---|---|---|---|---|---|---|---|---|---|---|---|---|---|---|---|---|
| MLP | 0.175 | 0.414 | 0.206 | 0.435 | 0.162 | 0.398 | 0.186 | 0.412 | 0.088 | 0.291 | 0.094 | 0.285 | 0.090 | 0.294 | 0.103 | 0.311 |
| FNO | 0.096 | 0.285 | 0.140 | 0.350 | 0.083 | 0.260 | 0.139 | 0.314 | 0.002 | 0.041 | 0.091 | 0.274 | 0.039 | 0.179 | 0.061 | 0.212 |
| GeoFNO | 0.389 | 0.619 | 0.570 | 0.733 | 0.472 | 0.675 | 0.509 | 0.701 | 0.215 | 0.447 | / | / | 0.681 | 0.684 | / | / |
| FFNO | 0.170 | 0.410 | 0.221 | 0.459 | 0.172 | 0.411 | 0.222 | 0.461 | 0.005 | 0.068 | 0.080 | 0.257 | 0.042 | 0.185 | 0.069 | 0.226 |
| FCNO | 0.370 | 0.605 | 0.403 | 0.630 | 0.372 | 0.607 | 0.390 | 0.622 | 0.007 | 0.079 | 0.068 | 0.239 | 0.042 | 0.186 | 0.072 | 0.232 |
| LNO | 0.083 | 0.262 | 0.223 | 0.429 | 0.213 | 0.417 | 0.204 | 0.418 | 0.126 | 0.350 | 0.212 | 0.434 | 0.183 | 0.419 | 0.308 | 0.523 |
| Transolver | 0.008 | 0.076 | 0.148 | 0.359 | 0.087 | 0.266 | 0.136 | 0.309 | 0.002 | 0.044 | 0.098 | 0.281 | 0.046 | 0.192 | 0.078 | 0.234 |

Table 43: Full metric breakdown of T2T Temporal Evolution Prediction under geometric OOD (fsplit). Models are trained on 5 defect types and tested on the held-out Type II multi-layer defect, evaluated on irregular and regular grids. Reported metrics include MSE, SSIM, RMSE, normalized RMSE (nRMSE), conserved value RMSE (cRMSE), maximum error (Max), and boundary RMSE (bRMSE).

| Model | Irregular Data | | | | | | | Regular Data | | | | | | |
| | MSE | SSIM | RMSE | nRMSE | cRMSE | Max | bRMSE | MSE | SSIM | RMSE | nRMSE | cRMSE | Max | bRMSE |
|---|---|---|---|---|---|---|---|---|---|---|---|---|---|---|
| MLP | 0.000044 | 0.999024 | 0.006512 | 0.006512 | 0.000619 | 0.089084 | 0.004014 | 0.000034 | 0.999024 | 0.006512 | 0.006512 | 0.000619 | 0.089084 | 0.004014 |
| FNO | 0.000074 | 0.998220 | 0.008446 | 0.008446 | 0.000255 | 0.084119 | 0.005494 | 0.000055 | 0.998220 | 0.008446 | 0.008446 | 0.000255 | 0.084119 | 0.005494 |
| GeoFNO | 0.009401 | 0.923601 | 0.096582 | 0.096582 | 0.012251 | 1.260375 | 0.056853 | 0.000542 | 0.923601 | 0.096582 | 0.096582 | 0.012251 | 1.260375 | 0.056853 |
| FFNO | 0.000397 | 0.989956 | 0.019853 | 0.019853 | 0.000353 | 0.103523 | 0.012143 | 0.000026 | 0.989956 | 0.019853 | 0.019853 | 0.000353 | 0.103523 | 0.012143 |
| FCNO | 0.015993 | 0.856374 | 0.126231 | 0.126231 | 0.001218 | 1.064372 | 0.089791 | 0.000076 | 0.856374 | 0.126231 | 0.126231 | 0.001218 | 1.064372 | 0.089791 |
| LNO | 0.021931 | 0.905011 | 0.145878 | 0.145878 | 0.013580 | 3.321057 | 0.066739 | 0.026910 | 0.905011 | 0.145878 | 0.145878 | 0.013580 | 3.321057 | 0.066739 |
| Transolver | 0.000003 | 0.999945 | 0.001646 | 0.001646 | 0.000193 | 0.027765 | 0.001242 | 0.000003 | 0.999945 | 0.001646 | 0.001646 | 0.000193 | 0.027765 | 0.001242 |

Table 44: T2T prediction performance (MSE / RMSE, scaled by $10^{-3}$) for the single layer defect configuration under normal training and three frequency-band OOD regimes (High-OOD, Mid-OOD, Low-OOD), on irregular and regular grids.

| Model | Irregular Data | | | | | | | | Regular Data | | | | | | | |
| | Normal | | High-OOD | | Mid-OOD | | Low-OOD | | Normal | | High-OOD | | Mid-OOD | | Low-OOD | |
| | MSE | RMSE | MSE | RMSE | MSE | RMSE | MSE | RMSE | MSE | RMSE | MSE | RMSE | MSE | RMSE | MSE | RMSE |
|---|---|---|---|---|---|---|---|---|---|---|---|---|---|---|---|---|
| MLP | 0.059 | 7.489 | 0.057 | 7.389 | 0.057 | 7.365 | 0.060 | 7.535 | 0.057 | 7.263 | 0.063 | 7.653 | 0.057 | 7.302 | 0.057 | 7.253 |
| FNO | 0.018 | 4.103 | 0.016 | 3.902 | 0.016 | 3.910 | 0.017 | 4.060 | 0.012 | 3.413 | 0.012 | 3.389 | 0.012 | 3.421 | 0.012 | 3.412 |
| GeoFNO | 2.236 | 46.393 | 52.685 | 92.078 | 5.628 | 72.783 | 39.912 | 101.928 | 1.613 | 39.508 | 1.335 | 36.016 | 0.560 | 23.254 | 0.889 | 29.433 |
| FFNO | 0.075 | 8.552 | 0.070 | 8.260 | 0.074 | 8.491 | 0.075 | 8.561 | 0.016 | 3.931 | 0.016 | 3.883 | 0.016 | 3.927 | 0.016 | 3.979 |
| FCNO | 1.587 | 39.261 | 1.504 | 38.283 | 1.594 | 39.391 | 1.605 | 39.593 | 0.051 | 6.970 | 0.051 | 6.918 | 0.050 | 6.917 | 0.055 | 7.155 |
| LNO | 22.794 | 147.080 | 14.688 | 119.185 | 19.192 | 135.844 | 17.901 | 130.424 | 10.626 | 100.108 | 9.714 | 96.903 | 9.452 | 95.009 | 9.623 | 96.516 |
| Transolver | 0.004 | 1.685 | 0.007 | 1.712 | 0.003 | 1.533 | 0.003 | 1.580 | 0.024 | 2.397 | 0.009 | 1.929 | 0.040 | 2.787 | 0.087 | 3.108 |

Table 45: T2T prediction performance (MSE / RMSE, scaled by $10^{-3}$) for the Type I double-layer defect under normal training and three frequency-band OOD regimes (High-OOD, Mid-OOD, Low-OOD), on irregular and regular grids.

| Model | Irregular Data | | | | | | | | Regular Data | | | | | | | |
| | Normal | | High-OOD | | Mid-OOD | | Low-OOD | | Normal | | High-OOD | | Mid-OOD | | Low-OOD | |
| | MSE | RMSE | MSE | RMSE | MSE | RMSE | MSE | RMSE | MSE | RMSE | MSE | RMSE | MSE | RMSE | MSE | RMSE |
|---|---|---|---|---|---|---|---|---|---|---|---|---|---|---|---|---|
| MLP | 0.069 | 8.074 | 0.069 | 8.082 | 0.067 | 7.963 | 0.067 | 7.971 | 0.036 | 5.799 | 0.037 | 5.859 | 0.035 | 5.701 | 0.037 | 5.910 |
| FNO | 0.016 | 3.888 | 0.015 | 3.814 | 0.015 | 3.819 | 0.015 | 3.799 | 0.015 | 3.673 | 0.016 | 3.895 | 0.015 | 3.716 | 0.016 | 3.835 |
| GeoFNO | 8.852 | 92.819 | 4.674 | 67.206 | 0.335 | 18.150 | 136.392 | 124.075 | 1.065 | 32.386 | 1.336 | 36.338 | 1.165 | 33.917 | 1.416 | 37.141 |
| FFNO | 0.093 | 9.474 | 0.095 | 9.605 | 0.090 | 9.325 | 0.091 | 9.353 | 0.019 | 4.205 | 0.021 | 4.403 | 0.019 | 4.200 | 0.021 | 4.370 |
| FCNO | 2.045 | 43.386 | 2.004 | 43.388 | 1.984 | 42.771 | 1.957 | 42.429 | 0.063 | 7.513 | 0.070 | 7.902 | 0.061 | 7.507 | 0.071 | 7.908 |
| LNO | 28.708 | 168.389 | 30.772 | 173.500 | 26.066 | 160.414 | 29.946 | 171.263 | 13.763 | 115.071 | 14.637 | 118.948 | 16.122 | 124.835 | 14.488 | 117.513 |
| Transolver | 0.002 | 1.523 | 0.004 | 1.674 | 0.002 | 1.521 | 0.007 | 1.854 | 0.004 | 1.785 | 0.044 | 2.696 | 0.037 | 2.381 | 0.020 | 2.513 |

Table 46: T2T prediction performance (MSE / RMSE, scaled by $10^{-3}$) for the Type II double-layer defect under normal training and three frequency-band OOD regimes (High-OOD, Mid-OOD, Low-OOD), on irregular and regular grids.

| Model | Irregular Data | | | | | | | | Regular Data | | | | | | | |
| | Normal | | High-OOD | | Mid-OOD | | Low-OOD | | Normal | | High-OOD | | Mid-OOD | | Low-OOD | |
| | MSE | RMSE | MSE | RMSE | MSE | RMSE | MSE | RMSE | MSE | RMSE | MSE | RMSE | MSE | RMSE | MSE | RMSE |
|---|---|---|---|---|---|---|---|---|---|---|---|---|---|---|---|---|
| MLP | 0.074 | 8.350 | 0.075 | 8.409 | 0.076 | 8.508 | 0.077 | 8.537 | 0.077 | 8.427 | 0.085 | 8.839 | 0.078 | 8.465 | 0.075 | 8.355 |
| FNO | 0.013 | 3.405 | 0.013 | 3.427 | 0.014 | 3.572 | 0.013 | 3.498 | 0.014 | 3.641 | 0.014 | 3.690 | 0.014 | 3.604 | 0.013 | 3.563 |
| GeoFNO | 0.528 | 22.728 | 0.833 | 28.569 | 0.676 | 25.753 | 4.811 | 67.175 | 0.598 | 23.963 | 0.858 | 28.832 | 0.396 | 19.528 | 0.769 | 27.361 |
| FFNO | 0.255 | 15.887 | 0.257 | 15.948 | 0.258 | 15.981 | 0.258 | 16.001 | 0.023 | 4.666 | 0.022 | 4.614 | 0.022 | 4.557 | 0.022 | 4.543 |
| FCNO | 10.821 | 103.653 | 10.783 | 103.447 | 10.860 | 103.783 | 10.983 | 104.432 | 0.083 | 8.710 | 0.081 | 8.709 | 0.077 | 8.462 | 0.077 | 8.456 |
| LNO | 39.300 | 190.294 | 32.585 | 172.867 | 46.420 | 206.818 | 39.659 | 191.092 | 12.252 | 107.545 | 14.793 | 118.676 | 12.977 | 110.083 | 11.222 | 102.261 |
| Transolver | 0.007 | 1.926 | 0.007 | 1.810 | 0.020 | 2.062 | 0.068 | 2.794 | 0.039 | 2.995 | 0.078 | 3.278 | 0.037 | 2.739 | 0.010 | 2.189 |

Table 47: T2T prediction performance (MSE / RMSE, scaled by $10^{-3}$) for the Type III double-layer defect under normal training and three frequency-band OOD regimes (High-OOD, Mid-OOD, Low-OOD), on irregular and regular grids.

| Model | Irregular Data | | | | | | | | Regular Data | | | | | | | |
| | Normal | | High-OOD | | Mid-OOD | | Low-OOD | | Normal | | High-OOD | | Mid-OOD | | Low-OOD | |
| | MSE | RMSE | MSE | RMSE | MSE | RMSE | MSE | RMSE | MSE | RMSE | MSE | RMSE | MSE | RMSE | MSE | RMSE |
|---|---|---|---|---|---|---|---|---|---|---|---|---|---|---|---|---|
| MLP | 0.052 | 7.001 | 0.055 | 7.147 | 0.059 | 7.503 | 0.054 | 7.160 | 0.038 | 5.961 | 0.037 | 5.929 | 0.041 | 6.181 | 0.039 | 6.056 |
| FNO | 0.016 | 3.722 | 0.016 | 3.735 | 0.024 | 4.730 | 0.022 | 4.547 | 0.018 | 4.094 | 0.019 | 4.231 | 0.023 | 4.626 | 0.020 | 4.306 |
| GeoFNO | 4.525 | 64.772 | 0.484 | 21.697 | 2.329 | 47.365 | 0.762 | 27.172 | 1.368 | 36.625 | 2.879 | 53.385 | 0.964 | 30.800 | 0.736 | 27.001 |
| FFNO | 0.245 | 15.550 | 0.250 | 15.701 | 0.271 | 16.368 | 0.253 | 15.821 | 0.024 | 4.648 | 0.022 | 4.459 | 0.028 | 4.942 | 0.023 | 4.568 |
| FCNO | 10.958 | 104.158 | 11.391 | 106.314 | 11.844 | 108.412 | 11.375 | 106.343 | 0.079 | 8.278 | 0.071 | 7.937 | 0.094 | 8.893 | 0.076 | 8.166 |
| LNO | 18.736 | 136.043 | 18.521 | 135.321 | 19.424 | 138.497 | 21.471 | 145.679 | 12.772 | 112.091 | 13.693 | 116.262 | 13.016 | 113.155 | 12.311 | 110.236 |
| Transolver | 0.015 | 1.883 | 0.006 | 1.843 | 0.006 | 1.903 | 0.006 | 1.830 | 0.013 | 2.045 | 0.012 | 2.235 | 0.006 | 1.935 | 0.004 | 1.775 |

Table 48: T2T prediction performance (MSE / RMSE, scaled by $10^{-3}$) for the Type I multi-layer defect under normal training and three frequency-band OOD regimes (High-OOD, Mid-OOD, Low-OOD), on irregular and regular grids.

| | Irregular Data | | | | | | | | Regular Data | | | | | | | |
|---|---|---|---|---|---|---|---|---|---|---|---|---|---|---|---|---|
| Model | Normal | | High-OOD | | Mid-OOD | | Low-OOD | | Normal | | High-OOD | | Mid-OOD | | Low-OOD | |
| | MSE | RMSE | MSE | RMSE | MSE | RMSE | MSE | RMSE | MSE | RMSE | MSE | RMSE | MSE | RMSE | MSE | RMSE |
| MLP | 0.045 | 6.544 | 0.048 | 6.764 | 0.046 | 6.599 | 0.046 | 6.586 | 0.023 | 4.576 | 0.022 | 4.551 | 0.022 | 4.499 | 0.022 | 4.525 |
| FNO | 0.012 | 3.240 | 0.012 | 3.296 | 0.012 | 3.277 | 0.012 | 3.274 | 0.016 | 3.842 | 0.014 | 3.685 | 0.016 | 3.844 | 0.014 | 3.660 |
| GeoFNO | 1.856 | 42.443 | 0.415 | 19.864 | 3.847 | 60.652 | 0.912 | 27.928 | 2.732 | 51.708 | 2.160 | 46.187 | 1.798 | 42.013 | 1.050 | 32.170 |
| FFNO | 0.258 | 16.016 | 0.261 | 16.108 | 0.262 | 16.135 | 0.262 | 16.133 | 0.024 | 4.755 | 0.022 | 4.576 | 0.023 | 4.674 | 0.022 | 4.595 |
| FCNO | 10.442 | 102.008 | 10.603 | 102.776 | 10.501 | 102.289 | 10.477 | 102.179 | 0.087 | 8.942 | 0.078 | 8.556 | 0.080 | 8.733 | 0.077 | 8.549 |
| LNO | 32.133 | 172.432 | 37.828 | 187.519 | 32.373 | 173.466 | 31.562 | 171.854 | 41.418 | 192.396 | 38.828 | 186.108 | 46.712 | 203.026 | 45.767 | 200.937 |
| Transolver | 0.006 | 1.737 | 0.003 | 1.605 | 0.004 | 1.724 | 0.004 | 1.675 | 0.005 | 1.832 | 0.006 | 1.824 | 0.058 | 2.857 | 0.016 | 2.337 |

Table 49: T2T prediction performance (MSE / RMSE, scaled by $10^{-3}$) for the Type II multi-layer defect under normal training and three frequency-band OOD regimes (High-OOD, Mid-OOD, Low-OOD), on irregular and regular grids.

| | Irregular Data | | | | | | | | Regular Data | | | | | | | |
|---|---|---|---|---|---|---|---|---|---|---|---|---|---|---|---|---|
| Model | Normal | | High-OOD | | Mid-OOD | | Low-OOD | | Normal | | High-OOD | | Mid-OOD | | Low-OOD | |
| | MSE | RMSE | MSE | RMSE | MSE | RMSE | MSE | RMSE | MSE | RMSE | MSE | RMSE | MSE | RMSE | MSE | RMSE |
| MLP | 0.057 | 7.367 | 0.057 | 7.400 | 0.058 | 7.422 | 0.056 | 7.305 | 0.043 | 6.376 | 0.045 | 6.522 | 0.024 | 4.760 | 0.044 | 6.457 |
| FNO | 0.018 | 4.113 | 0.017 | 3.970 | 0.017 | 4.002 | 0.017 | 4.038 | 0.013 | 3.498 | 0.013 | 3.587 | 0.014 | 3.709 | 0.013 | 3.566 |
| GeoFNO | 0.407 | 19.879 | 0.273 | 16.305 | 0.284 | 16.678 | 1.214 | 34.445 | 3.275 | 56.772 | 1.003 | 31.278 | 1.187 | 33.847 | 0.455 | 21.126 |
| FFNO | 0.280 | 16.666 | 0.287 | 16.883 | 0.281 | 16.703 | 0.276 | 16.561 | 0.013 | 3.494 | 0.012 | 3.452 | 0.013 | 3.541 | 0.012 | 3.343 |
| FCNO | 12.146 | 110.047 | 12.274 | 110.609 | 12.411 | 111.241 | 12.233 | 110.387 | 0.046 | 6.535 | 0.046 | 6.546 | 0.048 | 6.660 | 0.041 | 6.243 |
| LNO | 15.354 | 120.324 | 14.468 | 116.739 | 13.242 | 113.007 | 14.390 | 117.038 | 8.353 | 88.102 | 5.651 | 72.145 | 6.741 | 76.967 | 5.604 | 73.402 |
| Transolver | 0.006 | 1.767 | 0.006 | 1.790 | 0.003 | 1.561 | 0.003 | 1.513 | 0.004 | 1.741 | 0.010 | 1.865 | 0.015 | 2.178 | 0.009 | 1.985 |

Table 50: T2T prediction performance (MSE / RMSE, scaled by $10^{-3}$) for the Type I double-layer defect under single-frequency OOD (SFO). Models are trained on all frequencies (Normal) or on a single excitation frequency at 9, 36, or 81 kHz, and evaluated on irregular and regular grids.

| | Irregular Data | | | | | | | | Regular Data | | | | | | | |
|---|---|---|---|---|---|---|---|---|---|---|---|---|---|---|---|---|
| Model | Normal | | SFO-9kHz | | SFO-36kHz | | SFO-81kHz | | Normal | | SFO-9kHz | | SFO-36kHz | | SFO-81kHz | |
| | MSE | RMSE | MSE | RMSE | MSE | RMSE | MSE | RMSE | MSE | RMSE | MSE | RMSE | MSE | RMSE | MSE | RMSE |
| MLP | 0.069 | 8.074 | 0.099 | 9.635 | 0.114 | 10.260 | 0.180 | 12.362 | 0.036 | 5.799 | 0.069 | 7.997 | 0.078 | 8.393 | 0.102 | 9.332 |
| FNO | 0.016 | 3.888 | 0.777 | 25.365 | 0.542 | 18.687 | 0.881 | 23.594 | 0.015 | 3.673 | 1.615 | 35.575 | 1.058 | 24.189 | 1.383 | 26.221 |
| GeoFNO | 8.852 | 92.819 | 12.866 | 106.858 | 5.352 | 52.368 | 152.992 | 246.938 | 1.065 | 32.386 | 19.421 | 99.816 | 9.076 | 68.788 | 4.226 | 55.041 |
| FFNO | 0.093 | 9.474 | 0.825 | 27.201 | 0.669 | 22.140 | 1.021 | 26.806 | 0.019 | 4.205 | 1.365 | 32.851 | 0.703 | 19.443 | 0.870 | 20.886 |
| FCNO | 2.045 | 43.386 | 12.720 | 107.196 | 10.605 | 85.791 | 13.073 | 94.919 | 0.063 | 7.513 | 2.173 | 41.044 | 1.836 | 29.863 | 2.549 | 33.300 |
| LNO | 28.708 | 168.389 | 225.509 | 443.361 | 161.390 | 362.675 | 324.391 | 461.221 | 13.763 | 115.071 | 449.146 | 535.213 | 96.822 | 275.347 | 267.729 | 405.862 |
| Transolver | 0.002 | 1.523 | 0.499 | 10.611 | 0.221 | 5.755 | 0.064 | 6.352 | 0.004 | 1.785 | 6.388 | 47.871 | 0.238 | 8.083 | 0.071 | 5.761 |

Table 51: T2T prediction performance (MSE / RMSE, scaled by $10^{-3}$) for the Type III double-layer defect under single-frequency OOD (SFO). Models are trained on all frequencies (Normal) or on a single excitation frequency at 9, 36, or 81 kHz, and evaluated on irregular and regular grids.

| Model | Irregular Data | | | | | | | | Regular Data | | | | | | | |
|---|---|---|---|---|---|---|---|---|---|---|---|---|---|---|---|---|
| | Normal | | SFO-9kHz | | SFO-36kHz | | SFO-81kHz | | Normal | | SFO-9kHz | | SFO-36kHz | | SFO-81kHz | |
| | MSE | RMSE | MSE | RMSE | MSE | RMSE | MSE | RMSE | MSE | RMSE | MSE | RMSE | MSE | RMSE | MSE | RMSE |
| MLP | 0.052 | 7.001 | 0.096 | 9.261 | 0.089 | 9.001 | 0.096 | 9.256 | 0.038 | 5.961 | 0.054 | 6.930 | 0.050 | 6.549 | 0.058 | 7.006 |
| FNO | 0.016 | 3.722 | 0.069 | 7.401 | 0.024 | 4.793 | 0.024 | 4.740 | 0.018 | 4.094 | 1.215 | 31.000 | 0.856 | 20.269 | 1.028 | 21.012 |
| GeoFNO | 4.525 | 64.772 | 5.407 | 62.020 | 4.535 | 59.192 | 4.086 | 49.391 | 1.368 | 36.625 | / | / | 42.215 | 126.401 | 626.012 | 354.678 |
| FFNO | 0.245 | 15.550 | 0.478 | 21.286 | 0.380 | 19.051 | 0.409 | 19.582 | 0.024 | 4.648 | 0.984 | 27.519 | 0.881 | 20.621 | 0.992 | 21.481 |
| FCNO | 10.958 | 104.158 | 12.270 | 110.390 | 12.303 | 110.198 | 12.841 | 112.274 | 0.079 | 8.278 | 1.632 | 34.079 | 1.870 | 27.960 | 2.386 | 30.605 |
| LNO | 18.736 | 136.043 | / | / | 93.363 | 270.806 | 132.448 | 313.690 | 12.772 | 112.091 | 112.347 | 319.226 | 87.015 | 242.656 | 111.244 | 273.975 |
| Transolver | 0.015 | 1.883 | 12.117 | 72.850 | 0.551 | 12.382 | 0.014 | 3.161 | 0.013 | 2.045 | 41.022 | 150.911 | 3.514 | 29.458 | 0.028 | 3.694 |

Table 52: T2T prediction performance (MSE / RMSE, scaled by $10^{-3}$) for the Type II multi-layer defect under single-frequency OOD (SFO). Models are trained on all frequencies (Normal) or on a single excitation frequency at 9, 36, or 81 kHz, and evaluated on irregular and regular grids.

| Model | Irregular Data | | | | | | | | Regular Data | | | | | | | |
|---|---|---|---|---|---|---|---|---|---|---|---|---|---|---|---|---|
| | Normal | | SFO-9kHz | | SFO-36kHz | | SFO-81kHz | | Normal | | SFO-9kHz | | SFO-36kHz | | SFO-81kHz | |
| | MSE | RMSE | MSE | RMSE | MSE | RMSE | MSE | RMSE | MSE | RMSE | MSE | RMSE | MSE | RMSE | MSE | RMSE |
| MLP | 0.057 | 7.367 | 0.071 | 8.079 | 0.082 | 8.773 | 0.106 | 9.797 | 0.043 | 6.376 | 0.064 | 7.666 | 0.035 | 5.435 | 0.115 | 9.718 |
| FNO | 0.018 | 4.113 | 0.085 | 8.025 | 0.024 | 4.688 | 0.029 | 5.136 | 0.013 | 3.498 | 3.102 | 51.891 | 1.636 | 30.916 | 2.910 | 39.682 |
| GeoFNO | 0.407 | 19.879 | 8.876 | 82.315 | 2.241 | 43.227 | 2.509 | 45.897 | 3.275 | 56.772 | / | / | 78.469 | 169.930 | 15.630 | 98.539 |
| FFNO | 0.280 | 16.666 | 0.447 | 20.521 | 0.336 | 18.212 | 0.427 | 20.331 | 0.013 | 3.494 | 1.799 | 40.511 | 1.097 | 25.059 | 1.371 | 26.948 |
| FCNO | 12.146 | 110.047 | 13.511 | 115.886 | 12.561 | 111.858 | 14.269 | 118.904 | 0.046 | 6.535 | 2.383 | 46.862 | 2.221 | 34.881 | 4.122 | 45.655 |
| LNO | 15.354 | 120.324 | 205.032 | 391.724 | 109.808 | 307.897 | 154.712 | 336.819 | 8.353 | 88.102 | 197.387 | 418.471 | 398.395 | 457.331 | 124.080 | 289.876 |
| Transolver | 0.006 | 1.767 | 7.872 | 65.485 | 0.079 | 3.888 | 0.007 | 2.386 | 0.004 | 1.741 | 70.803 | 217.836 | 0.113 | 6.531 | 0.024 | 3.713 |

# K  MORE VISUALIZATIONS

The following presents visualisations of all experimental results across all tasks, including Forward Modeling, Inverse Source Reconstruction, Temporal Evolution Prediction and Surface-to-Source Reconstruction, involving regular/irregular grids and full-frequency/OOD environments. We selected OOD-high (high-frequency hold-out) regime for visualisation. We have selected experimental results from FNO, Transolver, GeoFNO, and LNO models for display, with visualisations provided for inputs and outputs under each configuration.

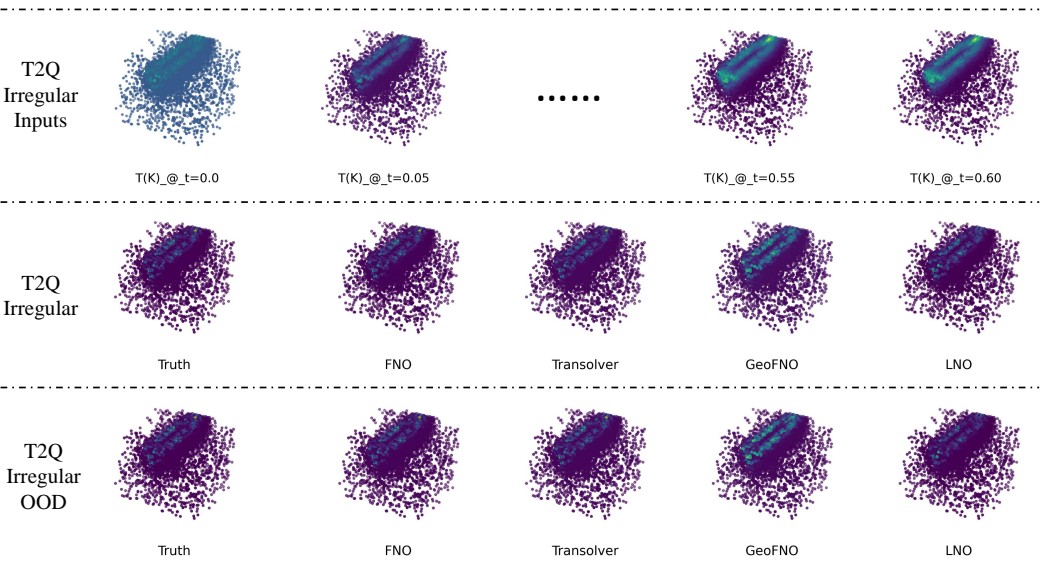

Figure 8: Visualization of experimental results under the temperature to heat (T2Q) inverse task on irregular grids, the experiment was trained and evaluated on the Type I double-layer subset of the Aletheia dataset.

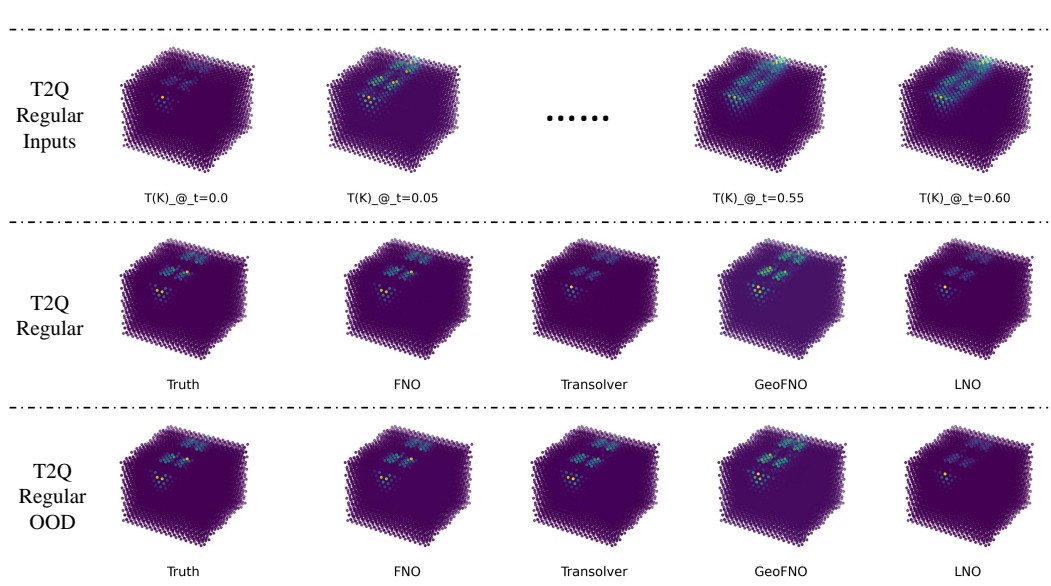

Figure 9: Visualization of experimental results under the temperature to heat (T2Q) inverse task on regular grids, the experiment was trained and evaluated on the Type I double-layer subset of the Aletheia dataset.

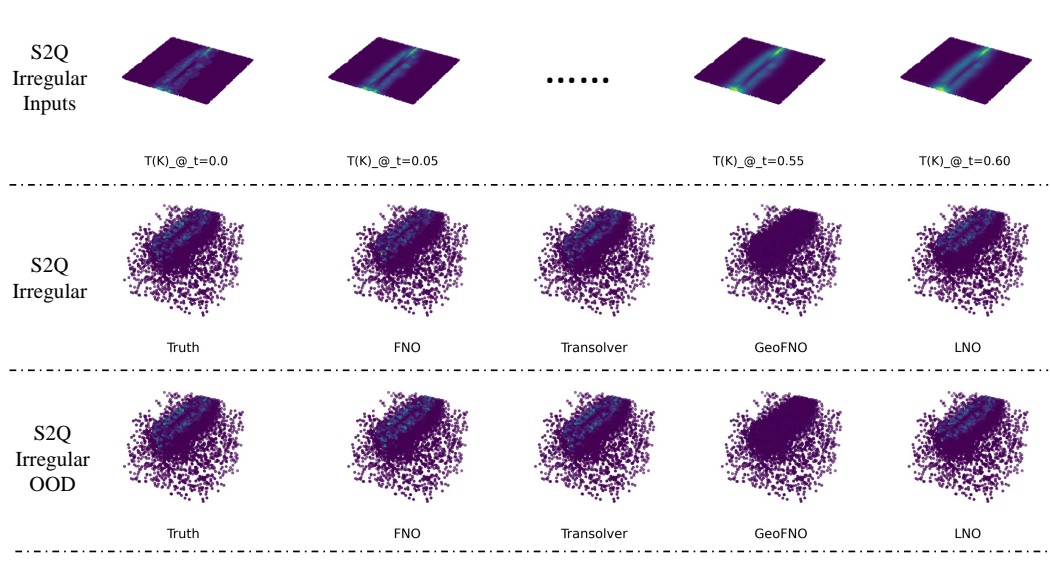

Figure 10: Visualization of experimental results under the surface temperature to heat (S2Q) inverse task on irregular grids, the experiment was trained and evaluated on the Type I double-layer subset of the Aletheia dataset.

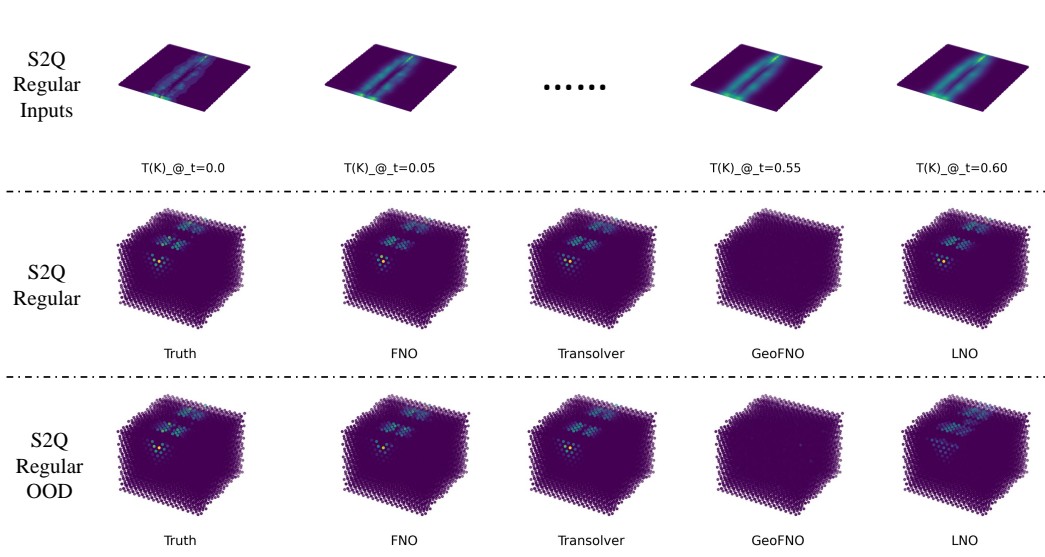

Figure 11: Visualization of experimental results under the surface temperature to heat (S2Q) inverse task on regular grids, the experiment was trained and evaluated on the Type I double-layer subset of the Aletheia dataset.

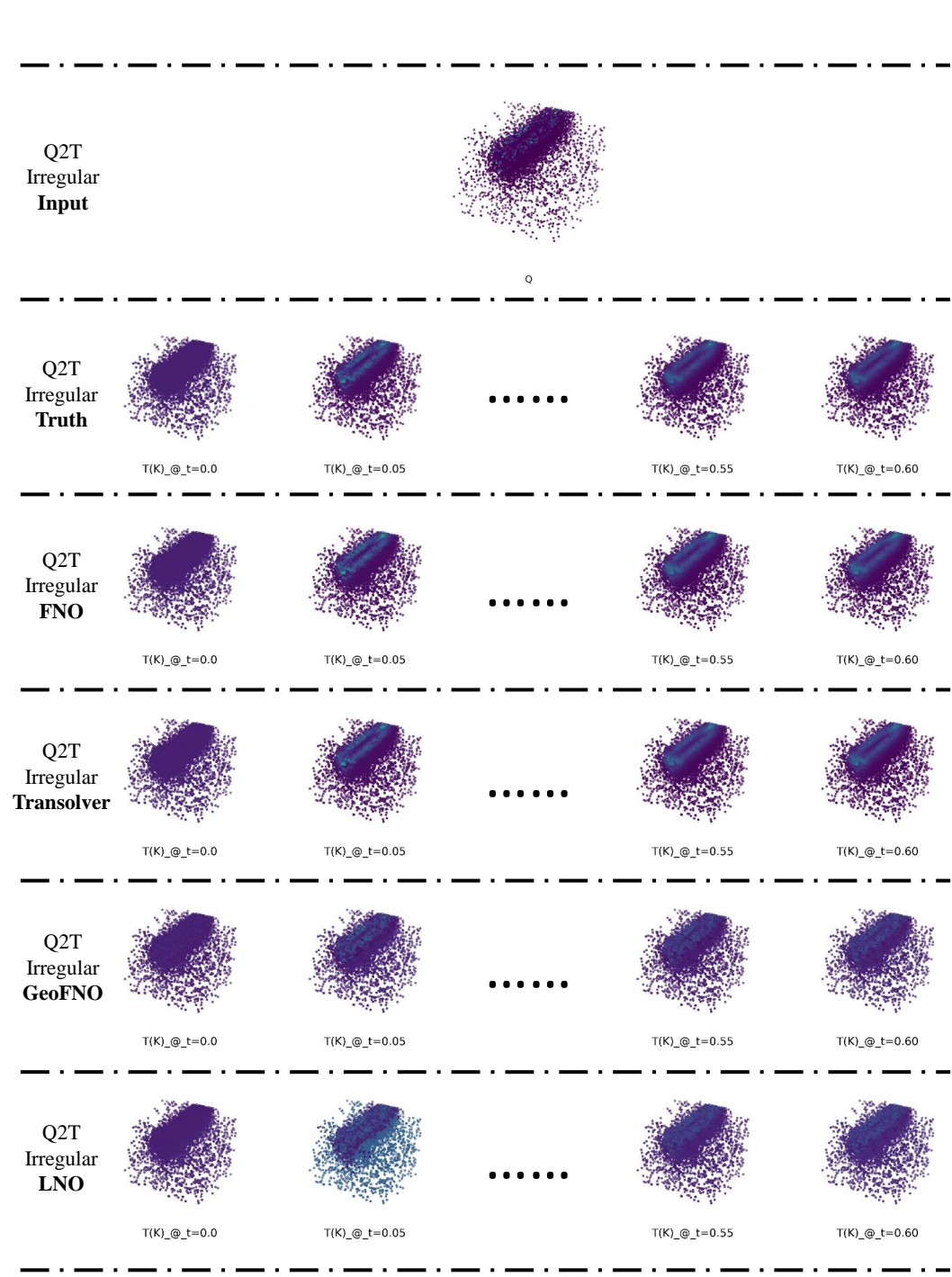

Figure 12: Visualization of experimental results under the heat-to-temperature (Q2T) forward task on irregular grids under full-frequency setting, the experiment was trained and evaluated on the Type I double-layer subset of the Aletheia dataset.

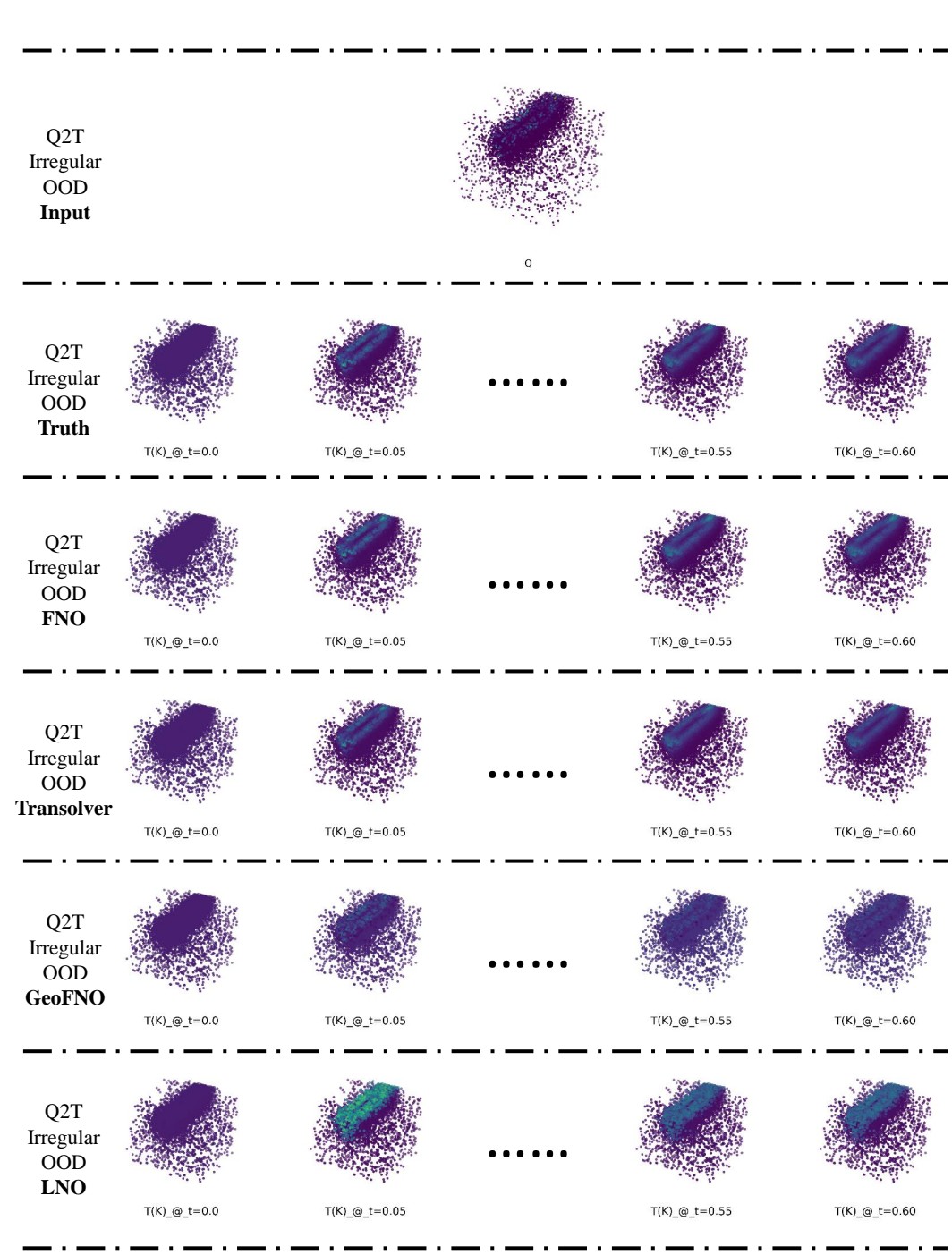

Figure 13: Visualization of experimental results under the heat-to-temperature (Q2T) forward task on irregular grids under OOD setting, the experiment was trained and evaluated on the Type I double-layer subset of the Aletheia dataset.

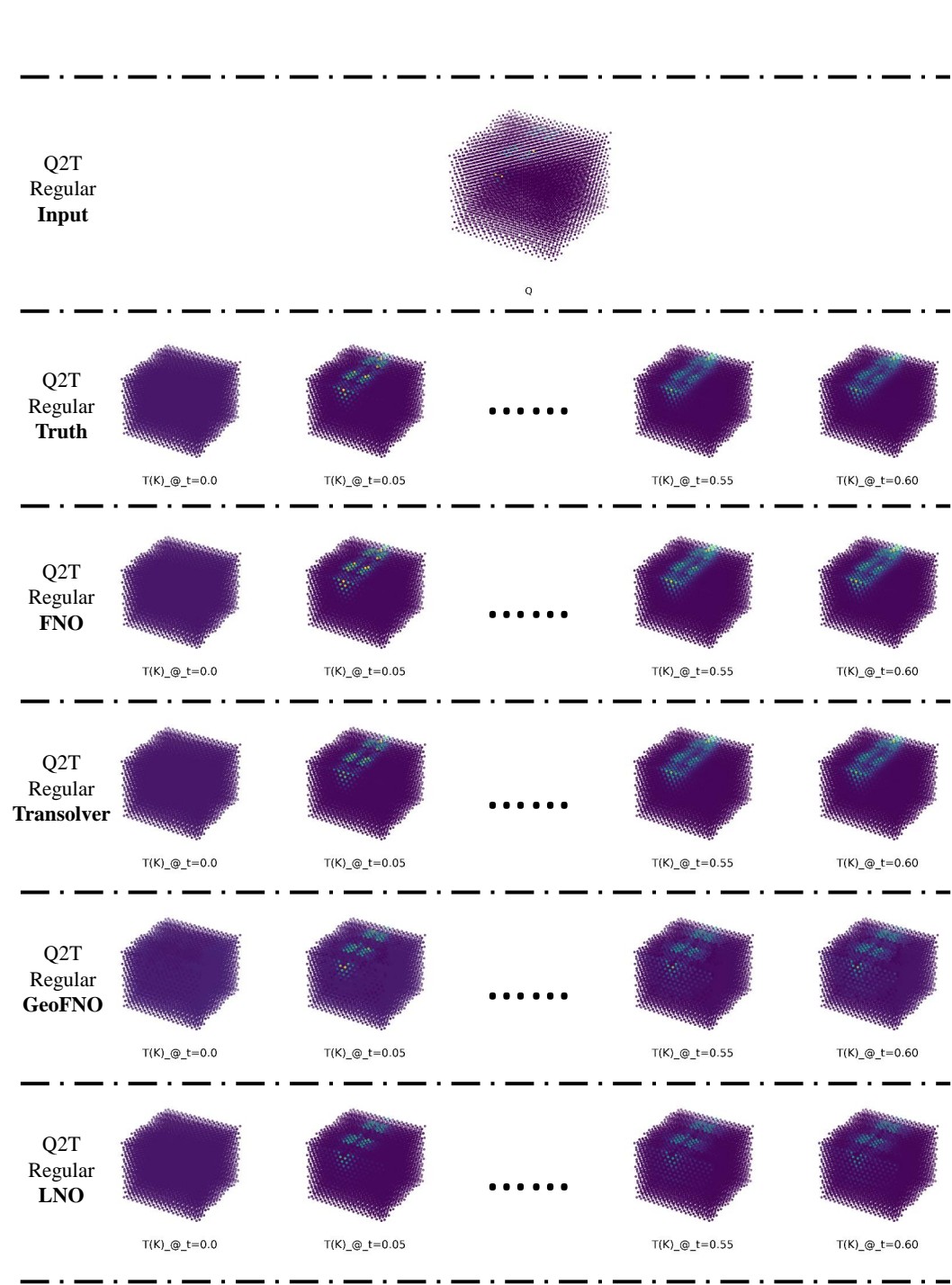

Figure 14: Visualization of experimental results under the heat-to-temperature (Q2T) forward task on regular grids under full-frequency setting, the experiment was trained and evaluated on the Type I double-layer subset of the Aletheia dataset.

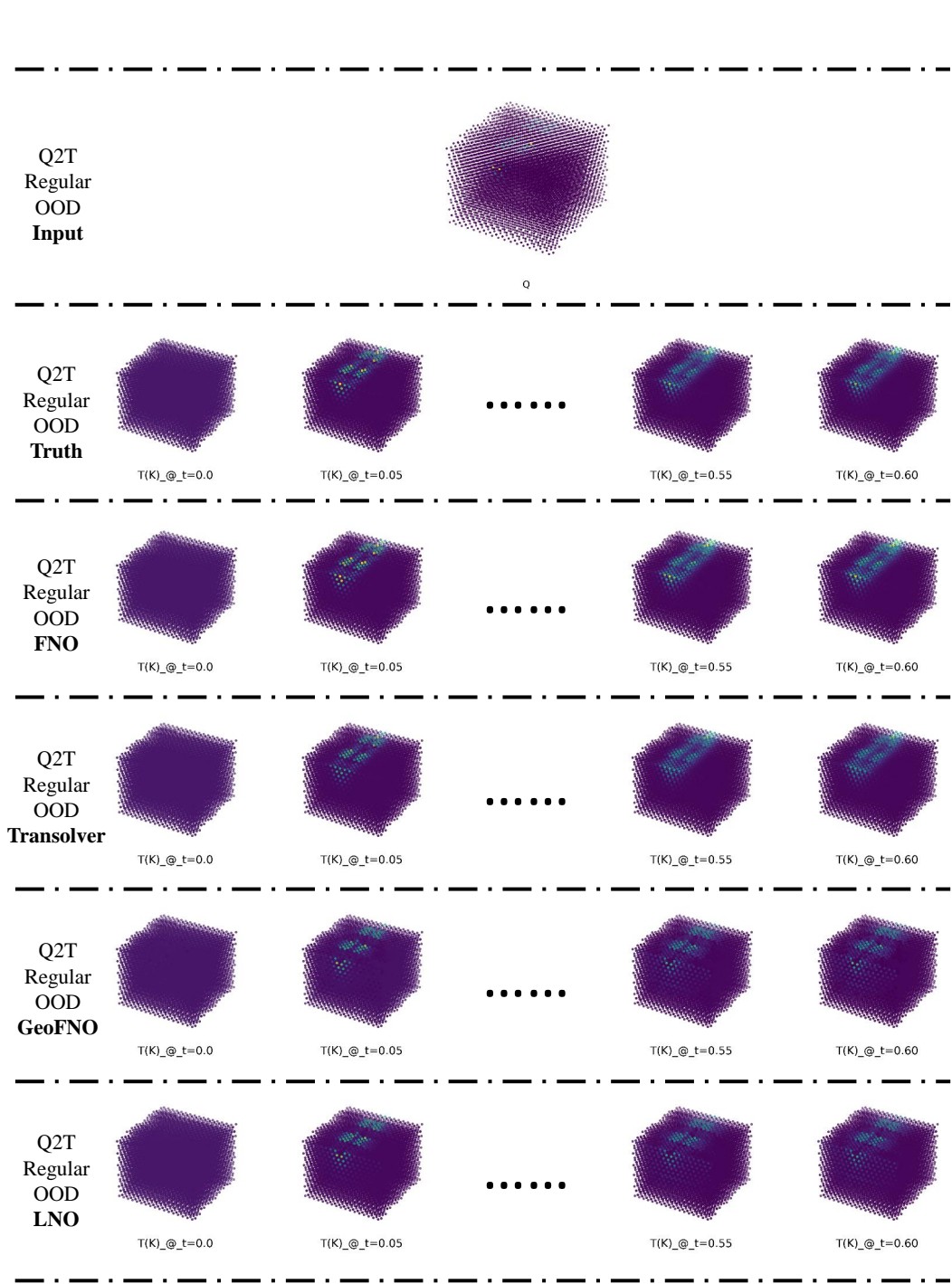

Figure 15: Visualization of experimental results under the heat-to-temperature (Q2T) forward task on regular grids under OOD setting, the experiment was trained and evaluated on the Type I double-layer subset of the Aletheia dataset.

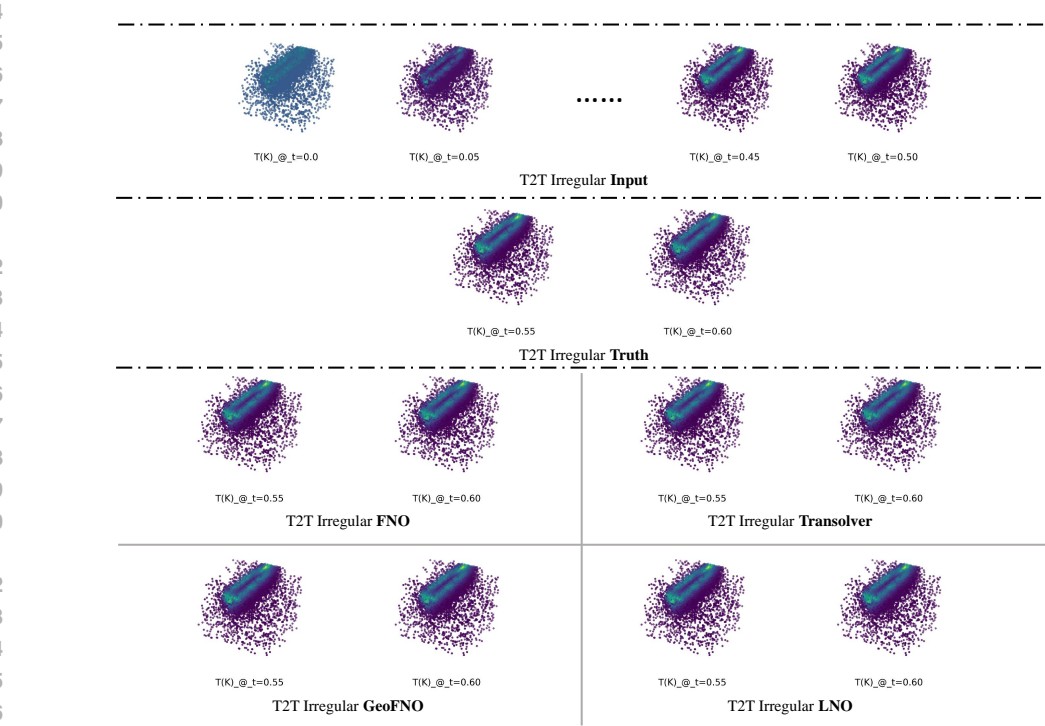

Figure 16: Visualization of experimental results under the temperature-to-temperature (T2T) prediction task on irregular grids under full-frequency setting, the experiment was trained and evaluated on the Type I double-layer subset of the Aletheia dataset.

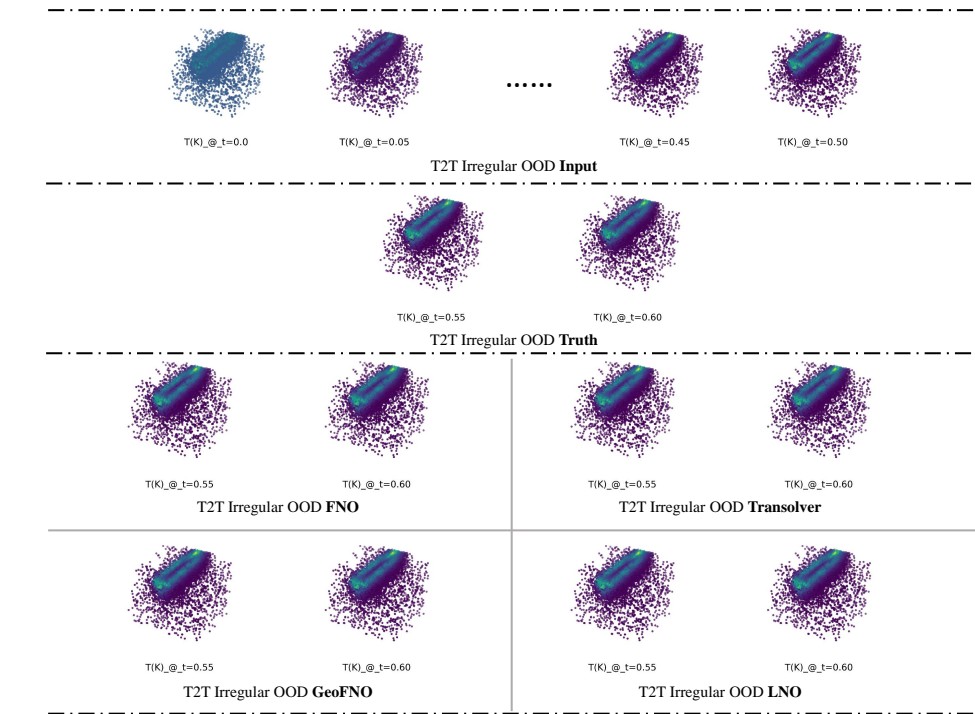

Figure 17: Visualization of experimental results under the temperature-to-temperature (T2T) prediction task on irregular grids under OOD setting, the experiment was trained and evaluated on the Type I double-layer subset of the Aletheia dataset.

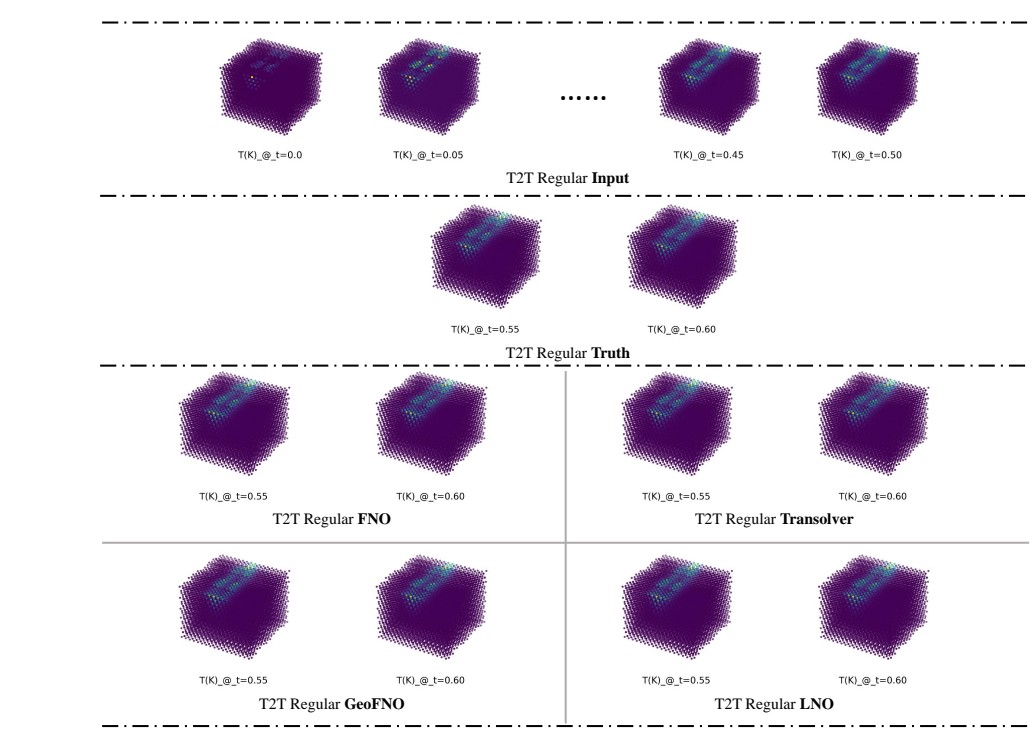

Figure 18: Visualization of experimental results under the temperature-to-temperature (T2T) prediction task on regular grids under full-frequency setting, the experiment was trained and evaluated on the Type I double-layer subset of the Aletheia dataset.

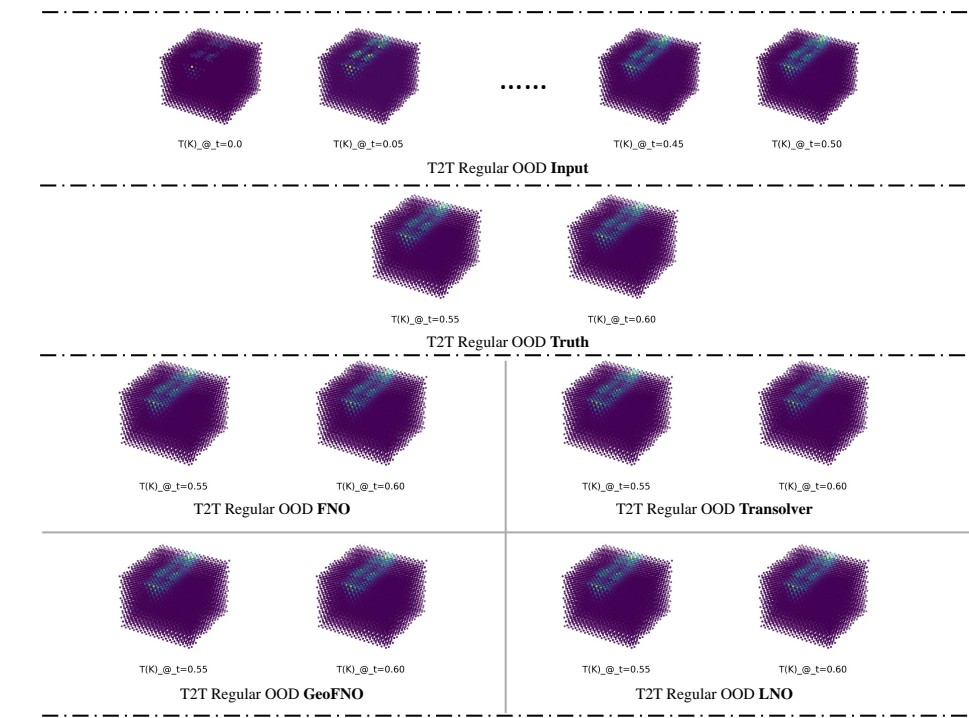

Figure 19: Visualization of experimental results under the temperature-to-temperature (T2T) prediction task on regular grids under OOD setting, the experiment was trained and evaluated on the Type I double-layer subset of the Aletheia dataset.

