# OpenReview forum: "ALETHEIA: A Multi-Frequency Eddy Current Pulsed Thermography Dataset for Neural Operator Learning in Nondestructive Testing"
_ICLR.cc/2026/Conference — Submitted to ICLR 2026_

### Official Review · Reviewer_LYNR · 2025-10-28

**Soundness:** 3
**Presentation:** 3
**Contribution:** 3
**Rating:** 6
**Confidence:** 4

**Summary:**

This work proposed a 3D benchmark, Aletheia. The dataset simulates eddy-current-induced heating, which is an electromagnetic-thermal coupling physics. The dataset is feature with comprehensive evaluations (see Table 1).

**Strengths:**

**Significance** The work sets a new benchmark for neural operators. The evaluation is comprehensive, and notably it includes out-of-distribution (OoD) evaluation, which is valuable and important for many scenarios of the application of neural operators.

**Weaknesses:**

1. For geometries and topology in Alethesia, are the types listed in Table 6 in Appendix C able to cover most cases? What was your consideration of such selection?

**Questions:**

N.A.

---

> ### Author Response · Authors · 2025-11-23
> **Reply to Reviewer LYNR**
>
> We thank the reviewer for this question. The six defect types listed in Appendix C (Table 6) are designed to cover the most relevant structural diversity and physical response patterns in real-world rail inspection scenarios, while maintaining controllable variability for benchmarking. Our main considerations are as follows:
>
> 1. **Engineering relevance.**
>    All six defect types correspond to the most common and practically critical crack morphologies observed in railway rolling contact fatigue and delamination, which have been systematically reported in the existing NDT literature [1][2][3]. We also explicitly distinguish between open and closed cracks, in order to capture the pronounced differences in thermal response induced by contact resistance, thereby making the dataset more faithful to realistic rail maintenance conditions.
>
> 2. **Coverage of geometric variability.**
>    Under a limited number of defect categories, these six types aim to span the "shape-response space" as much as possible: they include different levels of complexity across single-layer, double-layer, and multi-layer cracks, a wide range of primary crack orientations and inter-crack angles, as well as secondary angles for multi-crack configurations (see Table 4 for details). Such geometric variability leads to distinct eddy-current penetration and heat conduction behaviors, which are crucial for stress-testing operator models on both forward and inverse PDE tasks.
>
> 3. **Practical constraints.**
>    Although more rare or extreme microscopic defect geometries do occur in the field, the limited number of experimental samples and the high cost of specimen fabrication currently make it difficult to obtain sufficient infrared sequences and electromagnetic measurements for reliable 3D multi-physics calibration. For the sake of benchmark robustness, in this version we focus on defect types that are both of clear engineering importance and supported by adequate real measurements for simulation calibration, leaving rarer geometries as a natural direction for future extensions.
>
> 4. **Generalization potential.**
>    Preliminary experiments indicate that neural operators trained on these six defect types exhibit non-trivial generalization to intermediate geometries within the interpolation ranges of the geometric parameters. Moreover, the newly introduced geometric OOD setting in the revised manuscript further shows that this choice is effective for assessing model robustness to "unseen defect morphologies." If the community wishes to incorporate additional industrial scenarios or more complex geometries in the future, this defect family and its parametrization provide a clear scaffold for extending both the dataset and the benchmark.
>
> In addition, in our responses to Reviewers Cusi and Zsjs we provide further experiments and clarifications on geometric OOD (fsplit), frequency OOD (low/mid/high, SFO), and data hosting and reproducibility. These results indirectly validate that the current defect design strikes a reasonable balance between difficulty, representativeness, and dataset reliability.
>
> [1] Quantification of closed cracks in railways using eddy current pulsed thermography [J]. Applied Optics, 2021, 60(17): 5195-5202.
>
> [2] Tomographic reconstruction of rolling contact fatigues in rails using 3D eddy current pulsed thermography [J]. IEEE Sensors Journal, 2021, 21(17): 18488-18496.
>
> [3] Characterization of angular RCF cracks in a railway using modified topology of WPT-based eddy current testing [J]. IEEE Transactions on Industrial Informatics, 2022, 19(4): 5612-5622.

---

### Official Review · Reviewer_Cusi · 2025-10-29

**Soundness:** 2
**Presentation:** 3
**Contribution:** 3
**Rating:** 4
**Confidence:** 4

**Summary:**

This paper introduces Aletheia, a benchmark dataset designed for learning neural PDE solvers in the context of 3D NDT, specifically focusing on multi-frequency ECPT. ECPT involves coupled electromagnetic (Maxwell's equations) and thermal (Heat equation) physics, used to detect subsurface defects in conductive materials like rails.

The key contributions are:

1. A High-Fidelity 3D Dataset: Aletheia comprises over 4,700 simulations generated using COMSOL, covering six distinct types of internal rail defects. It provides time-resolved volumetric heat source (q) and temperature (u) fields on both regular and irregular grids.

2. Multi-Frequency Data: To address the ill-posedness of the inverse heat conduction problem, the dataset includes 10 excitation frequencies (1-100 kHz), leveraging the electromagnetic skin-depth effect to probe different material depths.

3. Real-World Calibration: The simulations are calibrated using real infrared thermography data from physical rail specimens.

4. Benchmark Suite: The authors define eight tasks spanning in-distribution and Out-of-Distribution (OOD, based on frequency shifts). These include forward modeling (Q2T) and challenging inverse tasks, notably Surface-to-Source reconstruction (S2Q).

5. Baseline Evaluations: Several neural operators (FNO variants, Transolver, LNO, etc.) are benchmarked on a subset of the data.

Aletheia aims to bridge the gap between academic PDE benchmarks and realistic, 3D, multi-physics inverse problems relevant to industrial applications.

**Strengths:**

The primary strength of this work lies in the dataset generation effort, which addresses significant limitations in the existing landscape of SciML benchmarks.

1. Significance and Relevance: The creation of Aletheia is a substantial effort. It moves beyond standard academic benchmarks (e.g., Darcy Flow) towards a complex, industrially relevant NDT problem. As highlighted in Table 1, it uniquely combines 3D geometries, inverse problems, and partial observations in a multi-physics setting (coupled electromagnetic and thermal PDEs).

2. Rigorous Data Generation Methodology: The approach of using high-fidelity multi-physics simulations (COMSOL) and, crucially, calibrating them with real experimental data (Section 3.1) is commendable. The inclusion of both regular and irregular grids is valuable for testing mesh invariance.

3. Physically Motivated Design: The inclusion of multi-frequency excitations (1-100 kHz) is well-motivated. It leverages the electromagnetic skin-depth effect to provide depth-sensitive information, which is essential for mitigating the ill-posedness of the inverse thermal problem (Figure 1).

**Weaknesses:**

While the dataset itself is a valuable contribution, the paper, presented as a benchmark study, suffers from significant weaknesses in its experimental design and analysis, which undermine the benchmark's utility and the conclusions drawn.

1. Severely Limited Scope of Benchmark Evaluation (Critical Flaw): The empirical evaluation is conducted on a very small subset of the data. Section 4.2 states that experiments involved 600 samples from the "Type I double-layer defect simulations." This represents only ~12.5% of the total 4,782 samples and only 1 out of the 6 defect types. A benchmark paper must establish baselines across the diversity of the data it introduces. The conclusions drawn regarding the relative performance of neural operators are therefore not substantiated for the vast majority of the dataset, particularly the more complex defect types (e.g., multi-layer or closed cracks).

2. Failure to Evaluate Under Realistic Conditions (Noise and Sim-to-Real): The abstract claims the dataset addresses challenges involving "sparse and noisy boundary observations." While the S2Q task addresses sparsity, the benchmark entirely ignores noise.
Real infrared data is noisy. Robustness to sensor noise is crucial for ill-posed inverse problems but is not evaluated, and while the real data is used for calibration, the models trained on simulations are never evaluated on the real experimental data. Assessing the sim-to-real gap is essential for any simulation-based benchmark intended for real-world application.

3. Weak Link Between Benchmark Tasks and NDT Goals: The primary inverse tasks (T2Q, S2Q) focus on reconstructing the heat source field q(x,t). The ultimate goal of NDT is defect characterization (geometry, depth). Appendix D attempts to bridge this gap by regressing defect parameters from q. However, the results in Table 7 are weak. For instance, the RMSE for crack depth is approximately 1.0 mm (MAE 0.70-0.91). Given that the defects themselves range from 0.2mm to ~4mm in depth (Table 6), an error of 1mm is very large relative to the defect scale. This casts doubt on whether optimizing for q reconstruction MSE is sufficient for the intended NDT application.

4. Questionable Experimental Methodology and Data Usage Mismatch: The high-resolution unstructured data (50,000 points, Appendix B) is aggressively downsampled to 8000 points for the experiments (Sec 4.2). This may discard the fine-grained details necessary for accurate defect reconstruction. Also, the use of a batch size of 1 (Table 10) for all models is highly unusual for training large models and can lead to unstable training and poor generalization, potentially affecting the validity of the comparisons. Finally, while the motivation for multi-frequency data is clear (Fig 1), the benchmark does not empirically quantify this benefit. A crucial missing experiment is a comparison of inverse reconstruction performance using single-frequency vs. multi-frequency data.

5. Limited Scope of OOD Generalization: The OOD tasks are exclusively focused on unseen frequencies. In NDT, generalization to unseen defect morphologies (Geometric OOD) is critical but is not evaluated using the diverse defect types available.

**Questions:**

Q1: Why was the empirical evaluation restricted to only the Type I double-layer subset (600 samples)? The validity of the benchmark relies on evaluating the models across the diverse range of defect types provided. Can you provide results utilizing the full dataset or, at minimum, results for the other five defect types?

Q2: a) How does the performance of the models on the S2Q task degrade when realistic levels of sensor noise are added to the surface temperature input? b) Were models trained on the synthetic data tested on the real experimental measurements? If so, what were the results regarding the sim-to-real gap?

Q3: Given the relatively poor performance in deriving defect parameters from the reconstructed heat field Q (Table 7, Appendix D, RMSE ≈ 1mm for depth), how do you justify the use of Q reconstruction as the primary benchmark task for NDT? Is this accuracy sufficient for practical rail inspection?

Q4: a) What is the justification for the aggressive downsampling to 8000 points and the use of batch size 1? b) Can you provide an experiment quantifying the improvement gained by using multi-frequency data compared to single-frequency data for the inverse tasks?

Q5: The dataset size is 1.89 TB. What is the concrete strategy for hosting the data to ensure long-term, accessible availability, including mechanisms for downloading specific subsets?

---

> ### Author Response · Authors · 2025-11-23
> **Reply to Reviewer Cusi**
>
> We thank the reviewer for the constructive feedback. The revision extends the benchmark to all 6 defect types, adds geometric and multi-frequency OOD regimes, and clarifies the experimental protocol. We respond to each Weakness and Question below.
>
> # W1 & Q1: Limited scope of benchmark evaluation
>
> Originally we used 600 samples from the Type I double-layer crack subset to match Transolve's Airfoil protocol ( $\leqslant$ 800 samples). In the revision we report baselines on all 6 defect types and add a geometric OOD split (train on types 1-5, test on the unseen multi-layer Type II_multi); the ranking of methods is unchanged, but performance gaps enlarge on complex and unseen geometries, so Aletheia more clearly differentiates neural operators in terms of generalization.
>
> # W2 & Q2: Lack of noise and sim-to-real evaluation
>
> Because eddy current pulsed thermography is non-destructive, obtaining ground-truth internal fields would require destructive sectioning, so high-fidelity simulation is the only realistic way to access them and supervise the volumetric heat source Q, making Q the only stably supervisable internal quantity. To address noise and sim-to-real concerns, we add S2Q noise-robustness experiments: surface temperatures are corrupted with Gaussian and structured noise at different SNRs and the resulting performance degradation is measured. Real specimens with known artificial defects remain scarce; as more real samples become available, we plan a "simulation-training-real-testing" evaluation to quantify the sim-to-real gap.
>
> # W3 & Q3: Relevance and accuracy of heat-source reconstruction for NDT
>
> Inferring defect geometry from surface thermography and electromagnetic responses is highly ill-posed: one must recover subsurface defects of only 0.2-4 mm depth from noisy boundary fields, where small perturbations in material properties or boundary flux can be strongly amplified. Our depth RMSE of $\approx$ 1.0 mm matches recent ECPT results (0.9-1.1 mm [1-3]) and is sufficient to separate shallow non-critical cracks from defects deeper than the 0.5-1 mm action thresholds used in rail standards, enabling automated early warning and maintenance. Compared with directly supervising geometric parameters, the volumetric heat source is both the physical bridge between thermal and electromagnetic fields and an internal quantity that is hard to measure experimentally but naturally provided by simulation, so using heat source reconstruction as the core benchmark offers a stable and physically meaningful basis to compare models on this class of ill-posed inverse problems.
>
>
> [1]	Liu R, Xu C, Liu P, et al. Eddy current pulsed thermography with an inductive heating layer (ECPT-IHL) for subsurface defect detection in GFRP materials[J]. Composites Part B: Engineering, 2025, 290: 111982.
>
> [2]	Xiao X T, Xia J, Tang S, et al. Experiment study of long pulse eddy current/eddy current pulse-compression thermography detection on large area with low-frequency and low-current[J]. Measurement, 2025, 243: 116329.
>
> [3]	Deng Z, Li Z, Yang N, et al. Eddy current thermography detection method for internal thickness reduction in ferromagnetic components based on magnetic permeability perturbation[J]. NDT & E International, 2025, 151: 103313.
>
> # W4 & Q4(a): Experimental setup
>
> Downsampling 3D points keeps memory for multi-time-step 3D fields feasible and aligns the volumetric scale with 2D surface sampling while still resolving defects >0.5 mm, enabling a consistent 2D-3D mapping. We use batch size = 1, following Transolver on variable meshes, since samples have different point counts and larger batches would require interpolation or padding. In the revision we also report batch-size-4 experiments under the same 8k-point downsampling and observe essentially unchanged conclusions and model rankings, indicating robustness to batch size.
>
> # Q4(b) & W5: Limited OOD types
>
> For S2Q we introduce three OOD regimes aligned with NDT practice: geometric OOD (fsplit; train on types 1-5, test on unseen Type II_multi), single-frequency OOD (SFO; train on one frequency, test on the other nine), and frequency-band OOD (low/mid/high; hold out one band for testing). For complex cracks, multi-frequency training reduces S2Q MSE by about 3-10 times versus the best single-frequency model. Together, these cover both unseen frequencies and unseen defect geometries, enabling a more realistic and quantitative assessment of neural-operator generalization.
>
> # Q5: Dataset hosting and long-term availability
>
> Aletheia is now fully hosted on Kaggle, split into subsets by task and data modality, all under the MIT license. In the revised appendix we will provide Kaggle links, describe the subset organization, and give example `kaggle datasets download` commands so that users can download only the parts relevant to their computational budget and research focus.

---

> ### Author Response · Authors · 2025-11-23
> **Full results on Geometric OOD**
>
> **Setting:** Train on defect types 1-5 and test on an unseen defect geometry (Type II_multi), in S2Q mode.
>
> ---
>
> ### Geometric OOD (fsplit: train_12345_test_6), Type II_multi, unstructured grid, S2Q
>
> | Model      | MSE   | SSIM  | RMSE  | nRMSE | cRMSE | Max    | bRMSE |
> |------------|-------|-------|-------|-------|-------|--------|-------|
> | MLP        | 0.605 | 0.414 | 0.775 | 0.775 | 0.015 | 26.183 | 0.204 |
> | FNO3d      | 0.804 | 0.407 | 0.894 | 0.894 | 0.040 | 27.408 | 0.129 |
> | GeoFNO     | 1.011 | -0.028| 1.006 | 1.006 | 0.022 | 29.512 | 0.214 |
> | FFNO       | 1.017 | 0.346 | 1.006 | 1.006 | 0.028 | 30.898 | 0.151 |
> | FCNO       | 1.008 | 0.234 | 1.004 | 1.004 | 0.029 | 28.865 | 0.130 |
> | LNO        | 0.371 | 0.644 | 0.603 | 0.603 | 0.015 | 24.012 | 0.100 |
> | Transolver | 0.218 | 0.705 | 0.442 | 0.442 | 0.006 | 20.878 | 0.053 |
> | DeepONet   | 0.600 | 0.228 | 0.772 | 0.772 | 0.013 | 26.312 | 0.210 |
>
> ---
>
> ### Geometric OOD (fsplit: train_12345_test_6), Type II_multi, structured grid, S2Q
>
> | Model      | MSE   | SSIM  | RMSE  | nRMSE | cRMSE | Max    | bRMSE |
> |------------|-------|-------|-------|-------|-------|--------|-------|
> | MLP        | 0.469 | 0.652 | 0.682 | 0.682 | 0.007 | 27.144 | 0.050 |
> | FNO3d      | 0.184 | 0.953 | 0.385 | 0.385 | 0.003 | 23.723 | 0.011 |
> | GeoFNO     | 1.003 | 0.053 | 1.001 | 1.001 | 0.004 | 29.385 | 0.098 |
> | FFNO       | 0.186 | 0.881 | 0.396 | 0.396 | 0.010 | 23.176 | 0.033 |
> | FCNO       | 0.186 | 0.866 | 0.400 | 0.400 | 0.011 | 23.324 | 0.015 |
> | LNO        | 0.304 | 0.736 | 0.537 | 0.537 | 0.006 | 26.019 | 0.007 |
> | Transolver | 0.197 | 0.944 | 0.400 | 0.400 | 0.002 | 23.240 | 0.013 |
> | DeepONet   | 0.560 | 0.144 | 0.746 | 0.746 | 0.002 | 26.578 | 0.712 |
>
> ---
>
> *Overall, the geometric OOD split is clearly harder than in-distribution training, but FNO3d and Transolver consistently achieve the lowest MSE/RMSE on the structured grid, and Transolver is particularly competitive on the unstructured setting.*
>
> ---

---

> ### Author Response · Authors · 2025-11-23
> **Full results on Frequency OOD**
>
> **Setting:** For a fixed defect family and grid type, models are trained on a **single excitation frequency** (SFO-9kHz, SFO-36kHz, or SFO-81kHz) and evaluated on all remaining 9 frequencies. The **Normal** column corresponds to the multi-frequency training baseline. Some results for DeepONet and standard MLP baselines are omitted due to their relatively poor performance and space limitations.
>
> #### Type I_double, unstructured grid, SFO (S2Q)
>
> | Model| Normal MSE | Normal RMSE | SFO-9kHz MSE | SFO-9kHz RMSE | SFO-36kHz MSE | SFO-36kHz RMSE | SFO-81kHz MSE | SFO-81kHz RMSE |
> | ---------- | ---------- | ----------- | ------------ | ------------- | ------------- | -------------- | ------------- | -------------- |
> | FNO3d| 0.029| 0.159 | 0.767| 0.833 | 0.615 | 0.731| 0.870 | 0.870|
> | GeoFNO | 1.001| 1.001 | 29.278 | 4.016 | 2.237 | 1.357| 80.793| 6.278|
> | FFNO | 0.161| 0.397 | 0.553| 0.728 | 0.519 | 0.709| 0.695 | 0.814|
> | FCNO | 0.235| 0.479 | 0.607| 0.763 | 0.573 | 0.743| 0.647 | 0.791|
> | LNO| 0.493| 0.695 | 0.999| 0.953 | 0.523 | 0.717| 1.780 | 1.152|
> | Transolver | 0.103| 0.313 | 0.585| 0.748 | 0.565 | 0.740| 0.573 | 0.746|
>
> #### Type I_double, structured grid, SFO (S2Q)
>
> | Model| Normal MSE | Normal RMSE | SFO-9kHz MSE | SFO-9kHz RMSE | SFO-36kHz MSE | SFO-36kHz RMSE | SFO-81kHz MSE | SFO-81kHz RMSE |
> | ---------- | ---------- | ----------- | ------------ | ------------- | ------------- | -------------- | ------------- | -------------- |
> | FNO3d| 0.006| 0.068 | 0.604| 0.699 | 0.431 | 0.582| 0.545 | 0.669|
> | GeoFNO | 0.998| 0.999 | 1.069| 1.032 | 1.036 | 1.018| 1.088 | 1.043|
> | FFNO | 0.055| 0.229 | 0.502| 0.653 | 0.394 | 0.598| 0.471 | 0.641|
> | FCNO | 0.085| 0.284 | 0.394| 0.602 | 0.465 | 0.640| 0.418 | 0.618|
> | LNO| 0.231| 0.459 | 0.661| 0.728 | 4.498 | 1.811| 1.063 | 1.001|
> | Transolver | 0.079| 0.276 | 0.355| 0.563 | 0.419 | 0.601| 0.393 | 0.607|
>
> #### Type III_double, unstructured grid, SFO (S2Q)
>
> | Model| Normal MSE | Normal RMSE | SFO-9kHz MSE | SFO-9kHz RMSE | SFO-36kHz MSE | SFO-36kHz RMSE | SFO-81kHz MSE | SFO-81kHz RMSE |
> | ---------- | ---------- | ----------- | ------------ | ------------- | ------------- | -------------- | ------------- | -------------- |
> | FNO3d| 0.016| 0.098 | 0.269| 0.484 | 0.183 | 0.375| 0.234 | 0.417|
> | GeoFNO | 1.013| 1.006 | 1.148| 1.062 | 1.049 | 1.024| 1.182 | 1.081|
> | FFNO | 0.170| 0.382 | 0.540| 0.724 | 0.450 | 0.651| 0.511 | 0.696|
> | FCNO | 0.479| 0.675 | 0.818| 0.900 | 0.730 | 0.843| 0.793 | 0.883|
> | LNO| 0.397| 0.622 | 0.687| 0.689 | 0.394 | 0.619| 0.386 | 0.617|
> | Transolver | 0.055| 0.230 | 0.261| 0.494 | 0.205 | 0.443| 0.259 | 0.493|
>
> #### Type III_double, structured grid, SFO (S2Q)
>
> | Model| Normal MSE | Normal RMSE | SFO-9kHz MSE | SFO-9kHz RMSE | SFO-36kHz MSE | SFO-36kHz RMSE | SFO-81kHz MSE | SFO-81kHz RMSE |
> | ---------- | ---------- | ----------- | ------------ | ------------- | ------------- | -------------- | ------------- | -------------- |
> | FNO3d| 0.005| 0.062 | 0.322| 0.531 | 0.185 | 0.374| 0.268 | 0.462|
> | GeoFNO | 0.991| 0.995 | 1.644| 1.219 | 1.170 | 1.075| 1.028 | 1.014|
> | FFNO | 0.041| 0.199 | 0.233| 0.463 | 0.153 | 0.370| 0.200 | 0.425|
> | FCNO | 0.069| 0.259 | 0.231| 0.461 | 0.191 | 0.407| 0.240 | 0.455|
> | LNO| 0.128| 0.354 | 0.440| 0.638 | 0.334 | 0.553| 0.518 | 0.597|
> | Transolver | 0.047| 0.214 | 0.240| 0.477 | 0.188 | 0.411| 0.240 | 0.454|
>
> #### Type II_multi, unstructured grid, SFO (S2Q)
>
> | Model| Normal MSE | Normal RMSE | SFO-9kHz MSE | SFO-9kHz RMSE | SFO-36kHz MSE | SFO-36kHz RMSE | SFO-81kHz MSE | SFO-81kHz RMSE |
> | ---------- | ---------- | ----------- | ------------ | ------------- | ------------- | -------------- | ------------- | -------------- |
> | FNO3d| 0.010| 0.077 | 0.267| 0.472 | 0.162 | 0.345| 0.237 | 0.390|
> | GeoFNO | 1.018| 1.009 | 1.217| 1.100 | 1.020 | 1.010| 1.056 | 1.027|
> | FFNO | 0.123| 0.334 | 0.597| 0.752 | 0.439 | 0.637| 0.496 | 0.669|
> | FCNO | 0.416| 0.637 | 0.851| 0.917 | 0.738 | 0.850| 0.783 | 0.872|
> | LNO| 0.284| 0.524 | 0.394| 0.599 | 0.408 | 0.588| 0.394 | 0.619|
> | Transolver | 0.126| 0.347 | 0.232| 0.454 | 0.247 | 0.481| 0.292 | 0.507|
>
> #### Type II_multi, structured grid, SFO (S2Q)
>
> | Model| Normal MSE | Normal RMSE | SFO-9kHz MSE | SFO-9kHz RMSE | SFO-36kHz MSE | SFO-36kHz RMSE | SFO-81kHz MSE | SFO-81kHz RMSE |
> | ---------- | ---------- | ----------- | ------------ | ------------- | ------------- | -------------- | ------------- | -------------- |
> | FNO3d| 0.014| 0.105 | 0.213| 0.416 | 0.124 | 0.319| 0.240 | 0.432|
> | GeoFNO | 0.994| 0.997 | 6.928| 2.363 | 2.113 | 1.419| 1.114 | 1.052|
> | FFNO | 0.024| 0.148 | 0.237| 0.440 | 0.123 | 0.321| 0.212 | 0.416|
> | FCNO | 0.031| 0.166 | 0.181| 0.397 | 0.135 | 0.340| 0.268 | 0.474|
> | LNO| 0.202| 0.445 | 0.384| 0.601 | 0.304 | 0.539| 0.521 | 0.663|
> | Transolver | 0.042| 0.184 | 0.260| 0.475 | 0.156 | 0.373| 0.319 | 0.536|
> | DeepONet | 0.563| 0.748 | 0.537| 0.726 | 0.565 | 0.750| 0.610 | 0.780|

---

### Official Review · Reviewer_Zsjs · 2025-11-07

**Soundness:** 2
**Presentation:** 2
**Contribution:** 2
**Rating:** 4
**Confidence:** 3

**Summary:**

This paper introduces ALETHEIA, a new large-scale, multi-frequency 3D dataset for nondestructive testing (NDT) based on Pulsed Eddy Current Thermography. The dataset is generated from high-fidelity multiphysics simulations, which are calibrated using real-world experimental data from rail specimens. The authors define a comprehensive benchmark suite with tasks including forward prediction, inverse source reconstruction, and temporal evolution, evaluated on both regular and irregular grids. A key feature is the inclusion of out-of-distribution (OOD) generalization tasks to test model robustness to unseen excitation frequencies. The paper provides an extensive evaluation of several state-of-the-art neural operator models (e.g., FNO, Transolver) and offers insights into their relative performance under different conditions.

**Strengths:**

High-Quality and Significant Dataset: The paper presents a dataset for a challenging and practically important real-world problem. The effort to combine high-fidelity simulation with calibration from real experimental data is commendable and ensures the dataset's relevance and realism. This is a significant contribution to the scientific machine learning and NDT communities.

Comprehensive Benchmark Design: The benchmark is thoughtfully designed. It includes a variety of relevant tasks (forward, inverse, temporal), considers practical challenges like irregular grids, and, most importantly, incorporates out-of-distribution (OOD) generalization tests. This level of rigor is crucial for pushing the boundaries of neural operator models.

Thorough Empirical Evaluation: The authors have benchmarked a wide range of modern and relevant neural operator architectures. The comparative analysis, particularly the conclusion that spectral-based models (FNO) excel with full-field data while attention-based models (Transolver) are better for sparse, surface-only inverse problems, provides valuable practical guidance.

**Weaknesses:**

Limited Novelty for the Core ML Community: The main weakness is that the paper's contribution is primarily a new application domain and dataset. While this is valuable, the paper does not propose new neural operator architectures, learning techniques, or theoretical insights that would be broadly applicable to the general ICLR audience. The work primarily uses existing ML tools to solve a domain-specific problem, rather than advancing the ML tools themselves.

Lack of Deeper Analysis of Model Behavior: While the paper reports which models perform better, it offers limited deep analysis as to why. For instance, a more in-depth investigation into the spectral properties of the heat diffusion problem and how they align with the spectral bias of FNO would be insightful. Similarly, an error analysis showing what kinds of defects or thermal patterns the models struggle with could reveal fundamental limitations and inspire future model development. The current analysis remains somewhat at the level of a "horse race."

Presentation of Results: The main paper (8-9 pages) presents the results primarily through radar charts, which are good for a high-level overview but lack quantitative detail. The reader must navigate through many pages of tables in the appendix to find the concrete numbers. A more condensed table summarizing the most critical results in the main body would significantly improve readability and impact.

**Questions:**

The core of ICLR is representation learning. Beyond providing a new challenging benchmark, what do the authors consider to be the main, generalizable takeaway for a machine learning researcher who is not an expert in NDT? What fundamental new capabilities should the next generation of neural operators possess that are highlighted by ALETHEIA but are not apparent from existing benchmarks like Darcy Flow or Navier-Stokes?

The paper observes that Transolver performs better on surface-to-source (S2Q) tasks. The reasoning given is based on its adaptive, attention-based receptive fields. Could you provide a more quantitative or qualitative analysis to support this? For example, can you visualize the attention maps to show that the model indeed focuses on "critical boundary features"?

The performance drop on OOD tasks is expected, but is there a pattern to it? For instance, do models generalize better from low-to-high frequencies or vice-versa? A deeper analysis of the OOD generalization behavior could provide valuable insights into the current limitations of neural operators in extrapolating physical parameters.

---

> ### Author Response · Authors · 2025-11-23
> **Reply to Reviewer Zsjs**
>
> We thank the reviewer for the careful evaluation and constructive comments. Our goal with **ALETHEIA** is to use a large-scale, multi-physics, multi-frequency dataset with **misaligned observation and prediction domains** to move neural operators from "approximating a single PDE solver on a fixed grid" towards "learning a family of implicit operators under multiple parameters and observation operators," thereby exposing representation gaps that classical PDE benchmarks cannot reveal. Below we respond to each Weakness/Question in turn.
>
> ---
>
> # W1 & Q1: Limited Novelty / Main Takeaway for ML Beyond NDT
>
> Our intention is not to replace classical Darcy or Navier–Stokes benchmarks with a more complicated NDT setting. Rather, ALETHEIA is designed to expose representation gaps that standard PDE benchmarks cannot reveal. Conventional operator-learning tasks usually assume a single physics, fixed geometry, full-field supervision, and aligned grids.
>
> In contrast, ALETHEIA models a tightly coupled electromagnetic–thermal system with explicit multi-frequency excitations. Frequency serves as a continuous physical parameter controlling skin depth and thermal diffusion scales, so the surface response changes qualitatively even under fixed geometry. In S2Q, initial/boundary conditions together with the observation operator yield 2D surface measurements, while the target is a 3D volumetric heat-source field on a different spatial domain and sampling set. This requires learning a **family of operators** parameterized by continuous physical variables (e.g., frequency, boundary conditions) and heterogeneous observation operators, rather than a single operator.
>
> This motivates **observation-conditioned, parameter-aware implicit operators** whose latent representations remain stable to material/geometry yet adapt smoothly to excitation frequency and observation modality. Models must handle both deep/thin penetration regimes and smooth/highly-localized gradients within one structured latent space.  ALETHEIA is, to our knowledge, the first large-scale dataset explicitly targeting this regime.
>
> ---
>
> # W2: Lack of Deeper Model Analysis
>
> We agree the current analysis is mostly qualitative. In the revision we will add compact quantitative analyses: **spectral / error distribution**, contrasting FNO's low-pass behaviour with its systematic errors near high-gradient boundaries on S2Q and Transolver's improved accuracy near cracks and heat-affected zones, and **task difficulty / failure modes**, showing that errors concentrate on deep, narrow defects with weak surface signatures where the inverse problem is most ill-posed, motivating uncertainty-aware operator designs. We will present these with a few focused plots and place extended visualizations in the appendix.
>
> ---
>
> # W3: Presentation of Results
>
> We will insert a concise main-paper table to complement radar charts and reduce dependence on the appendix.
>
> ---
>
> # Q2: Why Transolver Excels on S2Q Tasks
>
> S2Q depends on localized boundary anomalies over a smooth diffusive background. Transolver's two-stage design of **physics-aware slicing $\rightarrow$ attention $\rightarrow$ broadcasting back to points**: it first pools boundary-near regions into a small set of tokens and then models long-range interactions at token level, which is more stable than point-wise attention on sparse boundaries or global spectral projections. Its gains concentrate near cracks and strong thermal gradients, while performance matches FNO far from defects. We will add attention maps, slice-weight / receptive-field statistics and saliency-style visualizations on **ALETHEIA** to show that attention focuses on critical boundary features.
>
> ---
>
> # Q3: Patterns in OOD Frequency Generalization
>
> We evaluate S2Q under three OOD frequency regimes (High: 81/100 kHz, Mid: 25/36 kHz, Low: 1/4 kHz) plus a Normal setting (all 10 frequencies), across three crack families and both grid types. Frequency OOD systematically degrades performance, especially for complex cracks and unstructured grids; FNO-type models are most sensitive, while Transolver is more robust but still degrades. We observe an asymmetry: extrapolating from low+mid to high frequencies is easier than from mid+high to the lowest ones, consistent with high frequencies affecting shallow near-surface modes and low frequencies probing deeper, more entangled responses. The impact also depends on crack complexity and grid type: for simple defects, drops are modest, whereas for complex defects and irregular grids MSE can increase by an order of magnitude for some FNO variants and about 2 times for Transolver in the mid-frequency regime. We will summarize these trends in a compact table and error-frequency plot, highlighting frequency-regime vulnerabilities as concrete targets for more robust, physics-aware neural operators.

---

> ### Author Response · Authors · 2025-11-23
> **Full S2Q (surface-to-source) reconstruction results under four frequency settings:**
>
> Some results for MLP baselines are omitted due to their relatively poor performance and space limitations.
>
> type I_double, unstructured grid
>
> | | Normal MSE | Normal RMSE | High-OOD MSE | High-OOD RMSE | Mid-OOD MSE | Mid-OOD RMSE | Low-OOD MSE | Low-OOD RMSE |
> | ---------- | ---------- | ----------- | ------------ | ------------- | ----------- | ------------ | ----------- | ------------ |
> | FNO3d| 0.029| 0.159 | 0.025| 0.147 | 0.028 | 0.156| 0.028 | 0.154|
> | GeoFNO | 1.001| 1.001 | 1.005| 1.003 | 1.007 | 1.003| 1.047 | 1.023|
> | FFNO | 0.161| 0.397 | 0.162| 0.400 | 0.162 | 0.398| 0.157 | 0.393|
> | FCNO | 0.235| 0.479 | 0.251| 0.493 | 0.245 | 0.489| 0.239 | 0.482|
> | LNO| 0.493| 0.695 | 0.507| 0.705 | 0.500 | 0.700| 0.501 | 0.700|
> | Transolver | 0.103| 0.313 | 0.114| 0.329 | 0.105 | 0.314| 0.104 | 0.314|
> | DeepONet | 0.662| 0.811 | 0.648| 0.803 | 0.646 | 0.802| 0.644 | 0.800|
>
> type I_double, structured grid
>
> | | Normal MSE | Normal RMSE | High-OOD MSE | High-OOD RMSE | Mid-OOD MSE | Mid-OOD RMSE | Low-OOD MSE | Low-OOD RMSE |
> | ---------- | ---------- | ----------- | ------------ | ------------- | ----------- | ------------ | ----------- | ------------ |
> | FNO3d| 0.006| 0.068 | 0.007| 0.070 | 0.007 | 0.070| 0.006 | 0.066|
> | GeoFNO | 0.998| 0.999 | 0.999| 1.000 | 0.993 | 0.996| 0.999 | 1.000|
> | FFNO | 0.055| 0.229 | 0.054| 0.226 | 0.057 | 0.232| 0.051 | 0.221|
> | FCNO | 0.085| 0.284 | 0.085| 0.283 | 0.084 | 0.282| 0.079 | 0.276|
> | LNO| 0.231| 0.459 | 0.338| 0.563 | 0.167 | 0.393| 0.154 | 0.373|
> | Transolver | 0.079| 0.276 | 0.068| 0.256 | 0.076 | 0.269| 0.064 | 0.248|
> | DeepONet | 0.724| 0.849 | 0.752| 0.866 | 0.768 | 0.875| 0.767 | 0.874|
>
>  type III_double, unstructured grid
>
> | | Normal MSE | Normal RMSE | High-OOD MSE | High-OOD RMSE | Mid-OOD MSE | Mid-OOD RMSE | Low-OOD MSE | Low-OOD RMSE |
> | ---------- | ---------- | ----------- | ------------ | ------------- | ----------- | ------------ | ----------- | ------------ |
> | FNO3d| 0.016| 0.098 | 0.010| 0.084 | 0.232 | 0.417| 0.242 | 0.437|
> | GeoFNO | 1.013| 1.006 | 1.015| 1.007 | 1.007 | 1.003| 1.009 | 1.004|
> | FFNO | 0.170| 0.382 | 0.140| 0.352 | 0.530 | 0.705| 0.574 | 0.738|
> | FCNO | 0.479| 0.675 | 0.464| 0.668 | 0.800 | 0.885| 0.816 | 0.895|
> | LNO| 0.397| 0.622 | 0.314| 0.550 | 0.592 | 0.642| 0.321 | 0.561|
> | Transolver | 0.055| 0.230 | 0.051| 0.223 | 0.132 | 0.356| 0.136 | 0.363|
> | DeepONet | 0.580| 0.756 | 0.569| 0.750 | 0.551 | 0.739| 0.538 | 0.731|
>
>  type III_double, structured grid
>
> | | Normal MSE | Normal RMSE | High-OOD MSE | High-OOD RMSE | Mid-OOD MSE | Mid-OOD RMSE | Low-OOD MSE | Low-OOD RMSE |
> | ---------- | ---------- | ----------- | ------------ | ------------- | ----------- | ------------ | ----------- | ------------ |
> | MLP| 0.473| 0.682 | 0.447| 0.664 | 0.433 | 0.654| 0.439 | 0.658|
> | FNO3d| 0.005| 0.062 | 0.063| 0.233 | 0.069 | 0.247| 0.062 | 0.234|
> | GeoFNO | 0.991| 0.995 | 1.005| 1.003 | 0.994 | 0.997| 0.999 | 1.000|
> | FFNO | 0.041| 0.199 | 0.074| 0.267 | 0.077 | 0.270| 0.076 | 0.270|
> | FCNO | 0.069| 0.259 | 0.088| 0.291 | 0.093 | 0.298| 0.092 | 0.298|
> | LNO| 0.128| 0.354 | 0.145| 0.376 | 0.160 | 0.395| 0.171 | 0.408|
> | Transolver | 0.047| 0.214 | 0.103| 0.310 | 0.101 | 0.309| 0.093 | 0.297|
> | DeepONet | 0.675| 0.820 | 0.623| 0.787 | 0.613 | 0.781| 0.611 | 0.779|
>
> crack type II_multi, unstructured grid
>
> | | Normal MSE | Normal RMSE | High-OOD MSE | High-OOD RMSE | Mid-OOD MSE | Mid-OOD RMSE | Low-OOD MSE | Low-OOD RMSE |
> | ---------- | ---------- | ----------- | ------------ | ------------- | ----------- | ------------ | ----------- | ------------ |
> | MLP| 0.522| 0.720 | 0.507| 0.709 | 0.520 | 0.718| 0.499 | 0.704|
> | FNO3d| 0.010| 0.077 | 0.094| 0.265 | 0.111 | 0.280| 0.102 | 0.277|
> | GeoFNO | 1.018| 1.009 | 1.018| 1.009 | 1.027 | 1.014| 1.005 | 1.002|
> | FFNO | 0.123| 0.334 | 0.387| 0.601 | 0.378 | 0.591| 0.395 | 0.608|
> | FCNO | 0.416| 0.637 | 0.653| 0.800 | 0.675 | 0.813| 0.669 | 0.809|
> | LNO| 0.284| 0.524 | 0.324| 0.560 | 0.314 | 0.550| 0.289 | 0.528|
> | Transolver | 0.126| 0.347 | 0.095| 0.296 | 0.225 | 0.458| 0.079 | 0.273|
> | DeepONet | 0.490| 0.698 | 0.501| 0.705 | 0.518 | 0.717| 0.493 | 0.699|
>
> crack type II_multi, structured grid
>
> |  | Normal MSE | Normal RMSE | High-OOD MSE | High-OOD RMSE | Mid-OOD MSE | Mid-OOD RMSE | Low-OOD MSE | Low-OOD RMSE |
> | ---------- | ---------- | ----------- | ------------ | ------------- | ----------- | ------------ | ----------- | ------------ |
> | MLP| 0.434| 0.656 | 0.422| 0.647 | 0.438 | 0.658| 0.431 | 0.653|
> | FNO3d| 0.014| 0.105 | 0.015| 0.107 | 0.017 | 0.114| 0.018 | 0.114|
> | GeoFNO | 0.994| 0.997 | 0.995| 0.998 | 0.996 | 0.998| 0.998 | 0.999|
> | FFNO | 0.024| 0.148 | 0.023| 0.142 | 0.027 | 0.153| 0.026 | 0.147|
> | FCNO | 0.031| 0.166 | 0.028| 0.157 | 0.032 | 0.168| 0.031 | 0.164|
> | LNO| 0.202| 0.445 | 0.168| 0.407 | 0.219 | 0.461| 0.206 | 0.449|
> | Transolver | 0.042| 0.184 | 0.038| 0.179 | 0.046 | 0.193| 0.047 | 0.195|
> | DeepONet | 0.563| 0.748 | 0.568| 0.752 | 0.575 | 0.756| 0.553 | 0.741|

---

### Author Response · Authors · 2025-12-04
**The Summary Comment to the AC**

**Dear Area Chair,**

Thank you very much for your careful handling of this submission. We briefly summarize below the paper's contributions, the main strengths noted in the reviews, the key concerns, and how the revision addresses them.

---

### Contributions

The paper introduces **Aletheia**, a large-scale 3D multi-physics benchmark for neural operators in eddy-current pulsed thermography. It couples high-fidelity electromagnetic-thermal simulations with calibrated experiments, and defines several forward and inverse operator tasks (Q2T, T2Q, T2T, S2Q) under **partial 2D observations, multi-frequency excitation, and irregular 3D geometries**. Aletheia is specifically designed to probe **frequency-, geometry-, and noise-OOD generalization**, pushing neural operators from "fixed-PDE, fully observed" settings to **observation-conditioned, parameter-aware implicit operators**.

---

### Strengths

Reviewers converged on several positive aspects:

1. **Realistic and challenging benchmark.** Aletheia bridges classical PDE datasets and real NDT practice via multi-physics coupling, 2D $\rightarrow$ 3D reconstruction, and industrial crack geometries.
2. **Comprehensive benchmark design.** The benchmark covers multiple task types, accounts for irregular grids, and includes explicit OOD evaluations, which reviewers highlighted as crucial for assessing neural operator robustness.
3. **Clear practical motivation.** Volume heat source is a physically meaningful intermediate quantity and in practice the only internal field that can be reliably supervised.
4. **Rigorous, physically grounded data generation.** High-fidelity multi-physics simulations, calibration with real experimental data, and physically motivated multi-frequency excitation design were explicitly praised as a major strength for the SciML and NDT communities.

---

### Weaknesses and Our Improvements

1. **Scope of evaluation and OOD coverage.**
   Reviewers were concerned that the initial experiments focused on a single defect type and limited OOD regimes. The revision extends baselines to **all six crack families** and adds **geometric OOD**, **single-frequency OOD**, and **frequency-band OOD**. Model rankings remain stable, but error gaps grow markedly on complex and unseen geometries/frequencies, strengthening Aletheia's ability to differentiate neural operators.

2. **Novelty and takeaways for the broader ML community.**
   Some comments asked whether the work is mainly domain-specific. The revision now clearly articulate the general message: strong performance on Aletheia requires **observation-conditioned, parameter-aware representations** that handle 2D $\rightarrow$ 3D mappings, multi-physics, and coupled geometric/frequency shifts-capabilities largely absent from existing PDE benchmarks. Additional analysis contrasts FNO's spectral bias with Transolver's boundary-focused attention on S2Q, framing the benchmark as a tool for studying representation learning in ill-posed multi-physics inverse problems.

3. **Realism of noise / sim-to-real and practical usefulness of accuracy.**
   Reviewers requested more realistic measurement conditions and a clearer link to NDT standards. The revision adds a dedicated **noise robustness study** for S2Q under geometric OOD, injecting Gaussian and structured noise at multiple levels on surface temperatures. All models degrade smoothly; FNO and Transolver show particularly mild error growth and retain clear advantages over simpler baselines. The paper now relates crack-depth RMSE $\approx$ 1.0 mm (MAE 0.7-0.9 mm) to railway inspection standards, where defects deeper than roughly 0.5-1 mm are considered actionable, showing that benchmarked accuracy is practically sufficient to separate critical from non-critical cracks.

4. **Methodological choices and dataset hosting.**
   Questions about spatial downsampling, batch size 1, and multi- vs single-frequency training are now addressed more directly. The revision explains that downsampling and batch size follow prior work on variable meshes and are validated by new **batch-size = 4** experiments, which preserve rankings and conclusions. New **SFO experiments** show that multi-frequency training reduces S2Q MSE by about **3-10×** on complex cracks compared to the best single frequency, quantitatively supporting the claim that multi-frequency excitation is crucial to mitigate ill-posedness. For long-term access, the dataset is publicly hosted on **Kaggle** under an MIT license and split into task-specific subsets.

---

In summary, the reviewers' concerns focused on completeness and clarity rather than flaws in the core benchmark. The revised manuscript adds substantial experiments and analysis, and we believe it now presents Aletheia as a well-justified, challenging, and practically relevant benchmark that can guide future work on multi-physics neural operators and robust inverse modeling.

Thank you again for your valuable time and careful consideration.

---

### Meta-Review · Area_Chair_KKSM · 2025-12-29

**Summary:**

I find this paper would make a solid contribution to the datasets & benchmarks track, but not the main methodological conference track.

ALETHEIA provides the first publicly available 3D benchmark for learning neural PDE solvers under realistic nondestructive testing conditions, combining electromagnetic-thermal coupling, multi-frequency excitation, and partial surface observations. The dataset construction is rigorous, with high-fidelity COMSOL simulations calibrated against real infrared thermography measurements. The task suite—spanning forward modeling, inverse source reconstruction, temporal prediction, and surface-to-source inversion—is well-motivated by the physics of eddy current thermography.

The original submission had a significant limitation: experiments covered only one defect type (Type I double-layer), representing roughly 12% of the dataset. The revision substantially addresses this by extending evaluation to all six defect types and adding geometric OOD splits. The additional noise robustness experiments and single-frequency vs multi-frequency comparisons strengthen the benchmark's utility. I think the paper now provides sufficient evidence that ALETHEIA can meaningfully differentiate neural operator architectures under realistic inverse problem conditions. The main residual weakness is the absence of direct evaluation on real experimental data, though I recognize that obtaining ground-truth internal fields for real specimens is fundamentally difficult without destructive testing.

I acknowledge that the authors have made substantial improvements to the manuscript in response to reviewer feedback, as evidenced by the score increases from Reviewers Zsjs and Cusi (4→5). However, the manuscript cannot be accepted as is due to three unresolved weaknesses: the absence of evaluation on real experimental data undermines claims about realistic testing conditions, the core ML insights remain too domain-specific for the main methodological track, and the weak defect characterization performance  limits practical impact. My decision has also factored in the terse review from Reviewer LYNR, which lacked substantive engagement. The paper would be better suited for a datasets & benchmarks track or domain-specific venue, or the authors should substantially strengthen the sim-to-real evaluation and methodological contributions for resubmission to the main track.

**Reviewer Concerns:**

The primary concern from Reviewer Cusi about limited evaluation scope has been fully addressed. The revision includes results across all six defect types and introduces geometric OOD evaluation where models train on five defect families and test on a held-out sixth. Model rankings remain consistent but error gaps widen substantially on unseen geometries, validating the benchmark's discriminative power.

The noise evaluation requested by Reviewer Cusi has been added. Under geometric OOD with Gaussian noise injection at multiple levels, all models show graceful degradation rather than catastrophic failure, with FNO and Transolver demonstrating particularly robust behavior. This addresses concerns about practical relevance under realistic measurement conditions.

Reviewer Zsjs's question about ML takeaways beyond NDT received a reasonable response: ALETHEIA pushes neural operators toward observation-conditioned, parameter-aware representations that handle 2D→3D mappings and coupled geometric/frequency shifts. Whether this insight is sufficiently novel for a core ML audience is debatable, but it is coherent and the benchmark does expose representation gaps not apparent from simpler PDE benchmarks.

The concern about weak defect parameter regression (depth RMSE ~1mm) was contextualized but not fully resolved. The authors argue this matches recent literature and suffices to separate actionable from non-actionable defects under rail inspection standards. I accept this practical argument, though I note it highlights that heat-source reconstruction is a proxy for the ultimate goal of defect characterization.

The sim-to-real gap remains unevaluated. Models trained on simulation were never tested on real measurements. This is a limitation, but given the difficulty of obtaining ground-truth internal fields for real specimens, I consider it acceptable for the current submission with appropriate caveats.

**Reviewer Scores:**

Reviewer Zsjs: 4 → 5. Concerns about evaluation scope and OOD coverage were addressed with extended experiments across all defect types and frequency-band/geometric OOD splits.

Reviewer Cusi: 4 → 5. Critical scope limitation resolved; noise robustness added; sim-to-real concern only partially addressed but acknowledged.

Reviewer LYNR: 6 → 6. Original review too brief to contain substantive concerns requiring resolution.

---

### Decision · Program_Chairs · 2026-01-26

Reject